# Mothers are more egocentric towards their own child's bodily feelings

Louise P. Kirsch[1,5 ✉], Michal Tanzer[2,5], Maria Laura Filippetti[3], Mariana von Mohr[4] & Aikaterini Fotopoulou[2 ✉]

Our emotional state can influence how we understand other people's emotions, leading to biases in social understanding. Yet emotional egocentric biases in specific relationships such as parent-child dyads, where not only understanding but also emotional and bodily regulation is key, remain relatively unexplored. To investigate these biases and control for sensory priors, we first conducted two experiments in dyads of adult strangers (total $N = 75$) using a bodily Emotional Egocentricity Task that enables simultaneous affective tactile stimulation within a dyad. We showed its effectiveness in eliciting both classical and sensory-controlled egocentric biases. We then recruited 68 mother-child dyads and found that mothers exhibit higher classical and sensory-controlled emotional egocentric biases towards their own child compared to an unfamiliar child. Results suggest that mothers tend to rely on their bodily feelings more when judging the states of their own child than those of other children, possibly consistent with their regulatory parental role.

[1] Université Paris Cité, CNRS, Integrative Neuroscience and Cognition Center, F-75006 Paris, France. [2] Research Department of Clinical, Educational and Health Psychology, University College London, London, UK. [3] Centre for Brain Science, Department of Psychology, University of Essex, Colchester, UK. [4] Lab of Action and Body, Department of Psychology, Royal Holloway, University of London, London, United Kingdom. [5] These authors contributed equally: Louise P. Kirsch, Michal Tanzer. ✉email: louise.kirsch@u-paris.fr; a.fotopoulou@ucl.ac.uk

The ability to understand other's mental and affective states is a fundamental part of social communications. To understand other minds, we often partly rely on our own concurrent mental states which we project, or externalise to others[1]. Conversely, our own experiences may be influenced by how we read and 'internalise' others' states[2,3]. Converging neuroimaging and neuromodulation evidence has supported and expanded upon these behavioural findings, revealing shared neural representations between self and other in various contexts: people acting and observing others' actions[4], thinking about themselves and others, or sharing other's feelings[5–7]. While debates about the precise mechanisms underlying the overlap in self-other processing persist[8,9], there is consensus that this overlap is effective for social understanding only when experiences are aligned. To enable successful social understanding beyond aligned, concurrent experiences, a distinction has to be made between one's own mental states and those of others. This distinction is referred to as the self-other distinction[7,10]. Failure to make this distinction during incongruent experiences (for instance, when one individual feels sad while another feels happy) can result in difficulties in inhibiting one's own mental states while judging the ones of others, leading to what is known as the egocentricity bias (EB[11,12]). This failure can also result in difficulties in inhibiting the influence of other people's mental states while judging our own, leading to what is known as the altercentricity bias (AB[13]).

The self-other distinction has been studied in both the sensorimotor and cognitive domains[14], such as our ability to inhibit automatic imitation of actions[10,15], or to comprehend the mental states of others even when they conflict with our own[7,12]. Utilising a paradigm of congruent versus incongruent affective tactile stimulation in experimental dyads of volunteers who were previously unacquainted, Silani and colleagues demonstrated that when individuals had to judge the emotional states of their unfamiliar partners, they were influenced by the affectively incongruent stimulation they were experiencing, leading to the so-called emotional EB[16]. These studies have consistently converged on the finding that the self-other distinction relies on brain areas such as the temporo-parietal junction, which also plays a pivotal role in mental inference and attribution[17]. Interestingly, more anterior parts of this region, particularly the right supramarginal gyrus[12,16,18], have been implicated in the more affective dimension of the self-other distinction, such as when we need to inhibit our own emotions in order to empathise with the experience of others[7,12,16].

Importantly, the ability for self-other distinction varies across the lifespan, with children, adolescents and older adults showing increased EB compared to young and middle-aged adults[12,13,19]. Most existing studies on self-other distinction have been conducted between participants who were previously unfamiliar with each other and therefore cannot account for important features of social understanding in established relationships. As a result, open questions remain regarding the influence of specific, proximal relationships on these biases. A pertinent and promising dyadic relationship to study this question, is that of parent and child, that is characterised by biobehavioural synchrony (i.e., coordinated multimodal signals[20–22]). Questionnaire-based studies have assessed similar biases within parent-child relationships, and found that parents can be positively and egocentrically biased in how they evaluate their children[23]. However, to our knowledge, it has not been experimentally studied, nor explored in relation to the perception and projection of bodily feelings. Parents' own emotions are known to have a unique role in the development of affective regulation of their children's emotions[22,24,25] and interoceptive inference (i.e., sensing and predicting their child's

physiological and emotional states[22,26]). In early infancy, there are requirements for regulatory biobehavioural synchrony between infants and their caregivers[20], with infants and caregivers engaging in the sharing and co-regulating of their embodied states[26]. Over time, these repeated synchronised interactions shape children's self and social understanding. Crucially, as children's cognition and selfhood mature, the role of the parent-child dyad and its regulatory function also evolves[24,26,27]. Parents do not just need to understand their children's current needs, but they are also predisposed to transfer information (i.e., 'natural pedagogy'[28]). This entails teaching their children anticipatory regulatory strategies based on cultural and other shared, social or physical contextual factors (in physiological terms referred to as 'allostatic' regulation[22,25,27,29,30]). For example, when a parent feels cold, they might suggest to their child to put on an additional layer of clothing or use an extra blanket at night, even if the child is currently feeling that they are warm enough. In such instances, parents' bodily emotional egocentricity, i.e. using their own bodily temperature to determine and regulate their child's body temperature - could, not necessarily signify egocentric bias or a failure of empathic understanding, but rather serve the purpose of anticipatory regulation.

To capture the potential presence of egocentric biases within parent-child dyads, namely the projection of our own bodily feelings onto our children[22,26], while also controlling for factors of familiarity and salience in unisensory perception between parents and children, we first developed a bodily, Emotional Egocentricity Task (Experiments 1a and 1b; see set up in Fig. 1c). We then tested this task in 68 mother-child dyads (Experiment 2; see set up details in Fig. 1d). Dyads of participants were tested in proximal space, using a set-up that allows tactile stimulation of participant's own unseen forearm at the same time as they are watching another participant's forearm being touched. This approach moves beyond the utilisation of a pictorial stimulus, as seen in previous tasks, to include a bodily vicarious touch stimulus (see also below and Fig. 1). Consequently, this setup facilitates a concurrent experience of self, through unisensory touch (felt touch) and other, through unisensory observed touch (vicarious seen touch). As depicted in Fig. 1, participants were asked to judge how pleasant was the touch they felt (self) or saw (other), without knowing in advance which question they would need to answer. This required their attention to be focused on both self and other experiences. This simultaneous sensory experience can either be congruent or incongruent in terms of pleasantness (pleasant or unpleasant; see the design and set-up in Fig. 1).

A large tradition of sensory and multisensory cognitive and neuroscientific research has studied vicarious (or mirror) sensory phenomena. Notably, studies within this domain have highlighted the unique effects of seeing body parts being stimulated, in contrast to other non-body stimuli, as well as many other embodiment effects on vicarious touch (e.g. refs. [31–33]). For example, neuroimaging studies have identified areas within the occipito-temporal cortex, such as the extrastriate body area[34], and fusiform body area[35] responding selectively to seen bodies and body parts, and a corresponding topographic map of viewed body parts throughout occipitotemporal cortex[36]. This specificity of the human body (both our own and that of others) as a visual stimulus and a representation, particularly during vicarious responding, is one of the two reasons we named this task the 'bodily Emotional Egocentricity Task'. This task extends beyond 'imagining' the other person being touched with a stimulus while one feels touch on their own body, it also encompasses the representation of the other person's body and the seen touch itself during the vicarious touch experience. The

## a. Design Main Task

| | | What Participant SEE = OTHER (O) | |
|---|---|---|---|
| | | **Pleasant** | **Unpleasant** |
| **What Participant FEEL = SELF (S)** | **Pleasant** | S O | S O |
| | **Unpleasant** | S O | S O |

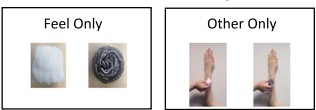

## c. Set-up Experiment 1

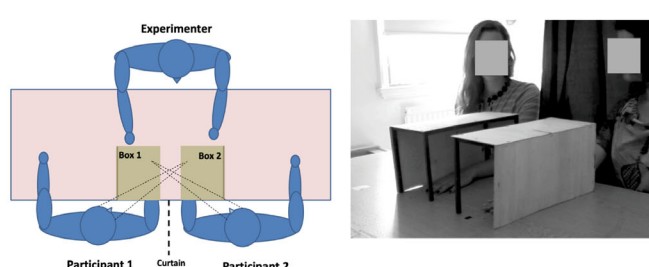

## b. Procedure

**1. Prior Estimates - Unisensory**

Feel Only Other Only

**2. Self and Other Pleasantness Estimates**

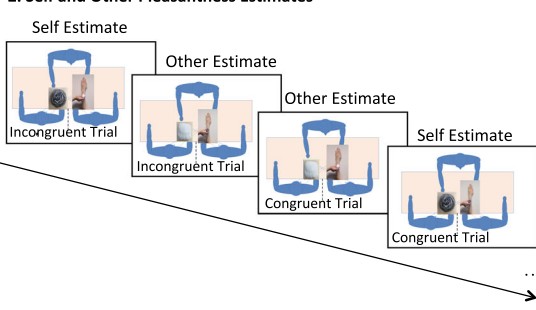

Self Estimate
Incongruent Trial

Other Estimate
Incongruent Trial

Other Estimate
Congruent Trial

Self Estimate
Congruent Trial

…

## d. Set- up Experiment 2

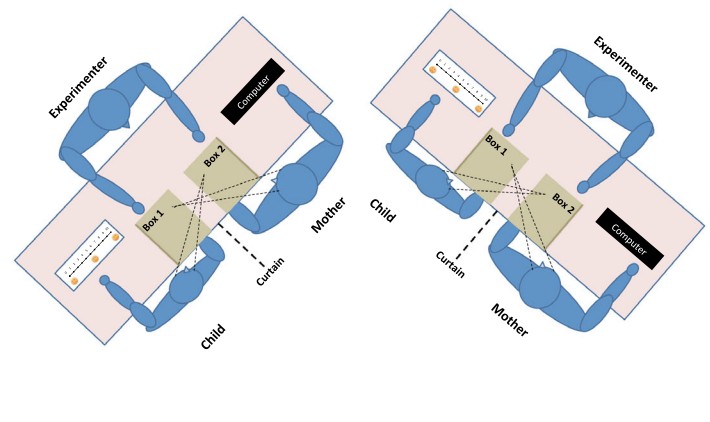

**Fig. 1 Experimental design and set-up. a** Experimental design. Valence of the fabric touching the self and the other being either pleasant/unpleasant and congruent/incongruent. Dyads of participants are tested in proximal space using a set-up (box closed on the top but open on the other participant's side) that allows tactile stimulation of participant's unseen arm at the same time as they are watching another participant's opposite arm being touched synchronously, allowing a concurrent, unisensory experience of self (tactile only) and other (vision only). These experiences can either be congruent or incongruent in pleasantness (pleasant or unpleasant), and unlike previous studies that have used affective images during congruent stimulation, both participants are seeing the other being touched directly on the body, as they are feeling touch in the corresponding body part. **b** Procedure. Prior to the simultaneous touch conditions, we tested unisensory (non-concurrent) perception for both the felt touch on the self and the seeing touch on the other, to get a measure of unisensory priors in participants (Prior estimates). Then, during the main task participants had to judge how pleasant was the touch they felt (self) or saw (other), without knowing in advance which question they will have to answer, so they had to focus on both self and other's experience. **c** Experimental set-up for Experiment 1a and 1b. with a pictorial representation of the set-up with two participants separated by a curtain. **d** Experiment 2 experimental set-up, enabling two Mother-Child dyads to receive separate but simultaneous tactile stimulation to their concealed forearm whilst observing tactile stimulation upon their partner.

second, related reason pertains to the capability of our task to provide subject-specific, important control (non-concurrent) measures for both the felt and the seen unisensory perception of touch. Various studies on vicarious, or mirror perception and visual enhancement of touch have shown that there are both intersubjective and intermodal effects on concurrent perception of felt and seen touch. While traditional emotional egocentricity tasks aim to mitigate some of these effects by employing emotionally 'congruent' conditions as a baseline, these tasks still remain within the context of concurrent, multimodal conditions. By contrast, our task gives the opportunity to have this control, as well as unisensory measures for each modality, without the concurrent stimulation. Indeed, in addition to the congruent and incongruent conditions tested during the main task, we also tested unisensory (non-concurrent) perception for both the felt touch on the self and the seen touch on the other. These control conditions serve the purpose of disentangling sensory from the multisensory effects inherent to the task.

Importantly, none of these conditions are susceptible to the top-down effects that object or animal stimuli could potentially exert on participants, as often encountered in the original egocentricity tasks that use pictorial stimuli for the seen touch and to characterise the felt touch.

Furthermore, our task not only allows for participants to see the touch the other person is receiving on their body with a given material, but it also allows us to match the specific stimulation parameters of the tactile and the vicarious stimuli, i.e. the self and other stimuli. Participants are in the same room, and they can see and feel specific tactile stimuli that could be either congruent or incongruent in pleasantness, applied in a specific synchronous velocity of 3 cm/s (as done in ref. [37]; see also "Methods" section). This sensory matching is not possible in the original task[16], as participants are only presented with pictorial stimuli of 'what' touches the other person (not 'how' the other person is actually touched), while also seeing similar pictorial stimuli representing what is supposedly touching their body, as they are being touched

with different materials out of sight (rather than actual spiders, swans etc). In light of the aforementioned distinctions, our task constitutes a more embodied rendition of an egocentricity task and allows us to test specifically the existence of emotional EB and AB in a bodily context.

This adapted paradigm facilitated the computation of both classical emotional EB and AB (subsequently referred to as "classical bias", EB and AB), as well as sensory-controlled EB and AB (subsequently referred to as "sensory-controlled bias" sEB and sAB– see Fig. 1b, and "Methods" section for details). Classically, based on the study by Silani et al.[16], emotional egocentric and altercentric biases are computed as the difference in pleasantness between incongruent and congruent trials. A supplementary and beneficial approach to understanding these biases is to gauge the disparity between the pleasantness provided during incongruent trials (which involve multisensory and socially concurrent stimulation) and the pleasantness ratings given during unisensory trials (i.e. trials involving either seen or felt stimulation, devoid of concurrent social stimuli). To compute an emotional EB using unisensory baselines, we can subtract the pleasantness rating of the visual unisensory condition (vision prior). This subtraction illuminates the distinct and unique contribution of the self feeling to the EB bias. Similarly, to compute an AB grounded in unisensory baselines, we can subtract the pleasantness rating of the tactile unisensory condition (feel prior). This computation unveils the unique influence of observing another person's touch on the AB. Having these added, complementary control measures becomes especially valuable when testing pairs of participants with diverse roles (e.g. parent, child), age disparities and familiarity variation (e.g. own-child or. other unfamiliar-child). Such factors can potentially yield differences in unisensory, multisensory, and multiagent attention and perception.

To facilitate the interpretation of the biases, we can provide the following examples. Firstly, for the classical EB (controlled by congruency): consider a scenario where participant A is touched by cotton (pleasant), while simultaneously participant B is touched by a sponge (incongruent, unpleasant). If participant A rates the pleasantness of the sponge as more pleasant than when both participants are touched by a sponge, this will suggest that participant A is biased by the pleasant sensation from the felt cotton when evaluating the pleasantness of the sponge touching the participant B. This inclination will manifest as a positive classical egocentricity bias. For the sensory-controlled EB, analogous conclusions can be drawn. However, instead of comparing incongruent to congruent conditions, incongruent stimulation is compared to vicarious perception, wherein the self is not concurrently touched (the "vision only" condition). Thus, if during the 'incongruency' condition, a participant judges the pleasantness for the other person as more pronounced than in the "vision only" condition, this additional pleasantness (the discrepancy) can be attributed to what the participant projects to another person during incongruent, but concurrent multiagent stimulation. This differential in pleasantness does not arise from any inherent individual differences between seeing and feeling touch per se. It is important to note that the term "bias" bears different meanings in different fields of study. The current work places itself within the social mirroring field, wherein our focus is directed towards subjective sensory and emotional judgements (but see ref. [38] for the links between emotional EB and AB and relative "accuracy" in the perceptual metacognition field; as well as the bias model of judgement, which extends beyond the scope of our present design).

We tested these biases (congruency and sensory-controlled EB and AB) in dyads of strangers (Experiment 1) and of

mother-child pairs (Experiment 2) using the bodily, Emotional Egocentricity Task. In Experiment 1, we expected that, within our bodily Emotional Egocentricity Task, adults would exhibit both a larger classical but also a larger *sensory-controlled* EB, compared to AB. This would confirm and extend previous research conducted on dyads of adult strangers. In Experiment 2, we expected mothers to exhibit greater classical and sensory-controlled EBs compared to ABs. Additionally, we hypothesised that mothers would display greater EBs towards their own child in comparison to an unfamiliar child, given the importance of their own emotional perspective in understanding and regulating their children. It is noteworthy that our study is designed to address the aforementioned hypotheses regarding mothers' emotional EB and AB towards their children, an area that has received limited attention in the existing literature. In contrast, several studies have been dedicated to studying egocentricity in children. Therefore, we conducted exploratory analyses within the latter category, expecting the presence of greater biases in EB[19] and AB[39] in children. This is also expected to vary depending on the dyad composition (own mother vs. other mother) and the age of the participants[19].

## Methods

All three experiments were conducted in accordance with the Declaration of Helsinki (except for pre-registration: none of the studies were preregistered.) and were approved by the Ethics Committee of the Research Department of Clinical, Educational and Health Psychology, University College London. Written, informed consent was obtained from all participants prior to their participation. Mothers gave informed consent on behalf of their child, however, each child also gave verbally their own consent, to ensure they were comfortable with the experimental set-up. For all experiments, exclusion criteria included any self-reported present or past neurological and psychiatric disorders.

**Experiment 1a: The bodily Emotional Egocentricity Task – participants**. Forty-five participants (29 women, 16 men, $M_{age} = 34.72$, $SD_{age} = 10.68$, $Range_{age} = 20$–$65$ years; 37 right-handed, 4 left-handed and 2 ambidextrous; as reported by participants) volunteered during a public event at the Royal Institution, London to take part in the study.

**Experiment 1a - Experimental set-up**. Two custom-made boxes were designed specifically to allow a direct view of the other participant, while hiding the self. This configuration allowed the experimenter to simultaneously see and be able to touch both participants synchronously (dimensions: $40 \times 17 \times 20$ cm). The boxes were positioned in a mirrored manner, so that they were respectively open on the right and left sides, granting each participant the vision of the other participant's forearm exclusively. The upper part of the box was made opaque to block the view of their own forearm. Participants were seated next to each other, separated by a curtain to prevent any visual contact with each other. They were touched on their respective right and left forearms (Fig. 1c). The distance between the respective right and left index fingers of the two participants were kept at a constant distance of 30 cm (participants rested their index fingers on a tape affixed inside the box).

**Experiment 1a - Touch stimuli**. Touch was applied on a 9-cm-long segment on the participants' forearms. Participants' forearm was marked beforehand to ensure the touch would be consistently delivered to the same location for both participants. The tactile stimuli consisted of two distinct types of material (cotton ball and scourer). Touch was applied by a

trained experimenter at constant pressure and speed simultaneously on the two participants' forearms. Each touch lasted for 3 seconds and was repeated twice per trial. The choice of a speed of 3 cm/s was based on its established effectiveness in stimulating C-Tactile fibres[37]. The selection of fabrics was informed by the same pilot data used in a different study[37]. In this pilot phase, the pleasantness of 12 materials of different degrees of pleasantness was rated under visual-only, tactile-only, and visuo-tactile conditions by an additional group of 12 participants. Based on these data, we opted for the two most pleasant and two least pleasant fabrics, while ensuring similar visual attributes (i.e. size and colour). As such, our choices of fabrics perceived as pleasant or unpleasant were derived from pre-established stimuli, built on core hypotheses and piloting conducted prior to our primary experiments.

**Experiment 1a - Procedure.** Participants first filled questionnaires assessing empathy (Interindividual Reactivity Index; IRI[40]) and autism traits (short autism questionnaire 10; AQ10[41]), and then were paired with an unfamiliar, same gender, and similar age (when possible) participant (Fig. 1). Due to experimental administration error, the self-report questionnaires for three participants are missing. A curtain was used to separate the participants, to prevent any view of the 'other' (Fig. 1c). The experimenter explained the task and provided both participants with a booklet in which they would indicate their rating using a pen on a scale from 0 'not at all pleasant' to 10 'very pleasant'. The participant situated on the right of the experimenter was asked to place their right forearm inside the box in front of them; whereas the participant on the left was asked to place their left forearm in the box. Participants were unable to see their own forearms and were instructed to consistently look at the forearm of the 'other' participant, and focus on both the pleasantness of the touch they felt, and the pleasantness for the other of the touch they saw. Participants did not know before the end of the touch which of the two questions they would be asked to answer after the touch: either "How pleasant was the touch for YOU?" or "How pleasant was the touch for the OTHER?". Each pair of participants received simultaneous tactile stimulation using either identical materials (Congruent condition) or different materials (Incongruent condition) (Fig. 1a). The experimenter carried out two practice trials, followed by 2 blocks of 8 trials each, with a short break between the two blocks. All trials were pseudo-randomised.

**Experiment 1b – 'Replication' and extension of the bodily Emotional Egocentricity Task - Participants.** Thirty participants took part in Experiment 1b and were recruited via an online SONA system (18 women, 12 men, $M_{age} = 25.00$, $SD_{age} = 7.91$, $Range_{age} = 19$–56; years; $SD = 7.91$ years; all right-handed; as reported by participants). All participants were compensated for their time (either monetary or with course credits).

**Experiment 1b - Procedure.** Experiment 1b followed the same procedure as in Experiment 1a, but conducted in a laboratory-controlled environment, with a few modifications. Firstly, an additional pair of materials was introduced (synthetic wool as a pleasant material and velcro as an unpleasant material) to increase variability and statistical power, leading to a total 32 trials for the main task. The incongruent pairs were kept constant and always comprised materials with opposite valences (one pleasant and the other unpleasant). In Experiment 1b, participants provided all their ratings on a computer, using a continuous analogue scale from 0 to 100, 0 representing 'not at

all pleasant' and 100 representing 'extremely pleasant', requiring data entry via the trackpad for collection in MATLAB. Furthermore, two supplementary measures were included as unisensory priors (before the main task). In the Touch only condition participants were required to rate the pleasantness of the touch they felt (pleasantness rating of their own tactile experience), with no vision of self nor other. In the Vision only condition participants had to rate the pleasantness of the touch they saw on the other (conducted twice for each object and each condition).

**Experiment 2 - The bodily Emotional Egocentricity Task in Mother-Child dyads – Participants.** Hundred thirty-six participants took part in Experiment 2 during two public events ("FunFamily Day" at the Royal Institution, London; and "Self Impressions" held at the TATE Modern). Participants were recruited in pairs of mother and child (Mothers: $n = 68$, $M_{age} = 43.1$, $SD_{age} = 4.9$, $Range_{age} = 30$–57 years; Children: $n = 68$, 22 boys and 43 girls, $M_{age} = 8.3$, $SD_{age} = 2.3$, $Range_{age} = 5$–15 years), to be tested either separately or together, but always simultaneously by two experimenters. The pairs were informed of this, and written consent was obtained from mothers on behalf of themselves and their child prior to participation. Given developmental findings on children's ability to take another person's perspective not to develop before the age of seven and that other studies have used seven years old as a standard cut-off[12,19], children younger than seven were excluded from further analysis ($n = 16$). Additionally, a further 7 children did not complete the full task and were excluded from the analysis (final children sample: $n = 45$, 14 boys and 31 girls, $M_{age} = 9.24$, $SD_{age} = 2.1$, $Range_{age} = 7$–15 years, as reported by parents). See Supplementary Methods for sample size justification.

**Experiment 2 - Task and procedure.** Experiment 2 followed the same design and procedure as Experiment 1b, with three exceptions: (i) Experiment 2 was conducted using a between-subject design, where mothers and children were paired either with their own child ($n = 35$) /mother ($n = 22$) or with another child ($n = 33$) / mother ($n = 23$) (Fig. 1d). (ii) For the mothers, the pleasantness scale was presented on a computer (0 to 100 VAS scale), as in Experiment 1b. However, children were asked to point to the relevant number in front of them on a paper-based scale, which was illustrated with cartoon faces to denote negative (unhappy) and positive (happy) affect, ranging from 0 = not at all pleasant/unhappy to 10 = extremely pleasant/happy. This 0–10 was used in order to facilitate ratings in children. (iii) Similar to Experiment 1b, two additional baseline measures were also included, the Touch only condition and the Vision only condition. As in Experiment 1, mothers answered demographics questions as well as the IRI[40]. Children answered the equivalent questionnaire developed for children (EmQue-CA[42]) measuring both affective and cognitive empathy as well as prosocial motivation. Due to technical error, only 28 children answered this questionnaire. Each experiment lasted for about 20 min.

**Statistical analysis.** We first computed the classical biases. Similarly to previous studies[13,16], classical EB was computed as the average of the difference between incongruent and congruent trials when participants had to judge the pleasantness of the touch for the other participant and classical AB was computed similarly but in trials when participants had to judge the pleasantness of the touch they felt.

$$\text{classical EB} = \frac{(\text{Incongruent Other Unpleasant} - \text{Congruent Other Unpleasant}) + (-1^*(\text{Incongruent Other Pleasant} - \text{Congruent Other Pleasant}))}{2}$$

$$\text{classical AB} = \frac{(\text{Incongruent Self Unpleasant} - \text{Congruent Self Unpleasant}) + (-1^*(\text{Incongruent Self Pleasant} - \text{Congruent Self Pleasant}))}{2}$$

Note that the valence label corresponds to what the target was feeling, e.g. 'Incongruent Other Unpleasant' corresponds to a condition where the 'self' was feeling a pleasant touch while seeing the 'other' being touched with an unpleasant material.

Next, in Experiment 1b and Experiment 2, using the unisensory priors measures, we computed a measure of sensory-controlled biases: sensory-controlled EB (sEB) and sensory-controlled AB (sAB). These were averaged across valences (Pleasant and Unpleasant), similarly as the classical biases. Each sensory-controlled bias was computed as follows:

$$\text{sEB} = \frac{(\text{Incongruent Other Unpleasant} - \text{Vision Only Unpleasant}) + (-1^*(\text{Incongruent Other Pleasant} - \text{Vision Only Pleasant}))}{2}$$

$$\text{sAB} = \frac{(\text{Incongruent Self Unpleasant} - \text{Tactile Only Unpleasant}) + (-1^*(\text{Incongruent Self Pleasant} - \text{Tactile Only Pleasant}))}{2}$$

Descriptive statistics of each variable are displayed in SM (Supplementary Tables 1, 3, 5 and 7), as well as correlations matrices between the different biases (Supplementary Tables 2, 4, 6 and 8).

To confirm the existence of these emotional biases, in line with previous studies[13,16], we used one sample $t$-tests to test whether ABs and EBs were significantly different from zero. To examine the hypothesised disparities between biases (AB vs EB), a multilevel modelling (MLM; REML) approach was employed using R Studio (RStudio Team, 2022; package *lme4*). The bias score (as computed from AB and EB scores) was employed as the continuous dependent variable. Bias type (Altercentric vs. Egocentric) was included as a binary categorical fixed factor (coded with $AB = 0$ and $EB = 1$), while participant age in years was entered as a continuous variable. Participants were treated as a random effect in the models to account for individual variability. Notably, age was integrated into the models due to findings from previous research indicating age-related variations in emotional egocentric and altercentric biases. For Experiment 2, the statistical analysis closely paralleled that of Experiment 1, with the exception of the use of type of pairs (Own vs. Other), that was introduced as an additional fixed binary factor (coded with $Own = 0$ and $Other = 1$). Additional details and full model results tables can be found in SM (Supplementary Tables 9 to 12).

When applicable, Bayesian statistics were performed using JASP (JASP Team (2022). JASP Version 0.16.3) to approximate the Bayes Factors ($BF_{10}$), using the default prior (Cauchy of 0.707), to allow further interpretation of the observed effects, in particular, the extent to which data provided support for the alternative versus null hypotheses. We employed these factors to interpret the evidence for each hypothesis, adhering to the benchmarks outlined by Jeffreys (1961). A $BF_{10}$ above 10 was considered as strong evidence supporting the alternative hypothesis, a $BF_{10}$ between 3 and 10 was considered as moderate, substantial evidence, while a $BF_{10}$ as below 0.3 was deemed as substantial evidence favouring the null hypothesis. For values between 1 and 3, the evidence was considered as anecdotal for the alternative hypothesis; and values between 0.3 and 1 as anecdotal evidence for the null hypothesis[43,44].

To explore the relationships between the biases (AB, EB) and social cognition measures (i.e., AQ10, and IRI) we combined the data from Experiment 1a and Experiment 1b and run an exploratory Spearman partial correlation analysis, controlling for age and experiment (i.e., Experiment 1a or Experiment 1b) – see the results in Supplementary Materials. For Experiment 2 we also explored the relationships of emotional biases with social cognition measures (as measured by the IRI in mothers and EmQu-CA in children). Correlations were corrected for false discovery rate (FDR) correction using the 'BH' method[45].

*Outliers detection.* For all experiments, we employed two methods to identify outliers. First, we examined participants' average response to the 'Tactile only' baseline unisensory pleasant condition ("Tactile only pleasant"') compared to the 'Tactile only' baseline unisensory unpleasant condition ("Tactile only unpleasant"). Individuals who rated the unpleasant stimuli as more pleasant than the pleasant stimuli (e.g., rating the touch of a scourer higher on average than that of a cotton ball), were highlighted as potential outliers. This examination identified 6 potential outliers only within the mother groups in Experiment 2. No outliers were identified in Experiment 1b nor in the children group in Experiment 2. In Experiment 1a, where we lacked a unisensory baseline condition, we compared the 'self pleasant congruent' condition to the 'self unpleasant congruent' condition. This comparison yielded one potential outlier. Subsequently, we conducted our analyses both with and without these potential outliers. However, the results remained consistent, leading us to retain them in the sample to account for possible individual differences (see SM for results without outliers; Supplementary Tables 13 and 15).

Second, we also examined potential outliers, based on criteria involving score above or below 2.5 SD from the group's average ratings scores across the different conditions. These analyses revealed one potential outlier in Experiment 1b, three in the mother group, and two in the children group in Experiment 2. Once again, our analyses were performed with and without these outliers, and as the observed effects remained unchanged, we decided to retain them in the sample (see SM for results without outliers; Supplementary Tables 14, 16 and 17).

**Reporting summary**. Further information on research design is available in the Nature Portfolio Reporting Summary linked to this article.

## Results

**Experiment 1: the bodily Emotional Egocentricity Task in dyads of strangers**. In a first sample (Experiment 1a), we showed that neurotypical adults exhibited significant classical emotional

EB and AB in our bodily Emotional Egocentricity Task (EB; $M_{EB} = 9.44$, $SD_{EB} = 12.89$, $t_{(44)} = 4.91$, $p < 0.001$, Cohen's $d = 0.73$, 95% CI [5.57; 13.32], $BF_{10} = 1564.5$; AB: $M_{AB} = 5.00$, $SD_{AB} = 10.94$, ; $t_{(44)} = 3.07$, $p = 0.004$, Cohen's $d = 0.46$, 95% CI [1.71, 8.29], $BF_{10} = 9.27$; see Fig. 2a and Supplementary Table 9). Moreover, to test our hypothesis regarding the difference between the biases (classical AB and EB) we employed multilevel models (MLM). In these models, the bias score (computed from the AB and EB scores, see "Methods" section) served as a continuous dependent variable, bias type (AB vs. EB) and age were used as fixed factors and participants as a random effect. A main effect of bias type (EB vs. AB) emerged ($b = 4.44$, SE $= 1.82$, $p = 0.017$; 95% CI [0.82, 8.07], see Fig. 2a and Supplementary Table 9), indicating a larger EB as compared to AB, while controlling for age.

In a second sample of neurotypical adults, we replicated partially these findings, with a statistically significant EB (finding a $BF_{10}$ of 3.13, indicating moderate support for H1 hypothesis), but no credible evidence for a significant AB (finding a $BF_{10}$ of 0.21 indicating a moderate support for H0 hypothesis; EB: $M_{EB} = 7.18$, $SD_{EB} = 15.28$, $t_{(29)} = 2.57$, $p = 0.015$, Cohen's $d = 0.47$, 95% CI [1.48, 12.89], $BF_{10} = 3.13$; AB: $M_{AB} = -0.37$,

$SD_{AB} = 6.06$, $t_{(29)} = -0.33$, $p = 0.742$, Cohen's $d = -0.06$, 95% CI [−2.63, 1.90], $BF_{10} = 0.21$). We also found a significant difference between EB and AB, while controlling for age ($b = 7.55$, SE $= 2.45$, $p = 0.004$, 95% CI [2.62, 12.48], see Fig. 2b and Supplementary Table 10a). The difference observed between the two samples might be due to the impact of interindividual differences (i.e., difference on average age of each sample and scores on the empathy questionnaire IRI, with participants in Exp1a being on average older and scoring higher on the IRI than participants in Exp1b, see SM).

In this second sample, using the measure of unisensory priors (see Methods), we also showed that the sensory-controlled EB and sAB were significantly different from zero (with both $BF_{10}$ indicating a strong support for H1 hypothesis), confirming the existence of these sensory-controlled biases (sEB: $Ms_{EB} = 11.19$, $SDs_{EB} = 13.42$, $t_{(29)} = 4.57$, $p < 0.001$, Cohen's $d = 0.86$, 95% CI [6.18, 16.20], $BF_{10} = 114.622$; and sAB: $Ms_{AB} = -7.42$, $SDs_{AB} = 9.73$, $t_{(29)} = -4.17$, $p < 0.001$, Cohen's $d = -0.76$, 95% CI [−11.06, −3.79], $BF_{10} = 302.63$; see Fig. 2c). Moreover, using MLM with sensory-controlled bias scores as continuous dependent variables, bias type (sAB vs. sEB) as binary and age continuous fixed factors and participants as a random effect, a

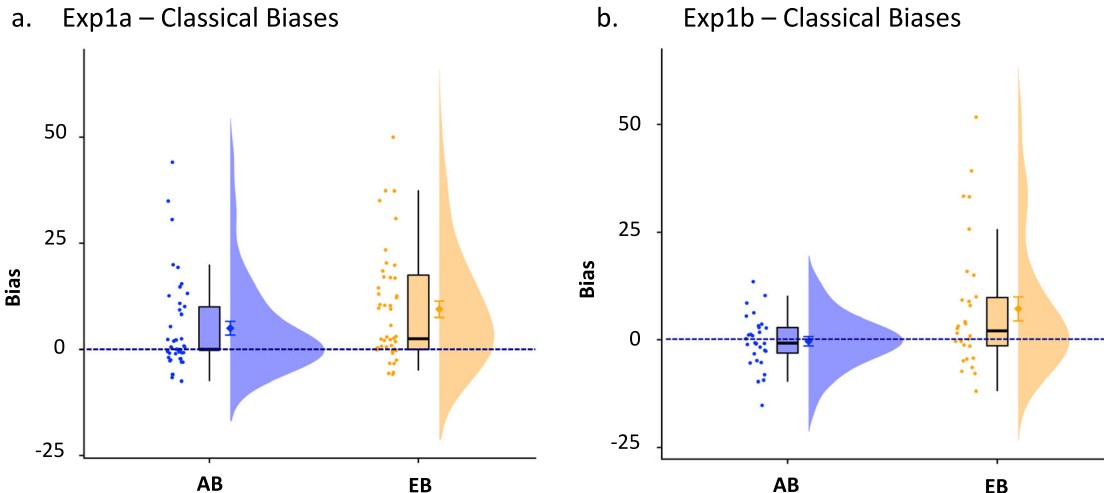

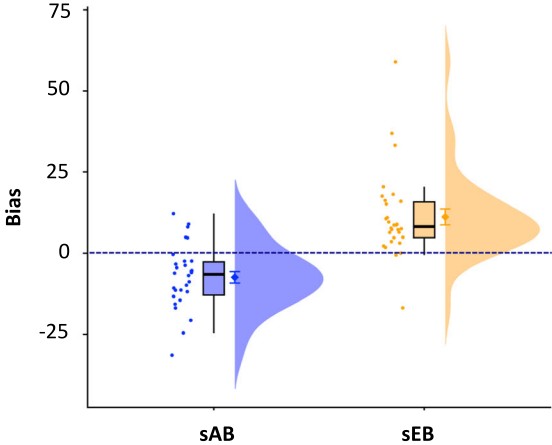

**Fig. 2 Egocentric and altercentric biases (EB/AB) in Experiment 1. a** Classical biases found in Experiment 1a. ($N = 45$). **b** Classical biases found in Experiment 1b ($N = 30$). **c** Sensory-controlled biases found in Experiment 1b. ($N = 30$). EB classical congruency-controlled egocentric bias, AB classical congruency-controlled altercentric bias, sEB sensory-controlled egocentric bias, sAB sensory-controlled altercentric bias. Jittered dots represent biases for each individual participant. Boxplot presents the central tendency. The half violin provides data distribution. Diamond represents the mean for each bias, and the error bars denote the standard error of the mean.

main effect of bias type (sEB vs. sAB) emerged ($b = 18.61$, SE = 2.38, $p < 0.001$, 95% CI [13.82, 23.39]; see Fig. 2c and Supplementary Table 10b), indicating a larger sEB as compared sAB, while controlling for age.

**Experiment 2: The bodily Emotional Egocentricity Task in Mother-Child dyads - Classical emotional EB and AB in Mothers**. First, similar to Experiment 1, we ran one sample $t$-tests and showed that both EB and AB were significantly different from zero, with $BF_{10}$ indicating strong evidence for EB and anecdotal evidence for AB (EB: $M_{EB} = 10.73$, $SD_{EB} = 18.30$, $t_{(67)} = 4.84$, $p = {<}0.001$, Cohen's $d = 0.59$, 95% CI [6.31, 15.16], $BF_{10} = 2174.14$; AB: $M_{AB} = 3.14$, $SD_{AB} = 10.11$, $t_{(67)} = 2.56$, $p = 0.013$, Cohen's $d = 0.31$, 95% CI [0.70, 5.59], $BF_{10} = 2.74$). To test our hypothesis regarding the impact of close relationships such as mother-child ones would have on the biases (AB and EB), 68 mother-child dyads were recruited, half of the mothers being

paired with their own and half with someone else's child (see Methods). Bias score (as computed from AB and EB score) was served as a continuous dependent variable in a multilevel model, with bias type (AB vs. EB), mother's age, child's age and pairs (Own vs. Other) as fixed factors and participants as a random effect. Similar to Experiment 1, we found a main effect of bias type (EB vs. AB) ($b = 13.17$, SE = 3.23, $p < 0.001$; 95%CI [6.78; 19.56]), indicating a larger classical EB as compared to classical AB. Crucially, and confirming our hypothesis, a two-way interaction between bias type (EB vs AB) and being paired with own vs. other child emerged ($b = -11.50$, SE = 4.63, $p = 0.014$, 95% $CI[-20.67; -2.33]$; see Fig. 3a, and full results in Supplementary Table 11a). Probing this interaction using planned comparisons, revealed that this effect was driven from a larger classical EB in the group that was paired with their own child as compared to the group that was paired with a stranger (i.e., other child), as confirmed by Bayes Factor (with $BF_{10}$ of 10.73 indicating a strong support for the alternative hypothesis) ($t_{(66)} = 3.34$, $p = 0.001$,

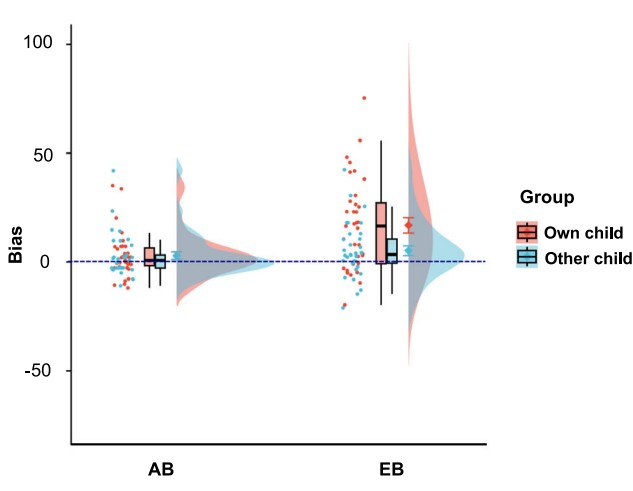

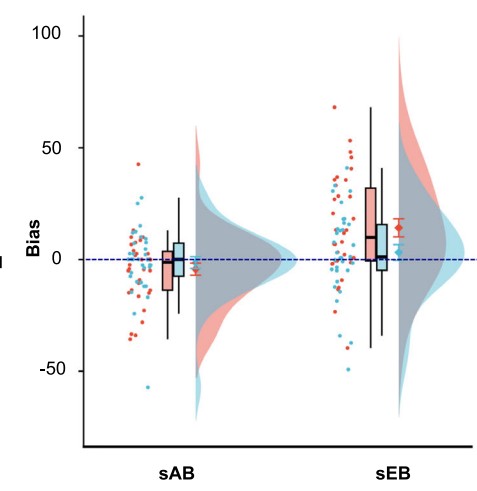

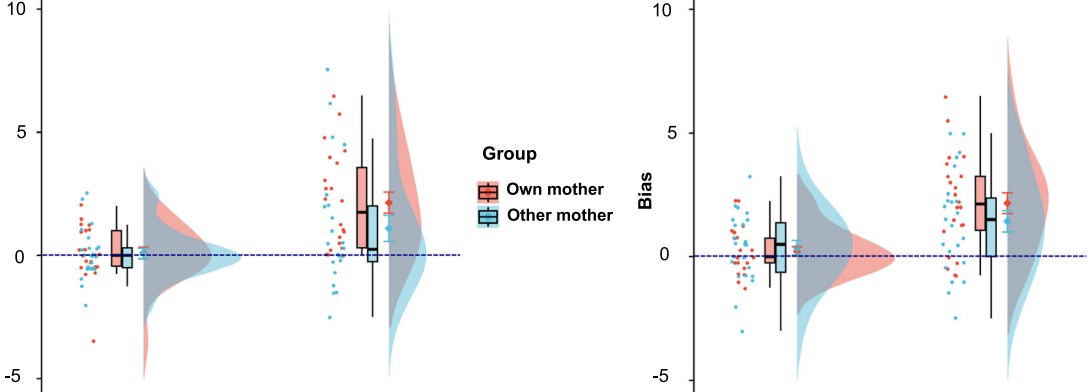

**Fig. 3 Emotional biases in mothers and children. a** Classical congruency-controlled Egocentric (EB) and Altercentric Biases (AB) in Experiment 2 in mothers paired with their own ($N = 35$) or other child ($N = 33$). **b** Sensory-controlled Egocentric (sEB) and Altercentric Biases (sAB) in Experiment 2 in mothers paired with their own ($N = 34$) or other child ($N = 33$). **c** Classical congruency-controlled Egocentric (cEB) and Altercentric Biases (cAB) in Experiment 2 in children paired with their own ($N = 22$) or other mother ($N = 23$). **d** Sensory-controlled Egocentric (sEB) and Altercentric Biases (sAB) in Experiment 2 in children paired with their ($N = 22$) or other mother ($N = 23$). Jittered dots represent biases for each individual participant. Boxplot presents the central tendency. The half violin provides data distribution. Diamonds represent the mean for each bias, and the error bars denote the standard error of the mean.

$95\%CI[4.83;18.81]$; $M_{EB\_own} = 16.36$, $SE = 3.54$; $M_{EB\_Other} = 4.77$, $SE = 3.53$; $BF_{10} = 10.73$]. There was no significant difference in classical AB between the two groups, with $BF_{10}$ of 0.19, indicating a moderate support for the null hypothesis ($t_{(66)} = 0.09$, $p = 0.92$, $95\%CI[-6.66;7.31]$; $M_{AB\_own} = 3.32$, $SE = 3.53$; $M_{AB\_Other} = 3.00$, $SE = 2.55$; $BF_{10} = 0.19$). In addition, the difference between classical EB and AB was significant only within the group that was paired with their own child, as confirmed by Bayes Factors, with a $BF_{10}$ of 26.07 for the group that paired with their own child, indicating a strong support for the H1 hypothesis, and a $BF_{10}$ of 0.22 in the other group, indicating a moderate support for the null hypothesis (own child: $t_{(66)} = 4.08$, $p < 0.001$, 95% CI[6.72;19.61], $BF_{10} = 26.07$; other child $t_{(66)} = 0.50$, $p = 0.62$, $95\%CI[-4.92;8.31]$, $BF_{10} = 0.22$).

**Experiment 2 - Sensory-controlled EB and AB effects in Mothers.** First, similar to Experiment 1, we ran one sample $t$-tests and showed that only sensory-controlled EB was significantly different from zero (with a $BF_{10}$ of 7.23, indicating a moderate support for the H1 hypothesis) but not credible evidence was found for sensory-controlled AB, with a $BF_{10}$ of 0.42 indicating anecdotal support for the null hypothesis (sEB: $M_{sEB} = 8.09$, $SD_{sEB} = 22.30$, $t_{(66)} = 2.97$, $p = 0.004$, Cohen's $d = 0.36$, 95% CI [2.65, 13.53], $BF_{10} = 7.23$; sAB: $M_{sAB} = -2.91$, $SD_{sAB} = 15.29$, $t_{(66)} = -1.56$, $p = 0.12$, Cohen's $d = -0.19$, 95% CI [-6.639, 0.819], $BF_{10} = 0.42$) (Fig. 3b). Second, MLM was used in order to test our hypothesis regarding the impact of the dyadic pair (Own vs. Other) on the sensory-controlled biases. Bias scores were served as a continuous dependent variable in a MLM, with bias type (sAB vs. sEB), mother's age, child's age and pairs (Own vs. Other) as fixed factors and participants as a random effect. Similar to Experiment 1, we found a significant main effect of sensory-controlled bias type ($b = 17.84$, $SE = 3.51$, $p < 0.001$; $95\%CI[10.96;24.72]$). Crucially, a two-way interaction between the sensory-controlled bias type (sEB vs sAB) and being paired with own vs. other child emerged ($b = -13.88$, $SE = 4.96$, $p = 0.006$, $95\%CI[-23.71;-4.06]$; see Fig. 3b and full results in Supplementary Table 11b). Probing this interaction using planned comparisons, revealed that this effect was driven from a larger sEB in the group that was paired with their own child as compared to the group that was paired with a stranger (i.e., other child) ($t_{(65)} = 2.30$, $p = 0.02$, $95\%CI[1.51;20.30]$; $M_{sEB\_own} = 13.42$, $SE = 4.00$; $M_{sEB\_Other} = 2.51$, $SE = 3.48$; $BF_{10} = 1.49$), with a $BF_{10}$ of 1.49 indicating anecdotal support for the H1 hypothesis. Similar to the classical AB, there was no credible evidence for a difference in sensory-controlled AB between the two groups ($t_{(65)} = -0.69$, $p = 0.49$, $95\%CI[-12.88;6.21]$; $M_{sAB\_own} = -4.42$, $SE = 3.32$; $M_{sAB\_Other} = -1.09$, $SE = 3.42$; $BF_{10} = 0.32$), with a $BF_{10}$ of 0.32 indicating moderate support for the null hypothesis. In addition, the difference between sensory-controlled EB and sensory-controlled AB was significant only in the group that was paired with their own child (own child: $t_{(65)} = -5.12$, $p < 0.001$, $95\%CI[-24.79;-10.88]$, $BF_{10} = 930.36$; other child: $t_{(65)} = -1.11$, $p = 0.26$, $95\%CI[-11.01;3.10]$, $BF_{10} = 0.34$), with a $BF_{10}$ of 930.36 indicating strong support for the H1 hypothesis in the group that was paired with their own child; and a $BF_{10}$ of 0.34 indicating anecdotal support for the null hypothesis in the other group.

**Experiment 2 - Exploratory analysis of emotional biases in Children - Classical emotional EB and AB in Children.** One sample $t$-tests showed that classical emotional EB was significantly different from zero, as confirmed by a $BF_{10}$ of 672.28 indicating strong support for the alternative hypothesis; but no credible evidence for classical AB, with a $BF_{10}$ of 0.19 indicating moderate support for the null hypothesis (EB: $M_{EB} = 1.60$,

$SD_{EB} = 2.32$, $t_{(44)} = 4.63$, $p = <0.001$, Cohens' $d = 0.69$, 95% CI [0.906, 2.3], $BF_{10} = 672.28$; AB: $M_{AB} = 0.09$, $SD_{AB} = 1.056$, $t_{(44)} = 0.58$, $p = 0.56$, Cohens' $d = 0.08$, 95% CI [-2.26, 4.09], $BF_{10} = 0.19$). As an exploratory analysis, we ran a similar MLM in children, with bias score (as computed from AB and EB score) as a continuous dependent variable, and bias type (AB vs. EB), child's age and pairs (Own vs. Other) as fixed factors and participants as a random effect. A main effect of bias type was found ($b = 2.03$, $SE = 0.54$, $p < 0.001$, $95\%CI[0.97;3.10]$) indicating evidence for a larger classical EB relative to classical AB. However, unlike the mother group, we did not find credible evidence for a two-way interaction between bias type and being paired with one's own vs other mother ($b = -1.02$, $SE = 0.75$, $p = 0.18$, 95% CI[-2.52;0.47]; see Fig. 3c and full results in Supplementary Table 12a).

**Experiment 2 - Exploratory analysis of emotional biases in Children - Sensory-controlled EB and AB in Children.** One sample $t$-tests showed that only sensory-controlled EB was significantly different from zero, with a $BF_{10}$ of $3.39e + 4$ indicating strong support for the H1 hypothesis; while a $BF_{10}$ of 0.34 for sensory-controlled AB indicated anecdotal support for the null hypothesis (sEB: $Ms_{EB} = 1.79$, $SDs_{EB} = 2.03$, $t_{(44)} = 5.90$, $p = <0.001$, Cohens' $d = 0.88$, 95% CI [1.176, 2.396], $BF_{10} = 3.39 e + 4$; sAB: $Ms_{AB} = 0.29$, $SDs_{AB} = 1.19$, $t_{(44)} = 1.64$, $p = 0.11$, Cohens' $d = 0.64$, 95% CI [-0.066, 0.649], $BF_{10} = 0.56$). In addition, we ran a similar MLM in children, with the sensory-controlled bias score as a continuous dependent variable, and bias type (sAB vs. sEB), child's age and pairs (Own vs. Other) as fixed factors and participants as a random effect. A significant main effect of bias type was found ($b = 1.94$, $SE = 0.50$, $p < 0.001$, $95\%CI[0.95;2.94]$; see Fig. 3d and full results in Supplementary Table 12b) indicating evidence for a larger sensory-controlled EB relative to sensory-controlled AB. Again, unlike the mother group, we did not find credible evidence for a two-way interaction between bias type and being paired with one's own vs other's mother ($b = -0.88$, $SE = 0.70$, $p = 0.21$, $95\%CI[-2.27;0.51]$; see Fig. 3d).

**Experiment 2 - Correlations with Social Cognition Measures.** To explore the relationships between the classical and the sensory-controlled biases and social cognition measures (i.e., IRI in mothers and EmQu-CA in children) we run an exploratory Spearman partial correlation analysis, controlling for age and using FDR corrections to control for multiple comparisons (reported $p$ values below are before FDR corrections).

In the mother group, partial correlation analysis yielded positive correlation between IRI empathic concern subscale and both the classical EB ($r_{(63)} = 0.33$, $p = 0.01$, $95\%CI[0.07;0.53]$) and sensory-controlled EB ($r_{(62)} = 0.41$, $p = 0.001$, 95% CI[0.12;0.60]). In addition, negative correlation between the sensory-controlled EB and personal distress subscale (i.e., self-oriented feelings of distress in interpersonal settings) was found ($r_{(62)} = -0.37$, $p = 0.003$, $95\%CI[-0.09;-0.62]$).). All reported effects survived FDR corrections and no other significant effects emerged. To further explore these correlations we explored the two groups separately (those who paired with their own child and those who were paired with a stranger), and found that the positive correlation with IRI empathic concern is only significant in the 'own group' (classical EB: $r_{(32)} = 0.43$, $p = 0.02$, 95% CI[0.06;0.70] and sensory-controlled EB: $r_{(31)} = 0.58$, $p < 0.001$, 95%CI[0.24;0.81]), with a trend in the "other" group (classical EB: $r_{(31)} = 0.32$, $p = 0.09$, $95\%CI[-0.09;0.70]$ and sensory-controlled EB: $r_{(30)} = 0.33$, $p = 0.08$, $95\%CI[-0.06;0.65]$)). Similarly, the negative correlation between sensory-controlled EB and personal distress was only significant in the group that was paired with

their own child ($r_{(31)} = -0.43$, $p = 0.02$, 95%CI[−0.06;0.71])), and in the other group showing only a trend ($r_{(30)} = -0.35$, $p = 0.06$, 95%CI[−0.73;0.12]).

In the children group, partial correlation analysis yielded positive correlations between the classical emotional AB and affective empathy ($r_{(20)} = 0.74$, $p < 0.001$, 95%CI[0.43;0.89]), suggesting that those who show higher AB scored higher on the affective empathy subscale and report more prosocial motivation ($r_{(20)} = 0.63$, $p = 0.004$, 95%CI[0.28;0.85]). All reported effects survived FDR corrections and no other significant effects emerged.

## Discussion

The ability to understand other minds relies on our own concurrent mental states and vice versa. Thus, to enable successful social and self-understanding beyond aligned experiences, we must be able to distinguish between our own and another's mental and affective states. A better understanding of the self-other distinction in the context of key relational dyads, such as parent-child dyads is crucial, given that in everyday social interactions with close others our social understanding relies on both current and past relational experiences, representations and goals[7,22,24,46,47].

In the present study, given the key role of bodily feelings within parent-child dyads, as well as the potential sensory confounds observed in previous self-other distinction tasks within such dyads, we first successfully employed a bodily Emotional Egocentricity Task. This was utilised across two separate experiments involving healthy adult volunteers. Our findings not only confirmed the expected bodily classical EB observed in previous studies, but also allowed us to calculate a bodily sensory-controlled EB in concurrently tested dyads. By employing the bodily Emotional Egocentricity Task, we uncovered evidence that supports our central hypothesis - specifically, that mothers exhibit a more pronounced emotional EB towards their own child in comparison to an unfamiliar child. Importantly, this observation held true for both classical EB and sensory-controlled EB measurements. This suggests that in situations where mothers witness their own child undergoing tactile sensations different from their own, they tend to project their own sensations onto their child. This bias cannot be explained by any relative differences in valence between self and the child during simultaneous stimulation (original bias calculations). Nor can it be explained by any biases in how the mothers perceive such stimuli on their own bodies while experiencing them in isolation (novel bias calculations employing unisensory controls).

According to the social neuroscience literature (e.g. refs. [12,48]), self-other distinction is seen as crucial to adapt to the complex social environment, a greater maternal EB towards their own child could be interpreted as a 'failure' in distinguishing between the self and the other. This could possibly indicate that mothers are less attuned to their child's states compared to their own. However, according to existing literature in social and developmental psychology, this interpretation might not be applicable to well-established intimate relationships, where egocentricity could offer certain relational advantages. For instance, the tendency to project one's self and idealised relationship perception onto judgments of their partners has been demonstrated to be associated with heightened empathic understanding, increased feelings of closeness, and greater satisfaction among married couples (see[42] for a review on embodied self-other overlap in romantic love, but also ref. [35]). This interpretation goes in line with our findings that within our sample, a higher EB in the mother group was associated with feelings of sympathy and concern for unfortunate others, as indicated by a self-report measure; and this was true particularly among the group paired with their own child.

These results are also consistent with a homeostatic regulation perspective, according to which the observed, child-specific, maternal EB could be interpreted as an indicator of an adaptive necessity for anticipatory bodily and affective regulation (as outlined in the Introduction). Indeed, parenting requires not only attending and responding to their children 'here and now' embodied and emotional needs (homeostastic regulation[22]), but also foreseeing future hypothetical alterations[26,29,30], that can be inferred by interpreting one's own embodied incongruous states. From a homeostatic regulation perspective, the distinction between self and other via day-to-day embodied parent-child interactions such as feeding, hugging, bathing and so on, can play a unique role in interoceptive inference (see[26]). For instance, parents might employ their own body temperature to determine the most suitable attire for their child to wear to school. Alternatively, a parent might anticipate that a child will be willing to explore new tastes and partake in a special family meal due to the delight and sense of curiosity that the parents themselves experience within the given context.

Furthermore, our results are also in line with the "social referencing theory"[49], according to which the greater egocentricity bias of mothers towards their own child could also be related to the parents' social role as a reference point for guiding the child's response. Children often seek their caregiver's responses (e.g., facial cues) to guide their own responses, with caregivers acting as significant figures from whom children gather informational cues. In the context of our findings, this larger egocentricity bias could signify that parents focus more on their own affective responses to communicate and guide their child, rather than solely attending to their child's reaction. This is also consistent with studies suggesting that the parent's role as source of information is intertwined with their adaptive or pedagogic epistemic role within a dynamic environment[28]. Consequently, the greater egocentricity bias within the 'own group' may highlight the parent's function as a reliable source of information transmitted to their child, thus facilitating social learning.

Alternatively, our results could pertain to the distinction between familiar situations (i.e., being paired with their own child) and unfamiliar or novel situations (i.e., being paired with an unfamiliar child). In familiar instances, the parent is accustomed to the scenario of gauging or guessing their own feelings, potentially promoting the utilisation of egocentric information, which in turn results in a larger egocentricity bias. In contrast, in unfamiliar conditions, participants might engage in more deliberate, controlled thinking about the child's perspective, leading to reduced reliance on 'perspective shortcuts' and a diminished egocentric bias (though see ref. [50] for the application of a heuristic mode in uncertain situations).

Future studies should try to disentangle between these different, yet interconnected interpretations. For instance, one could test the possible interrelation between epistemic trust and attachment with egocentricity bias, as in order to enable social learning, one needs to establish trust[51]. It could be that children with higher epistemic trust and greater parental referencing will show larger allocentric bias, as they are more 'open' to and focus towards receiving information from the other. One could also assess the interrelation between individuals' tolerance for uncertainty[52], parental reflective function[53] and, altercentric and egocentric biases.

It is crucial to acknowledge the presence of interindividual variability observed in both the EB and AB scores. While, on average, mothers exhibited a significant and large EB, suggesting a heightened influence of their own feelings when evaluating their own child, this does not hold true for every mother. As illustrated in Fig. 3, it becomes evident that some mothers even displayed a negative EB, suggesting that they were not influenced by their own feelings when their child

was experiencing an incongruent sensation. These differences could potentially arise from individual factors such as heightened concerns for others, as suggested by the present study, or attachment styles that warrant further investigation in future research. Noteworthy, in the current paper, we adhere to the established research tradition that explores "emotional EBs" and generally employs the term "emotion" to refer to the evaluation of self-other affective states through pleasantness ratings (e.g.[16]). Accordingly, we employ the terms "bodily" and "emotional feelings" as designating the pleasantness arising from tactile stimulation, as measured by the behavioural ratings of "how pleasant was the touch". Given this specific definition, the inferences that can be drawn from the present study are confined to judgments of tactile pleasantness, and do not necessarily encompass more intricate emotional states such as anger and happiness.

In relation to our exploratory analyses in the children's sample, similar to previous findings, we identified larger emotional EBs compared to ABs in children, which were not linked to the group-pairing condition. However, contrary to our hypothesis, we did not find conclusive evidence that children exhibit a greater AB towards their own mother than an unfamiliar mother. This could be due to the heightened egocentricity of mothers towards their own child, or possibly the relatively wide age range within our sample. These findings suggest that children's egocentricity might represent a generic, non-parent oriented mechanism, and thus potentially not influenced by relational closeness. Subsequent studies could assess this hypothesis by focusing on different more homogeneous age groups.

## Limitations
Limitations of this study should be acknowledged. Firstly, although we successfully conducted the bodily Emotional Ego-centricity Task across three different experiments, our primary investigation involving mother-child dyads was tested only once. This calls for replication in future studies, and the generalisation of findings to father-child dyads should also be explored. Secondly, the mother-child experiment was conducted in a public event, which restricted our ability to include additional measures such as socioeconomic or educational status, attachment style, or other developmental traits or state measures that have previously demonstrated an impact on self-other distinction bias[13,54]. These factors could potentially interact with or account for the observed effects. Future studies should examine this possibility in a controlled laboratory setting. Thirdly, in our design, we compared being paired with one's "own" child vs an unfamiliar child, which did not directly control for the influence of familiarity. However, we acknowledge that the role of familiarity in this study is not solely a 'confounding' factor; it is also a hypothesised main effect. Specifically, we do believe that parents might exhibit distinct biases towards their own child due to familiarity with them, and with their habitual needs and their need for parental regulation. While there are other dimensions of familiarity that can act as confounds, such as perceptual or attentional familiarity, ease during testing, and lower-level perceptual factors, these could be partially controlled by testing other familiar children in future studies. Nevertheless, we note that this approach could introduce new confounds, given that family and friendship roles and levels of familiarity are not identical beyond superficial levels. Importantly, our unisensory baselines do control for some of these lower-level perceptual familiarity aspects, as they enable the calculation of biases based on unisensory measures of both seen and felt touch, unique to each tested child. Lastly, to date, no study has investigated the impact of time, and the stability of emotional biases over time. Future studies could be designed to address such questions, which our current design, involving only two to four repetitions per condition, did not permit.

## Conclusions
In conclusion, the present study provides insights into the nature of the self-other distinction within mother-child relationships. By employing the bodily Emotional Egocentricity Task, which was designed to assess self-other distinction biases while effectively accounting for sensory, salience and familiarity confounds, our findings demonstrate that mothers exhibit a higher degree of emotional egocentricity towards their own child compared to another child. As discussed, this phenomenon could be attributed to the demands associated with parental affective regulation and the underlying relational and teaching goals aimed at facilitating social learning. This study paves the way for new investigations into the effects of context on bodily emotional biases, as well as into parental behaviours.

## Data availability
De-identified data for all experiments are available on the Open Science Framework (https://osf.io/xwvhg; https://doi.org/10.17605/OSF.IO/XWVHG).

## Code availability
All relevant analysis scripts in R (RStudio Team, 2022) are available on the Open Science Framework and on request to the authors (https://osf.io/xwvhg; https://doi.org/10.17605/OSF.IO/XWVHG).

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

## Acknowledgements

Thank you to all KatLab members who helped with this study, especially during the public events, to name a few, Laura Crucianelli, Sonia Ponzo, Emma Johnsson and Elena Panagiotopoulou. We are grateful to all the participants and families who took part in the studies. This project was supported by the European Union's Horizon 2020 research and innovation programme under grant agreement No.818070, for the Consolidator Award METABODY (to A.F.).

## Author contributions

L.P.K.: designed the study, conducted testing and data collection, analysed the data, wrote the manuscript, revised and approved the final version of the manuscript. M.T.: analysed the data, wrote the manuscript, revised and approved the final version of the manuscript. M.-L.F.: conducted testing and data collection, revised and approved the final version of the manuscript. M.v.M.: conducted testing and data collection, revised and approved the final version of the manuscript. A.F.: designed the study, conducted testing and data collection, wrote the manuscript, revised and approved the final version of the manuscript.

## Competing interests

The authors declare no competing interests.
