## [Peer Review File · Communications Psychology]

1st Feb 23

Dear Dr Kirsch,

Thank you for your patience during the peer-review process. Your manuscript titled "Mother knows best: Mothers are more egocentric towards their own child's bodily emotions" has now been seen by 3 reviewers, whose comments are appended below. You will see that they find your work of potential interest. However, they have raised quite substantial concerns that must be addressed. In light of these comments, we cannot accept the manuscript for publication, but would be interested in considering a revised version that fully addresses these serious concerns.

We hope you will find the Reviewers' comments useful as you decide how to proceed. Should additional work allow you to address these criticisms, we would be happy to look at a substantially revised manuscript. If you choose to take up this option, please highlight all changes in the manuscript text file, and provide a detailed point-by-point reply to the reviewers.

Editorially, we consider it especially important to address the methodological concerns of Reviewers #1 and #2. This will require extensive additional analysis to alleviate the referees' concerns about the strength of support for your conclusion. The reviewers (esp Reviewers #3 and #1) also highlight concerns regarding the wide ranging interpretation of your findings. Given the presence of potential confounds (including familiarity / relationship to the child, Ref #1), we ask you to carefully revise the discussion, remaining close to the data in your interpretations and including a transparent discussion of limitations. In addition to these changes, you will also need to revise the presentation of the methods and results throughout, as all referees highlighted issues with the clarity of presentation and therefore, the transparency of the experimental detail and results. This extends to reporting and interpreting statistics in line with our guidelines:

<https://www.nature.com/commspsychol/submit/submission-guidelines#statistical-guidelines>.

Finally, please note that as per our policy, any novelty claims must be removed from the manuscript, and that you will need to ensure greater clarity with regard to how your paradigm compares to existing approaches.

If the revision process takes significantly longer than five months, we will be happy to reconsider your paper at a later date, provided it still presents a significant contribution to the literature at that stage.

We understand that due to the current global situation, the time required for revision may be longer than usual. We would appreciate it if you could keep us informed about an estimated timescale for resubmission, to facilitate our planning. Of course, if you are unable to estimate, we are happy to accommodate necessary extensions nevertheless.

Please use the following link to submit your revised manuscript, point-by-point response to the

Reviewers' comments with a list of your changes to the manuscript text (which should be in a separate document to any cover letter) and any completed checklist:

[link redacted]

Please do not hesitate to contact me if you have any questions or would like to discuss the required revisions further. Thank you for the opportunity to review your work.

Best regards,

Jennifer Bellingtier

Jennifer Bellingtier, PhD
Senior Editor
Communications Psychology

EDITORIAL POLICIES AND FORMATTING

Editorial Policy: [Policy requirements](https://www.nature.com/documents/nr-editorial-policy-checklist.pdf) (Download the link to your computer as a PDF.)

Furthermore, please align your manuscript with our format requirements, which are summarized on the following checklist:

[Communications Psychology formatting checklist](https://www.nature.com/documents/commsj-psychol-style-formatting-checklist-article.pdf)

and also in our style and formatting guide [Communications Psychology formatting guide](https://www.nature.com/documents/commsj-psychol-style-formatting-guide-accept.pdf) .

* **CODE AVAILABILITY:** All Communications Psychology manuscripts must include a section titled

"Code Availability" at the end of the methods section. In the event of publication, we require that the custom analysis code supporting your conclusions is made available in a publicly accessible repository; please choose a repository that provides a DOI for the code; the link to the repository and the DOI must be included in the Code Availability statement. Publication as Supplementary Information will not suffice. We ask you to prepare and upload code at this stage, to avoid delays later on in the process.

*** DATA AVAILABILITY:**

All Communications Psychology research manuscripts must include a section titled "Data Availability" at the end of the Methods section or main text (if no Methods). More information on this policy, is available at <http://www.nature.com/authors/policies/data/data-availability-statements-data-citations.pdf>.

At a minimum the Data availability statement must explain how the data can be obtained and whether there are any restrictions on data sharing. Communications Psychology strongly endorses open sharing of data. If you do make your data openly available, please include in the statement:

We recommend submitting the data to discipline-specific, community-recognized repositories, where possible and a list of recommended repositories is provided at <http://www.nature.com/sdata/policies/repositories>.

If a community resource is unavailable, data can be submitted to generalist repositories such as [figshare](https://figshare.com/) or [Dryad Digital Repository](http://datadryad.org/). Please provide a unique identifier for the data (for example a DOI or a permanent URL) in the data availability statement, if possible. If the repository does not provide identifiers, we encourage authors to supply the search terms that will return the data. For data that have been obtained from publicly available sources, please provide a URL and the specific data product name in the data availability statement. Data with a DOI should be further cited in the methods reference section.

REVIEWER EXPERTISE:

Reviewer 1 emotional biases

Reviewer 2 empathic accuracy

Reviewer 3 emotional biases

Reviewer #1 (Remarks to the Author):

The hypothesis is interesting and makes sense; the studies do a fine job of testing it.

The relationship between this task and the 'classical' one as used in Silani et al (2013) needs to be explained more clearly. Their task also included tactile sensations, so in what sense is this paradigm more bodily? In what sense is it an advance?

Please explain more clearly in the main text what the advantages are of calculating the 'bodily-emotional' EB/AB as opposed to the 'classical' one. And consider using a different label for the former, as it does not appear to be any more or less bodily than the classical one.

A stronger EB towards one's children could also be functional in relation to social referencing, where children do not have a definite emotion or interpretation of the valence of a situation or a stimulus, and use the adult as a reference point. These are cases where the adult in effect shows the child what to feel; it would be counterproductive for the adult to try to identify the child's emotion here. It could enrich the paper to link the findings with this research as well.

The description of the procedure and setup is not entirely clear to me. It looks in Fig 1b like participants should be able to see their own hand just as well as the other participant's hand, but I think I understand the text to indicate that they have only tactile information about their own hand and only visual information about the other's hand. How does that work?

I do not understand how the stimulus validation worked. Some people might find velcro pleasant for example; was it just assumed that some stimuli are unpleasant or was this determined for each participant for each stimulus?

Image quality: The images of the stimuli in Fig 1 (a and b) are not of sufficient quality for me to see them properly.

It is indicated in the methods that the research was carried out in accordance with the Declaration of Helsinki, but this is not accurate given that the experiments were not pre-registered (See paragraph 35 here:

<https://www.wma.net/policies-post/wma-declaration-of-helsinki-ethical-principles-for-medical-research-involving-human-subjects/>)

p.13: "Partial correlation analysis yielded positive correlations between IRI empathy concern subscale and the classical emotional EB ($r(63) = .33, p = .008, p_{FDRcorrected} = .053$), suggesting that those who show higher EB also report more feelings of sympathy and concern for unfortunate others."

^[1]_{SEP}Theoretically, this should be surprising and demands explanation. It conflicts for example with work by Peter Hobson (Hobson, R. P. (1999). Beyond cognition: A theory of autism. Perspectives on the Nature of Autism, 253-281.)

The authors do give a reason for expecting mothers to exhibit a stronger EB towards their own children than towards unfamiliar children, but not for this more general statement.

p.14: "we found evidence supporting our main hypothesis that mothers show greater emotional EB towards their own child than an unfamiliar child. Importantly, this was the case for both classical emotional EB and the sensory-controlled, bodily EB measure, suggesting also a bodily emotional

egocentric effect.” It is not clear to me what inference the authors are suggesting we make here. ^[1]_{SEP}

Limitations: It would have been ideal to use familiar other children rather than unfamiliar other children as a control. As it is, we can't be sure whether the key difference is that the mothers' own children were their own or that they were more familiar to them.

Language: It would be good to get a native speaker to proofread the language. In particular, the use of the apostrophe in possessives is incorrect in most instances (e.g. in the abstract: 'adult stranger's dyads')

Reviewer #2 (Remarks to the Author):

I was interested in reviewing this work on egocentric biases in mothers towards their children, which seems like a timely and relevant topic that may advance understanding in the field.

However, I could not always follow the authors' reasoning. I had quite a few methodological questions that emerged during reading that made it difficult for me to evaluate this work in its entirety.

(1) The first and major claim of the paper seems to be that mothers display a stronger egocentric bias (EB) towards their own children's than towards other children's bodily feelings. The results seem to support this idea, although I have to say that it was not 100% clear to me how "bodily feelings" were defined and whether the terms feelings and emotions were used synonymous here. I also have various questions with regard to the methods (see below). Second, it is implied that the mothers' EG bias is adaptive. I do not believe that convincing evidence was presented for this claim (see also comment #21). Additionally, the authors claim to have validated a new, "bodily" emotional egocentricity task. I tend to disagree (see also the following comments #2-#5; #12-#14).

(2) The authors argue that they present a new "bodily" emotional egocentricity task. When reading this, I was under the impression that the original emotional egocentricity task as presented by Silani et al. (2013) did not incorporate tactile stimulation. However, this is not the case, as the original task features visuo-tactile stimulation.

(3) I did not understand why the newly added "bodily" scores are considered "bodily" as compared to the "classical" scores in this task. When looking at the methods and the calculation of the scores, it seemed clear to me all terms pertain to ratings of experienced (self) or inferred (other) pleasantness of touch, e.g. for EB:

$$EB = ((\text{Incongruent Other Unpleasant} - \text{Congruent Other Unpleasant}) + (-1 * (\text{Incongruent Other Pleasant} - \text{Congruent Other Pleasant}))) / 2$$

Bodily EB =

$$(|(\text{Other Incongruent Unpleasant} - \text{Only Vision Unpleasant})| + |(\text{Other Incongruent Pleasant} - \text{Only Vision Pleasant})|) / 2$$

If I understand it correctly, for the classical EB, assumed pleasantness of touch in the congruent trials is subtracted from assumed pleasantness in the incongruent trials, whereas, for the bodily EB, the assumed pleasantness of touch before the trials (also called bodily priors) is subtracted from the incongruent trials (i.e., only the subtrahend changes). Naming one type of score "bodily" and the other "classical" seems a bit misleading to me as all scores pertain to ratings of pleasantness of

bodily experiences.

(4) I did not understand why for the classical scores difference scores are used, but for the bodily scores absolute difference scores. Bodily EB/AB scores are thereby constrained to be positive, whereas classical EB/AB can also be negative. This also influences the interpretation of the scores, as the direction of a bias can only be disentangled when using differences scores, not absolute differences. However, just as for classical EB/AB, it may be possible for the bodily EB/AB scores that the direction of differences varies between individuals, which I think should be captured in the scores. It seems to me that the bias can only be considered egocentric when a person imposes their own experience onto the other person's experience. For the classical EB score, this would mean that a person A overestimates the assumed pleasantness of touch for person B when person A receives a pleasant stimulation while B receives an unpleasant stimulation ("other incongruent unpleasant" condition) and underestimates the pleasantness in the "other incongruent pleasant" condition, resulting in an overall positive difference score. However, if a person answers in the opposite way (resulting in a negative score), this could probably not be considered egocentric, as the person would not use their own bodily sensations to judge the other. In essence, I think the bodily EB/AB should also be calculated using difference scores.

(5) Overall, I was not really sure what the bodily EB/AB scores measure and why subtracting the "priors" from the response in the incongruent trials is considered an appropriate measure for a bodily bias. To me, it seems that both the "congruent" and the "only vision" condition are different ways to create a reference to which the incongruent condition can be compared. I currently cannot follow as to why the "only vision" condition is better suited to "get internal bodily biases created by affective incongruency between self and other" (p. 5) than the congruence condition. In the congruence condition, it is assumed that a person A can rely on their own experience to make a judgment about the other person B (aligned experience). In the "only vision" condition (or "prior"), it is unclear what person A's inference is based on really. It is possible that person A still uses information about how they themselves would feel if touched by the different materials – it seems to me that we just do not know?

(6) I think it would be really helpful if the authors could provide examples of how they interpret the scores. For example, if classical EB is lower than zero, I would think that a person behaves directly opposite to what is hypothesized (i.e., not egocentric). How to interpret the bodily scores (in relation to how they are calculated)?

(7) It would be important to see the correlation between the EB/ bodily EB and the AB/bodily AB, given that both scores rely on overlapping information. More generally, I would be interested in a correlation matrix for all variables.

(8) I would like to see descriptive information (means and SDs) on how each condition was rated (before differences were computed).

(9) I would also be interested in within-person variability across trials. For example, in Experiment 1, there are 4 conditions, and there are 2 Blocks with 8 trials. This would mean that participants were subjected to each condition 4 times. In the statistical analyses, it is mentioned that scores were computed as the average of the difference. I was wondering how variable/ reliable the answers for each condition were?

(10) Given that no power analysis was conducted, I am not sure that the sample sizes are appropriate. For example, in Experiment 2, there are only 34 mothers interacting with their own and 34 mothers interacting with another child. These sample sizes seem quite low for dyadic research questions. Sensitivity analyses could help to see what effect sizes could be detected with these sample sizes.

(11) The description of the analyses is not detailed enough to replicate the analyses. For example, it would be important to know how the variables were coded, whether REML or FIML was used, how

predictors were centered in the multilevel model, and so forth.

(12) Given that the EB/AB rely on a difference score (which can be positive or negative) and the bodily EB/AB rely on absolute differences (which can only be positive), testing whether these scores are non-zero does not appear to be equally meaningful to me. It seems somewhat unlikely to me that the average absolute differences scores for bodily AB/EB (which are, e.g., based on ratings from 0-100 in Experiment 1b) approach zero (on average). Given that participants will have some variation in their answers and given that there will be measurement error, it seems likely that some differences will emerge, and as all differences will be positive, the average score is likely to be different from zero. To me, this does not necessarily imply that these differences are inherently meaningful.

(13) For the EB/AB scores: An average score of zero for AB as in Experiment 1b (looking at Figure 2b), seems to imply here that there is a similar amount of people with negative scores as there are with positive – what does that mean? How do the authors interpret this finding? To me, it seems to show that there is no AB.

(14) The authors used the term “validation” when showing that the biases in their new task are non-zero and conclude that the task was validated. However, I think a validation would have to be more conclusive than that (also given comments #12/13) and should also be more comprehensive. I would expect more information about validity and reliability.

(15) The authors write that they “test our hypothesis regarding the size of the biases (AB and EB) [...]” (p.7), but I was not sure which hypothesis this pertains to/ I did not find it in the text.

(16) It is reported that “neurotypical adults” were assessed (p. 7). If I understand it correctly, the autism questionnaire (AQ) was administered. However, a score on the AQ should not be interpreted as determining or excluding neurodiversity (since it is a self-report measure and also does not account for all types of neurodiversity, e.g., ADHD). It is also not clear whether any individuals scored above a cut-off and were excluded.

(17) Overall, I was wondering whether outliers in the data were identified. Some of the data points in the figures seem extreme (e.g., Figure 3A, EB).

(18) It was difficult for me to follow the results. I would prefer to see the full model, probably in a table (this may be a personal preference).

(19) I was not sure why the authors seem to consider only the correlations with caution given the limited sample size (p. 13), but not the other results.

(20) Also, with regard to the correlation between IRI-EC and classical EB, two p values are reported, and it was not clear to me which p-value the authors consider appropriate (uncorrected or corrected), and it was also unclear to me how the p-value was corrected. Finally, it was unclear to me why they controlled for age (given that there are many other potential confounds that cannot be controlled for).

(21) The authors claim: “Thus, this novel finding suggests the self-other distinction has an adaptive and changing role in parenting and opens new avenues for research in the relationship between the self-other distinction, affective regulation and parental interoception.” (p. 15). It seems to me that the only finding suggesting adaptive value of EB is the correlation with the IRI-EC, which, however, turned non-significant when applying correction methods (see comment #20). I would not agree that the cited claim is warranted.

(22) Finally, coming from a different research tradition, it was difficult for me to think about bias without accuracy or “truth” (see, e.g., West & Kenny, 2011). I would think that, given that there are interindividual differences in how pleasant a touch can be perceived in general, people also differ in how pleasant they consider a “shared” or “aligned” experience (such as being touched with the same material). In a shared experience, two people’s experience may be similar, but maybe still not the same. Furthermore, projecting one’s own experience to another person could be considered

egocentric even when this projection (or also termed assumed similarity) is mostly accurate. Maybe the authors could take this into consideration. See, e.g.: West, T. V., & Kenny, D. A. (2011). The truth and bias model of judgment. *Psychological Review*, 118(2), 357–378.
<https://doi.org/10.1037/a0022936>

Reviewer #3 (Remarks to the Author):

The authors have built and validated a new emotional egocentricity task and, with it, examined how the egocentric and altercentric biases (EB and AB) of mothers differ between judging how their own child and an unknown child feels.

They have first replicated previously reported EB and AB effects and showed the existence of such biases for bodily signals. Then they have shown that mothers have a higher EB for their own child than for the unknown child, both for the emotional and bodily biases.

The study is well-designed and their results may have important implications for the field or even clinical parental psychology in general. But such implications require more cautious interpretations. I don't have any major concerns but a series of minor issues:

1) The interpretation according to which the mothers' egocentricity bias for their children is actually adaptive is not convincing enough yet. I suggest to document further empirical and theoretical evidence in favour of this interpretation. If the authors have already made a thorough literature search then they should acknowledge it is a novel and speculative interpretation that remains to be confirmed. I personally think that relying on egocentric bodily states is insufficient to provide adequate care. Children are more fragile and have more needs than adults and thus representing their bigger needs is not helped at all by relying on egocentric needs. But of course, I agree that egocentric needs can remind us of our children's needs if we haven't represented their needs yet. In any case, I find it not intuitive that the authors would expect that the mothers would show higher EB for their own child.

2) Biases based on internal bodily signals must be better explained. I have a hard time to grasp it despite I am familiar with the EB and AB.

3) The Figure 1 must be improved: the stimuli pictures are pixelated and too small. It would be best to see a photograph of the top view (or even better: participant centred view) of the setup.

4) In study 1 second sample, they did not find a significant AB, which I agree can be due to interindividual differences but I recommend the authors to test for demographic and questionnaires data differences between the two samples. E.g., higher % of women in sample 1.

5) In study 2, for mothers emotional biases, please report as well whether the AB and EB were significantly different from zero.

6) In study 2, for mother bodily biases, I don't follow why they didn't test the same interaction pair x type of bias. Instead they tested effect of pair separately for EB and AB.

7) Line 265: "empathic concern" not "empathy concern".

8) Another interpretation regarding the authors' main finding is that mothers were worried to perform better with an unknown child than a known child. There is plenty of literature in social psychology showing that less uncertainty, more familiarity, more similarity, being in "comfort zone" lead us to adopt a more heuristic mode of functioning which is to rely on egocentric information. In contrast, an unknown child is more likely to activate a more controlled and effortful consideration of what the child thinks.

Altogether, given the message the authors want to convey about mothers and their child I would recommend a more cautious tone. The authors should remind the readers that the effects are

statistically significant at the group level but maybe not so significant for everyday life given the effect sizes and the high interindividual heterogeneity.

Rebuttal Letter - COMMSPSYCHOL-22-0043A

We would like to thank the three anonymous reviewers for providing helpful critical feedback on our manuscript. We believe that we have answered all the Reviewers' concerns and that it greatly improved the manuscript with respect to its original version (please see below a point by point response, and main changes are written in blue in the revised manuscript). Specifically, we:

1. Explained in more details our task and its relevance to investigate our hypotheses, and why we believe it is a bodily task. Following the reviewers' comments, we have revised our terminology and have chosen to now call the "bodily-emotional" EB/AB as "*sensory-controlled*" EB/AB.
2. As suggested, we have added relevant developmental theoretical interpretations to our findings, such as social referencing and epistemic trust, and related limitations to our discussion.
3. We clarified the description of our experimental design and added further details about the set-up and procedure in the introduction and in the methods (including justification of our sample size).
4. Following the reviewers comments we also explored the possibility of outliers and added an 'outliers section' in our results, as well as supplementary results.
5. Last, we revised our discussion to remain closer to the data, including a discussion of the limitations of the study.

Reviewer #1

The hypothesis is interesting and makes sense; the studies do a fine job of testing it.

R1.1. The relationship between this task and the 'classical' one as used in Silani et al (2013) needs to be explained more clearly. Their task also included tactile sensations, so in what sense is this paradigm more bodily? In what sense is it an advance?

Response: We thank the reviewer for giving us the opportunity to explain better the novelty of our task, and why we believe it is a more bodily task. While we agree with the reviewer that both tasks involve felt tactile sensations, there are two important differences: (1) the nature of the seen, vicarious touch is different between the tasks, with one being embodied and the other not; (2) the tactile stimulation parameters differ between the paradigms. Both of these differences have important and well-established implications in sensory neuroscience, as we now explain in the manuscript and below.

(1) First as regards, the 'seen touch'. There are two key reasons why our task can be considered a bodily task. We now explain each of these reasons in the text as follows:
p. 5-6:

"Dyads of participants were tested in proximal space, using a set-up that allows tactile stimulation of participant's own unseen forearm at the same time as they are watching

another participant's forearm being touched. This approach moves beyond the utilisation of a disembodied stimulus, as seen in previous tasks, to include a vicarious touch stimulus (see also below and Figure 1). Consequently, this setup facilitates a concurrent experience of self, through unisensory touch (felt touch) and other, through unisensory observed touch (vicarious seen touch). As depicted in Figure 1, participants were asked to judge how pleasant was the touch they felt (self) or saw (other), without knowing in advance which question they would need to answer. This required their attention to be focused on both self and other experiences. This simultaneous sensory experience can either be congruent or incongruent in terms of pleasantness (pleasant or unpleasant; see the design and set-up in Figure 1).

A large tradition of sensory and multisensory cognitive and neuroscientific research has studied vicarious (or mirror) sensory phenomena. Notably, studies within this domain have highlighted the unique effects of seeing body parts being stimulated, in contrast to other non-body stimuli, as well as many other embodiment effects on vicarious touch (e.g.³¹⁻³³). For example, neuroimaging studies have identified areas within the occipito-temporal cortex, such as the extrastriate body area³⁴, and fusiform body area³⁵ responding selectively to seen bodies and body parts, and a corresponding topographic map of viewed body parts throughout occipitotemporal cortex³⁶. This specificity of the human body (both our own and that of others) as a visual stimulus and a representation, particularly during vicarious responding, is one of the two reasons we named this task the 'bodily Emotional Egocentricity Task'. This task extends beyond 'imagining' the other person being touched with a stimulus while one feels touch on their own body, it also encompasses the representation of the other person's body and the seen touch itself during the vicarious touch experience. The second, related reason pertains to the capability of our task to provide subject-specific, important control (non-concurrent) measures for both the felt and the seen unisensory perception of touch. Various studies on vicarious, or mirror perception and visual enhancement of touch have shown that there are both intersubjective and intermodal effects on concurrent perception of felt and seen touch. While traditional emotional egocentricity tasks aim to mitigate some of these effects by employing emotionally 'congruent' conditions as a baseline, these tasks still remain within the context of concurrent, multimodal conditions. By contrast, our task gives the opportunity to have this control, as well as unisensory measures for each modality, without the concurrent stimulation. Indeed, in addition to the congruent and incongruent conditions tested during the main task, we also tested unisensory (non-concurrent) perception for both the felt touch on the self and the seen touch on the other. These control conditions serve the purpose of disentangling sensory from the multisensory effects inherent to the task. Importantly, none of these conditions are susceptible to the top-down effects that object or animal stimuli could potentially exert on participants, as often encountered in the original egocentricity tasks that use pictorial stimuli for the seen touch and to characterise the felt touch."

(2) Second, as regard to the 'tactile stimulation parameters', we now explain the key difference and advance of our task as follow:

Page 7:

"Furthermore, our task not only allows for participants to see the touch the other person is receiving on their body with a given material, but it also allows us to match the specific stimulation parameters of the tactile and the vicarious stimuli, i.e. the self and other

stimuli. Participants are in the same room, and they can see and feel specific tactile stimuli that could be either congruent or incongruent in pleasantness, applied in a specific synchronous velocity of 3 cm/s (as done in Filippetti et al., 2019 ; see also Methods). This sensory matching is not possible in the original task¹⁶, as participants are only presented with pictorial stimuli of ‘what’ touches the other person (not ‘how’ the other person is actually touched), while also seeing similar pictorial stimuli representing what is supposedly touching their body, as they are being touched with different materials out of sight (rather than actual spiders, swans etc). In light of the aforementioned distinctions, our task constitutes a more embodied rendition of an egocentricity task and allows us to test specifically the existence of emotional EB and AB in a bodily context.”

R1.2. Please explain more clearly in the main text what the advantages are of calculating the ‘bodily-emotional’ EB/AB as opposed to the ‘classical’ one. And consider using a different label for the former, as it does not appear to be any more or less bodily than the classical one.

Response: We thank the reviewer for this point that allows us to clarify the value of our added control conditions and the more bodily nature of our tasks and biases, as better explained now in the introduction (see response to point R1.1 above). It also allows us to revise our terminology and we chose to now call the “bodily-emotional” EB/AB as “sensory-controlled” EB/AB.

Classically, based on Silani et al. 2013 study, emotional egocentric and altercentric biases are computed as the difference in pleasantness between incongruent and congruent trials. These biases can account for the differential effects of valence between such conditions but they cannot distinguish between the effects of sensory stimulation (i.e. being exposed to sensory stimuli in different modalities) and socially congruent stimulation (being exposed to both self and other stimuli), as both of these conditions share these characteristics. For example, some people may find visual stimuli far more salient than tactile stimuli. So, the difference between congruent and incongruent trials may be explained by the fact that they focused more on the seen aspects of the experience rather than the felt ones. Quantitatively, the effect may be the same with other people that focused more on the other rather than the self, but the reasons for this bias would be different. Accordingly, a further, useful, complementary way to look at emotional egocentric and altercentric biases is to compute the discrepancy between the pleasantness rating we give in incongruent (multisensory and socially concurrent stimulation) trials and pleasantness ratings given during unisensory trials (i.e. seen or felt stimulation, without the socially concurrent stimuli). In order to compute an emotional egocentric bias based on unisensory baselines, we can subtract the pleasantness for the visual unisensory condition (vision prior), to get the unique contribution of the self-feeling on the EB bias. Similarly, to compute an altercentric bias based on unisensory baselines, we can subtract the pleasantness for the tactile unisensory condition (feel prior), to get the unique contribution of the vision of the other on the AB bias. For these reasons we call the new biases the “sensory-controlled biases”. Furthermore, having this added, complementary control

measures this has even greater added value when testing pairs of participants that may have different roles (e.g. parent, child), age differences and familiarity differences (e.g. own-other child), with all of which possibly causing differences in unisensory and multisensory, and multiagent attention and perception.

We now explain this more clearly in the introduction as follow:

Page 6-7:

“This adapted paradigm facilitated the computation of both classical emotional EB and AB (subsequently referred to as “classical bias”, EB and AB), as well as new sensory-controlled EB and AB (subsequently referred to as “sensory-controlled bias”sEB and sAB– see Figure 1B, and Methods for details). Classically, based on the study by Silani et al. (2013), emotional egocentric and altercentric biases are computed as the difference in pleasantness between incongruent and congruent trials. A supplementary and beneficial approach to understanding these biases is to gauge the disparity between the pleasantness provided during incongruent trials (which involve multisensory and socially concurrent stimulation) and the pleasantness ratings given during unisensory trials (i.e. trials involving either seen or felt stimulation, devoid of concurrent social stimuli). To compute an emotional EB using unisensory baselines, we can subtract the pleasantness rating of the visual unisensory condition (vision prior). This subtraction illuminates the distinct and unique contribution of the self feeling to the EB bias. Similarly, to compute an AB grounded in unisensory baselines, we can subtract the pleasantness rating of the tactile unisensory condition (feel prior). This computation unveils the unique influence of observing another person’s touch on the AB. Having these added, complementary control measures becomes especially valuable when testing pairs of participants with diverse roles (e.g. parent, child), age disparities and familiarity variation (e.g. own-child or. other unfamiliar-child). Such factors can potentially yield differences in unisensory, multisensory, and multiagent attention and perception.”

R1.3. A stronger EB towards one’s children could also be functional in relation to social referencing, where children do not have a definite emotion or interpretation of the valence of a situation or a stimulus, and use the adult as a reference point. These are cases where the adult in effect shows the child what to feel; it would be counterproductive for the adult to try to identify the child’s emotion here. It could enrich the paper to link the findings with this research as well.

Response: We thank the reviewer for suggesting this intriguing and relevant interpretation to our finding. We have added this and another related social interpretation on the possible use of a more heuristic mode of functioning as a possible explanation of the observed larger egocentric bias in the mother ‘own’ group (along a suggestion by reviewer 3, see R3.8).

We now explain it in more detail in the discussion (integrated with the suggestions from R2.1. R3.1 and R3.8):

Discussion Page 17-18:

“..... Furthermore, our results are also in line with the “social referencing theory” (Klennert et al., 1986) according to which the greater egocentricity bias of mothers towards their

own child could also be related to the parents' social role as a reference point for guiding the child's response. Children often seek their caregiver's responses (e.g., facial cues) to guide their own responses, with caregivers acting as significant figures from whom children gather informational cues. In the context of our findings, this larger egocentricity bias could signify that parents focus more on their own affective responses to communicate and guide their child, rather than solely attending to their child's reaction. This is also consistent with studies suggesting that the parent's role as source of information is intertwined with their adaptive or pedagogic epistemic role within a dynamic environment (Csibra & Gergely, 2009). Consequently, the greater egocentricity bias within the 'own group' may highlight the parent's function as a reliable source of information transmitted to their child, thus facilitating social learning (...)"

R1.4. The description of the procedure and setup is not entirely clear to me. It looks in Fig 1b like participants should be able to see their own hand just as well as the other participant's hand, but I think I understand the text to indicate that they have only tactile information about their own hand and only visual information about the other's hand. How does that work?

Response: We apologise if our original description of this set-up was not clear. We have now added further details about the set-up and procedure in the introduction and in the methods. Moreover, we have added a picture of the set-up in Figure 1C, so the reader can better see that participants could not see each other's face (curtain), could only see the forearm of the other participant but could not see their own forearm based on specially constructed boxes with different opaque and transparent sides.

We have now added more details in the methods about the set-up as follow:

Page 21:

"Two custom-made boxes were designed specifically to allow a direct view of the other participant, while hiding the self. This configuration allowed the experimenter to simultaneously see and be able to touch both participants synchronously (dimensions: 40*17*20cm). The boxes were positioned in a mirrored manner, so that they were respectively open on the right and left sides, granting each participant the vision of the other participant's forearm exclusively. The upper part of the box was made opaque to block the view of their own forearm. Participants were seated next to each other, separated by a curtain to prevent any visual contact with each other. They were touched on their respective right and left forearms (see Figure 1C). The distance between the respective right and left index fingers of the two participants were kept at a constant distance of 30 cm (participants rested their index fingers on a tape affixed inside the box)."

A. Design Main Task

		What Participant SEE = OTHER (O)			
		Pleasant		Unpleasant	
What Participant FEEL = SELF (S)	Pleasant	S O	S O	S O	S O
	Unpleasant	S O	S O	S O	S O

C. Set-up Experiment 1

B. Procedure

1. Prior Estimates - Unisensory

2. Self and Other Pleasantness Estimates

D. Set-up Experiment 2

Figure 1. (...) C. *Experimental Set-up* for Experiment 1a and 1b. D. with a pictorial representation of the set up with two participants separated by a curtain. (..)

R1.5. I do not understand how the stimulus validation worked. Some people might find velcro pleasant for example; was it just assumed that some stimuli are unpleasant or was this determined for each participant for each stimulus?

Response: We thank the reviewer for raising this issue and allowing us to clarify this validation phase in more details. The choice of the fabrics was based on pilot data from a different sample (as used also in Filippetti et al. 2019), whereby the pleasantness of 12 materials of different degrees of pleasantness was rated in visual only, tactile only, and visuo-tactile conditions. Based on these data, we chose the two most pleasant and two least pleasant fabrics with similar visual appearance (i.e. size and colour), as rated by 12 participants. Therefore, our choices of pleasant and unpleasant fabrics were derived from pre-designated stimuli terms based on central hypotheses and piloting performed prior to our main experiments.

Moreover, our sensory baselines (felt and seen) allowed us to check for the individual preferences, and ensure that participants do not prefer for example the scourer to the cotton ball. We have now elaborated on these criteria in the outliers sections (see below). Note that none of the participants in Experiment 1b and Experiment 2 (children group) were excluded based on that criteria. In the mothers groups of Exp 2, 6 out of 67 could be excluded on this criteria (i.e., rated the pleasant stimuli as less pleasant than the unpleasant one). However,

given that analysis with and without these 'outliers' remained the same we kept these participants in (see supplementary for our main findings without these 6 possible outliers). Moreover, please note that previous studies did not exclude participants based on this criterion as it does not allow to capture the individual differences between participants. This information is now included in the manuscript as follow:

Page 21: *Fabrics piloting*:

"The selection of fabrics was informed by the same pilot data used in the study by Filippetti et al. (2019). In this pilot phase, the pleasantness of 12 materials of different degrees of pleasantness was rated under visual-only, tactile-only, and visuo-tactile conditions by an additional group of 12 participants. Based on these data, we opted for the two most pleasant and two least pleasant fabrics, while ensuring similar visual attributes (i.e. size and colour). As such, our choices of fabrics perceived as pleasant or unpleasant were derived from pre-established stimuli, built on core hypotheses and piloting conducted prior to our primary experiments."

Page 25-26:

"Outliers detection. For all experiments, we employed two methods to identify outliers. First, we examined participants' average response to the 'Tactile only' baseline unisensory pleasant condition ("Tactile only pleasant") compared to the 'Tactile only' baseline unisensory unpleasant condition ("Tactile only unpleasant"). Individuals who rated the unpleasant stimuli as more pleasant than the pleasant stimuli (e.g., rating the touch of a scourer higher on average than that of a cotton ball), were highlighted as potential outliers. This examination identified 6 potential outliers only within the mother groups in Experiment 2. No outliers were identified in Experiment 1b nor in the children group in Experiment 2. In Experiment 1a, where we lacked a unisensory baseline condition, we compared the 'self pleasant congruent' condition to the 'self unpleasant congruent' condition. This comparison yielded one potential outlier. Subsequently, we conducted our analyses both with and without these potential outliers. However, the results remained consistent, leading us to retain them in the sample to account for possible individual differences (see SM for results without outliers; Tables 13S, 15S).

Second, we also examined potential outliers, based on criteria involving score above or below 2.5 SD from the group's average ratings scores across the different conditions. These analyses revealed one potential outlier in Experiment 1b, three in the mother group, and two in the children group in Experiment 2. Once again, our analyses were performed with and without these outliers, and as the observed effects remained unchanged, we decided to retain them in the sample (see SM for results without outliers; Tables 14S, 16S, 17S)."

R1.6. Image quality: The images of the stimuli in Fig 1 (a and b) are not of sufficient quality for me to see them properly.

Response: We have now uploaded a higher resolution figure.

R1.7. It is indicated in the methods that the research was carried out in accordance with the Declaration of Helsinki, but this is not accurate given that the experiments were not pre-registered (See paragraph 35 here:

<https://www.wma.net/policies-post/wma-declaration-of-helsinki-ethical-principles-for-medical-research-involving-human-subjects/>)

Response: We thank the reviewer for pointing this out, as we admit we had not realised this clause in the declaration. We have now added this info in the methods section, as follows:

Page 20:

“All three experiments were conducted in accordance with the Declaration of Helsinki (except for pre-registration) and were approved by the Ethics Committee of the Research Department of Clinical, Educational and Health Psychology, University College London”.

R1.8. p.13: “Partial correlation analysis yielded positive correlations between IRI empathic concern subscale and the classical emotional EB ($r(63) = .33$, $p = .008$, $pFDRcorrected = .053$), suggesting that those who show higher EB also report more feelings of sympathy and concern for unfortunate others.” Theoretically, this should be surprising and demands explanation. It conflicts for example with work by Peter Hobson (Hobson, R. P. (1999). Beyond cognition: A theory of autism. Perspectives on the Nature of Autism, 253-281.)

The authors do give a reason for expecting mothers to exhibit a stronger EB towards their own children than towards unfamiliar children, but not for this more general statement.

Response: We thank the reviewer for this comment, and giving us the opportunity to discuss this result further. Indeed, this result could at first be surprising, suggesting that people with higher empathic concern exhibit higher egocentric biases. But several considerations need to be taken into account.

First and most importantly, these results are from the mother-child experiment, which could bias the results due to the context. The IRI-empathic concern subscale is classically a measure for sympathy and concern for unfortunate others. So indeed, at first it could seem in contradiction with an egocentric bias; as typical, greater EB can be seen as less empathy or less ‘other-oriented’. Noticeably, in adult-adult dyads in Experiment 1, no correlation reached significance between IRI subscales and egocentric biases. The only significant correlation was found between the AB and the IRI fantasy scale, suggesting that those who show higher altercentric bias also report higher tendency to transpose themselves imaginatively into the feelings and actions of fictitious characters (see SM).

Second, in the present study we hypothesised that a parent's bodily egocentricity towards their child could constitute an adaptive bias and, in that case, egocentric biases may not necessarily signify failures of empathic understanding, but serve anticipatory regulation or a reference point.

Moreover, in the current version, based on this comment by the reviewer, we explored further our finding on the positive correlation between IRI empathic concern subscale and EB

and conducted two separated correlations (those who paired with their own child and those who were paired with a stranger), and found that the positive correlation is only significant in the 'own group' (EB: $r=.42$, $p=.02$ and sEB: $r=.58$, $p<.001$), with a trend in the "other" group (EB: $r=.32$, $p=.10$ and sEB: $r=.33$, $p=.08$). As such, our results suggest that the more sympathy and concerns for others a parent has, the larger their egocentricity bias for their own child is (and not in general). Relatedly, according to Lamm et al., 2019, "empathy has to be distinguished from sympathy and compassion (...) sympathy and compassion do not only involve affect sharing or "feeling as" the other, but rather a "feeling for" and thus of concern for the other", which could be measured by the IRI-empathic concern and could be consistent with higher EB in the 'own' mother group.

We thus removed the original, general statement from the results section and added this post hoc analysis to the results section.

Page 15:

"To further explore these correlations we explored the two groups separately (those who paired with their own child and those who were paired with a stranger), and found that the positive correlation is only significant in the 'own group' (classical EB: $r_{(34)}=.42$, $p=.02$ and sensory-controlled EB: $r_{(31)}=.58$, $p<.001$), with a trend in the "other" group (classical EB: $r_{(30)}=.32$, $p=.10$ and sensory-controlled EB: $r_{(30)}=.33$, $p=.08$). Similarly, the negative correlation between sensory-controlled EB and personal distress was only significant in the group that was paired with their own child ($r_{(31)}=-.44$, $p=.02$), and in the other group showing only a trend ($r_{(30)}=-.36$, $p=.06$)."

R1.9. p.14: "we found evidence supporting our main hypothesis that mothers show greater emotional EB towards their own child than an unfamiliar child. Importantly, this was the case for both classical emotional EB and the sensory-controlled, bodily EB measure, suggesting also a bodily emotional egocentric effect." It is not clear to me what inference the authors are suggesting we make here.

Response: We agree with the reviewer that the end of this sentence was unclear, and we have now removed the part "suggesting also a bodily emotional egocentric effect". We now extend the conclusions in the discussion as follow:

Page 16:

"By employing the bodily emotional egocentricity task, we uncovered evidence that supports our central hypothesis - specifically, that mothers exhibit a more pronounced emotional EB towards their own child in comparison to an unfamiliar child. Importantly, this observation held true for both classical EB and sensory-controlled EB measurements. This suggests that in situations where mothers witness their own child undergoing tactile sensations different from their own, they tend to project their own sensations onto their child. This bias cannot be explained by any relative differences in valence between self and the child during simultaneous stimulation (original bias calculations). Nor can it be explained by any biases in how the mothers perceive such stimuli on their own bodies

while experiencing them in isolation (novel bias calculations employing unisensory controls).”

R1.10. Limitations: It would have been ideal to use familiar other children rather than unfamiliar other children as a control. As it is, we can't be sure whether the key difference is that the mothers' own children were their own or that they were more familiar to them.

Response: We agree with the reviewer that our effects may be related to familiarity, but as we explain below this is both central to our hypothesis and to a lesser degree a possible confound that could be controlled by testing other familiar children (friends). We acknowledge this limitation and its complexity in our manuscript, as follows:

Page 20:

“Thirdly, in our design, we compared being paired with one's “own” child vs an unfamiliar child, which did not directly control for the influence of familiarity. However, we acknowledge that the role of familiarity in this study is not solely a ‘confounding’ factor; it is also a hypothesised main effect. Specifically, we do believe that parents might exhibit distinct biases towards their own child due to familiarity with them, and with their habitual needs and their need for parental regulation. While there are other dimensions of familiarity that can act as confounds, such as perceptual or attentional familiarity, ease during testing, and lower-level perceptual factors, these could be partially controlled by testing other familiar children in future studies. Nevertheless, we note that this approach could introduce new confounds, given that family and friendship roles and levels of familiarity are not identical beyond superficial levels. Importantly, our unisensory baselines do control for some of these lower-level perceptual familiarity aspects, as they enable the calculation of biases based on unisensory measures of both seen and felt touch, unique to each tested child.”

R1.11. Language: It would be good to get a native speaker to proofread the language. In particular, the use of the apostrophe in possessives is incorrect in most instances (e.g. in the abstract: ‘adult stranger’s dyads’)

Response: The revised version of the manuscript has now been proofread by a native English speaker, and we reformulated many sentences to improve clarity.

Reviewer #2 (Remarks to the Author):

I was interested in reviewing this work on egocentric biases in mothers towards their children, which seems like a timely and relevant topic that may advance understanding in the field. However, I could not always follow the authors' reasoning. I had quite a few methodological questions that emerged during reading that made it difficult for me to evaluate this work in its entirety.

R2.1. The first and major claim of the paper seems to be that mothers display a stronger egocentric bias (EB) towards their own children's than towards other children's bodily feelings. The results seem to support this idea, although I have to say that it was not 100% clear to me how "bodily feelings" were defined and whether the terms feelings and emotions were used synonymous here. I also have various questions with regard to the methods (see below). Second, it is implied that the mothers' EG bias is adaptive. I do not believe that convincing evidence was presented for this claim (see also comment #21). Additionally, the authors claim to have validated a new, "bodily" emotional egocentricity task. I tend to disagree (see also the following comments #2-#5; #12-#14).

Response: We thank the reviewer for their feedback and giving us the opportunity to better explain (1) what we mean with bodily feelings and how they are defined in relationship to emotions, (2) the possibility that the biases we found are adaptive. We also respond fully below to the methodological and terminological issues the reviewer raises more specifically in his further comments below, regarding our methods and the validation of a 'bodily' task.

Regarding the first comment, indeed we used the word feelings and emotional feelings interchangeably to designate the pleasantness arising from the tactile stimulation, measured via the behavioural ratings of "how pleasant was the touch". In this, we admit that we just followed this existing research tradition that refers to the exploration of "emotional egocentricity biases" and more generally uses the word emotion to designate the judgement of the self-other affective states, via pleasantness ratings (e.g. Silani et al. 2013). We are also aware that different research traditions and studies use the terms feeling, emotion and affect in different and at times inconsistent ways and hence, in hindsight we acknowledge this terminological issue in the manuscript and importantly we now define exactly what we mean by 'bodily feelings' in text as follows:

Page 18:

"Noteworthy, in the current paper, we adhere to the established research tradition that explores "emotional EBs" and generally employs the term "emotion" to refer to the evaluation of self-other affective states through pleasantness ratings (e.g. Silani et al. 2013). Accordingly, we employ the terms "bodily" and "emotional feelings" as designating the pleasantness arising from tactile stimulation, as measured by the behavioural ratings of "how pleasant was the touch". Given this specific definition, the inferences that can be drawn from the present study are confined to judgments of tactile pleasantness, and do not necessarily encompass more intricate emotional states such as anger and happiness."

We also note that we have not used the word emotion on its own, except in the title, but only in association to "emotional egocentric/altercentric biases". In order to make it more consistent, we now changed the title to "**Mother knows best: Mothers are more egocentric towards their own child's bodily feelings**".

Second, regarding our claim that mother's EB is adaptive, we toned down our interpretations and stayed closer to our findings (i.e., a positive correlation between EB and empathic concern in the 'own' group). In addition, and in agreement with the reviewer that

our manuscript has not provided direct evidence for an ‘adaptive’ role, we have now suggested this interpretation as one possibility (i.e., anticipatory bodily and affective regulation) among other alternative interpretations such as ‘social-referencing’ (see comment R1.3) or ‘uncertainty condition’ (see comment R3.8). See page 17-18 for the extended discussion.

R2.2. The authors argue that they present a new “bodily” emotional egocentricity task. When reading this, I was under the impression that the original emotional egocentricity task as presented by Silani et al. (2013) did not incorporate tactile stimulation. However, this is not the case, as the original task features visuo-tactile stimulation.

Response: We thank the reviewer for this point, which is closely in line with Reviewer 1’s comment (R1.1), and, which gives us the opportunity to better explain our task, and why we believe it is a more bodily task. While we agree with the reviewer that both tasks involve visuotactile sensations, there are two important differences: (1) the nature of the seen, vicarious touch is different between the tasks, with one being embodied and the other not; (2) the tactile stimulation parameters differ between the paradigms. Both of these differences have important and well-established implications in sensory neuroscience, as we now explain in the manuscript and below.

(1) First as regards, the ‘seen touch’. There are two key reasons why our task can be considered a bodily task. We now explain each of these reasons in the text as follows:

p. 5-6:

“Dyads of participants were tested in proximal space, using a set-up that allows tactile stimulation of participant’s own unseen forearm at the same time as they are watching another participant’s forearm being touched. This approach moves beyond the utilisation of a disembodied stimulus, as seen in previous tasks, to include a vicarious touch stimulus (see also below and Figure 1). Consequently, this setup facilitates a concurrent experience of self, through unisensory touch (felt touch) and other, through unisensory observed touch (vicarious seen touch). As depicted in Figure 1, participants were asked to judge how pleasant was the touch they felt (self) or saw (other), without knowing in advance which question they would need to answer. This required their attention to be focused on both self and other experiences. This simultaneous sensory experience can either be congruent or incongruent in terms of pleasantness (pleasant or unpleasant; see the design and set-up in Figure 1).

A large tradition of sensory and multisensory cognitive and neuroscientific research has studied vicarious (or mirror) sensory phenomena. Notably, studies within this domain have highlighted the unique effects of seeing body parts being stimulated, in contrast to other non-body stimuli, as well as many other embodiment effects on vicarious touch (e.g.³¹⁻³³). For example, neuroimaging studies have identified areas within the occipito-temporal cortex, such as the extrastriate body area³⁴, and fusiform body area³⁵ responding selectively to seen bodies and body parts, and a corresponding topographic map of viewed body parts throughout occipitotemporal cortex³⁶. This specificity of the human body (both our own and that of others) as a visual stimulus and a representation, particularly during vicarious responding, is one of the two reasons we named this task the ‘bodily Emotional

Egocentricity Task'. This task extends beyond 'imagining' the other person being touched with a stimulus while one feels touch on their own body, it also encompasses the representation of the other person's body and the seen touch itself during the vicarious touch experience. The second, related reason pertains to the capability of our task to provide subject-specific, important control (non-concurrent) measures for both the felt and the seen unisensory perception of touch. Various studies on vicarious, or mirror perception and visual enhancement of touch have shown that there are both intersubjective and intermodal effects on concurrent perception of felt and seen touch. While traditional emotional egocentricity tasks aim to mitigate some of these effects by employing emotionally 'congruent' conditions as a baseline, these tasks still remain within the context of concurrent, multimodal conditions. By contrast, our task gives the opportunity to have this control, as well as unisensory measures for each modality, without the concurrent stimulation. Indeed, in addition to the congruent and incongruent conditions tested during the main task, we also tested unisensory (non-concurrent) perception for both the felt touch on the self and the seen touch on the other. These control conditions serve the purpose of disentangling sensory from the multisensory effects inherent to the task. Importantly, none of these conditions are susceptible to the top-down effects that object or animal stimuli could potentially exert on participants, as often encountered in the original egocentricity tasks that use pictorial stimuli for the seen touch and to characterise the felt touch"

(2) Second, as regard to the 'tactile stimulation parameters', we now explain the key difference and advance of our task as follow:

Page 6

"Furthermore, our task not only allows for participants to see the touch the other person is receiving on their body with a given material, but it also allows us to match the specific stimulation parameters of the tactile and the vicarious stimuli, i.e. the self and other stimuli. Participants are in the same room, and they can see and feel specific tactile stimuli that could be either congruent or incongruent in pleasantness, applied in a specific synchronous velocity of 3 cm/s (as done in³⁷ ; see also Methods). This sensory matching is not possible in the original task¹⁶, as participants are only presented with pictorial stimuli of 'what' touches the other person (not 'how' the other person is actually touched), while also seeing similar pictorial stimuli representing what is supposedly touching their body, as they are being touched with different materials out of sight (rather than actual spiders, swans etc). In light of the aforementioned distinctions, our task constitutes a more embodied rendition of an egocentricity task and allows us to test specifically the existence of emotional EB and AB in a bodily context."

R2.3. I did not understand why the newly added "bodily" scores are considered "bodily" as compared to the "classical" scores in this task. When looking at the methods and the calculation of the scores, it seemed clear to me all terms pertain to ratings of experienced (self) or inferred (other) pleasantness of touch, e.g. for EB:

$$EB = ((\text{Incongruent Other Unpleasant} - \text{Congruent Other Unpleasant}) + (-1 * (\text{Incongruent Other Pleasant} - \text{Congruent Other Pleasant}))) / 2$$

Bodily EB =

$$\frac{(|(\text{Other Incongruent Unpleasant} - \text{Only Vision Unpleasant})| + |(\text{Other Incongruent Pleasant} - \text{Only Vision Pleasant})|)}{2}$$

If I understand it correctly, for the classical EB, assumed pleasantness of touch in the congruent trials is subtracted from assumed pleasantness in the incongruent trials, whereas, for the bodily EB, the assumed pleasantness of touch before the trials (also called bodily priors) is subtracted from the incongruent trials (i.e., only the subtrahend changes). Naming one type of score “bodily” and the other “classical” seems a bit misleading to me as all scores pertain to ratings of pleasantness of bodily experiences.

Response: We thank the reviewer for this point that allows us to clarify the value of our added control conditions and the more bodily nature of our tasks and biases, as better explained now in the introduction (see response above to R2.2, but also in response to reviewer 1 that had a similar comment, i.e. R1.2). It also allows us to revise our terminology and we chose to now call the “bodily-emotional” EB/AB as “sensory-controlled” EB/AB.

Classically, based on Silani et al. 2013 study, emotional egocentric and altercentric biases are computed as the difference in pleasantness between incongruent and congruent trials. These biases can account for the differential effects of valence between such conditions but they cannot distinguish between the effects of sensory stimulation (i.e. being exposed to sensory stimuli in different modalities) and socially congruent stimulation (being exposed to both self and other stimuli), as both of these conditions share these characteristics. For example, some people may find visual stimuli far more salient than tactile stimuli. So, the difference between congruent and incongruent trials may be explained by the fact that they focused more on the seen aspects of the experience rather than the felt ones. Quantitatively, the effect may be the same with other people that focused more on the other rather than the self, but the reasons for this bias would be different. Accordingly, a further, useful, complementary way to look at emotional egocentric and altercentric biases is to compute the discrepancy between the pleasantness rating we give in incongruent (multisensory and socially concurrent stimulation) trials and pleasantness ratings given during unisensory trials (i.e. seen or felt stimulation, without the socially concurrent stimuli). In order to compute an emotional egocentric bias based on unisensory baselines, we can subtract the pleasantness for the visual unisensory condition (vision prior), to get the unique contribution of the self feeling on the EB bias. Similarly, to compute an altercentric bias based on unisensory baselines, we can subtract the pleasantness for the tactile unisensory condition (feel prior), to get the unique contribution of the vision of the other on the AB bias. For these reasons we call the new biases the “sensory-controlled biases”. Furthermore, having this added, complementary control measures this has even greater added value when testing pairs of participants that may have different roles (e.g. parent, child), age differences and familiarity differences (e.g. own-other child), with all of which possibly causing differences in unisensory and multisensory, and multiagent attention and perception.

We now explain this more clearly in the introduction as follow:

Page 6-7:

“This adapted paradigm facilitated the computation of both classical emotional EB and AB (subsequently referred to as “classical bias”, EB and AB), as well as new sensory-controlled EB and AB (subsequently referred to as “sensory-controlled bias” sEB and sAB— see Figure 1B, and Methods for details). Classically, based on the study by Silani et al. (2013), emotional egocentric and altercentric biases are computed as the difference in pleasantness between incongruent and congruent trials. A supplementary and beneficial approach to understanding these biases is to gauge the disparity between the pleasantness provided during incongruent trials (which involve multisensory and socially concurrent stimulation) and the pleasantness ratings given during unisensory trials (i.e. trials involving either seen or felt stimulation, devoid of concurrent social stimuli). To compute an emotional EB using unisensory baselines, we can subtract the pleasantness rating of the visual unisensory condition (vision prior). This subtraction illuminates the distinct and unique contribution of the self feeling to the EB bias. Similarly, to compute an AB grounded in unisensory baselines, we can subtract the pleasantness rating of the tactile unisensory condition (feel prior). This computation unveils the unique influence of observing another person’s touch on the AB. Having these added, complementary control measures becomes especially valuable when testing pairs of participants with diverse roles (e.g. parent, child), age disparities and familiarity variation (e.g. own-child or. other unfamiliar-child). Such factors can potentially yield differences in unisensory, multisensory, and multiagent attention and perception”.

R2.4. I did not understand why for the classical scores difference scores are used, but for the bodily scores absolute difference scores. Bodily EB/AB scores are thereby constrained to be positive, whereas classical EB/AB can also be negative. This also influences the interpretation of the scores, as the direction of a bias can only be disentangled when using differences scores, not absolute differences. However, just as for classical EB/AB, it may be possible for the bodily EB/AB scores that the direction of differences varies between individuals, which I think should be captured in the scores. It seems to me that the bias can only be considered egocentric when a person imposes their own experience onto the other person’s experience. For the classical EB score, this would mean that a person A overestimates the assumed pleasantness of touch for person B when person A receives a pleasant stimulation while B receives an unpleasant stimulation (“other incongruent unpleasant” condition) and underestimates the pleasantness in the “other incongruent pleasant” condition, resulting in an overall positive difference score. However, if a person answers in the opposite way (resulting in a negative score), this could probably not be considered egocentric, as the person would not use their own bodily sensations to judge the other. In essence, I think the bodily EB/AB should also be calculated using difference scores.

Response: We thank the reviewer for highlighting this difference, and in agreement with this comment we have now changed how we compute the new bodily sensory-controlled

egocentric and altercentric biases to follow the same logic as the classical one, and without using absolute scores and thus ensuring greater coherence between the two methodologies:

new calculated sensory-controlled EB = ((Incong UnP - Feeling UnP)-(Incong P - Feeling P))/2

new calculated sensory-controlled AB = ((Incong UnP - SeeingUnP)-(Incong P - Seeing P))/2

Our main findings remained the same as reported before (see Page 10 for results of Experiment 1, and Page 13-15 for results in Experiment 2).

R2.5. Overall, I was not really sure what the bodily EB/AB scores measure and why subtracting the “priors” from the response in the incongruent trials is considered an appropriate measure for a bodily bias. To me, it seems that both the “congruent” and the “only vision” condition are different ways to create a reference to which the incongruent condition can be compared. I currently cannot follow as to why the “only vision” condition is better suited to “get internal bodily biases created by affective incongruency between self and other” (p. 5) than the congruence condition. In the congruence condition, it is assumed that a person A can rely on their own experience to make a judgment about the other person B (aligned experience). In the “only vision” condition (or “prior”), it is unclear what person A’s inference is based on really. It is possible that person A still uses information about how they themselves would feel if touched by the different materials – it seems to me that we just do not know?

Response: Thank you for pointing this out, and allowing us to better explain the rationale behind the two types of biases. As explained in the response to point R2.3, the sensory-controlled EB/AB biases are taking into account sensory baselines, as explained now in the introduction page 7.

Please also note that as we now explain in the manuscript (please see above points R2.2 and R2.3) the unisensory control comparisons are complementary to the ‘congruent’ control comparison exactly because they can separate between multisensory and multiagent and unisensory, non-concurrent stimulation effects. To use the example that the reviewer suggests, while it is possible that during the “only vision” condition person A uses information about how they themselves would feel if touched by the different materials to vicariously perceive the pleasantness of touch on the other (please note that our task allows for specific, body-based vicarious touch perception as explained above, rather than just the imagination of tactile sensations based on pictures of object/animal stimuli) they are not actually touched in that same moment and hence they do not need to divide their attention, effort etc between agents or modalities. Hence, this control does not conflate the multisensory with the multiagent aspects of this experience. Please also see the following point where we explain the added value in interpretation of our novel bias measures.

R2.6. I think it would be really helpful if the authors could provide examples of how they interpret the scores. For example, if classical EB is lower than zero, I would think that a person

behaves directly opposite to what is hypothesized (i.e, not egocentric). How to interpret the bodily scores (in relation to how they are calculated)?

Response: We thank the reviewer for this comment. In response to comment R2.4, we have now changed the computation of the “sensory-controlled biases” (formerly bodily scores); so, both types of biases can now be compared to 0. Indeed, a negative classical and sensory-controlled egocentric bias (EB and sEB) suggests that a person is not egocentric. The two measures complement each other, as EB control for effects of congruent conditions (with imagined stimuli for vicarious perception), and sEB controls for any differences in unisensory, non-concurrent perception (with bodily stimuli for vicarious perception and more precise matching of sensory conditions). As suggested, we now give an example for each type of egocentric bias:

Page 7:

“To facilitate the interpretation of the biases, we can provide the following examples. Firstly, for the classical EB (controlled by congruency): consider a scenario where participant A is touched by cotton (pleasant), while simultaneously participant B is touched by a sponge (incongruent, unpleasant). If participant A rates the pleasantness of the sponge as more pleasant than when both participants are touched by a sponge, this will suggest that participant A is biased by the pleasant sensation from the felt cotton when evaluating the pleasantness of the sponge touching the participant B. This inclination will manifest as a positive classical egocentricity bias. For the *sensory-controlled EB*, analogous conclusions can be drawn. However, instead of comparing incongruent to congruent conditions, incongruent stimulation is compared to vicarious perception, wherein the self is not concurrently touched (the “vision only” condition). Thus, if during the ‘incongruency’ condition, a participant judges the pleasantness for the other person as more pronounced than in the “vision only” condition, this additional pleasantness (the discrepancy) can be attributed to what the participant projects to another person during incongruent, but concurrent multiagent stimulation. This differential in pleasantness does not arise from any inherent individual differences between seeing and feeling touch per se.”

R2.7. It would be important to see the correlation between the EB/ bodily EB and the AB/bodily AB, given that both scores rely on overlapping information. More generally, I would be interested in a correlation matrix for all variables.

Response: As suggested, we have now added correlation matrices for each experiment in SM (Tables 2S, 4S, 6S, 8S).

Indeed, as expected given that both biases share some information as the Reviewer says (ratings in incongruent conditions) and we used these biases as complementary measures to inform us on different aspects of the egocentric bias (as the unshared variance depends on the control we use in each case), EB and sEB are positively correlated.

R2.8. I would like to see descriptive information (means and SDs) on how each condition was rated (before differences were computed).

Response: We have now added all the descriptive information to the SM with Mean and SDs for each condition (congruent, incongruent, pleasant, unpleasant, own vs. other child) for each experiment and each group of experiment 2 (Tables 1S, 3S, 5S, 7S).

R2.9. I would also be interested in within-person variability across trials. For example, in Experiment 1, there are 4 conditions, and there are 2 Blocks with 8 trials. This would mean that participants were subjected to each condition 4 times. In the statistical analyses, it is mentioned that scores were computed as the average of the difference. I was wondering how variable/ reliable the answers for each condition were?

Response: Even though we do agree with the reviewer that investigating variability across time is interesting, we do not believe our design was optimised to test this added issue and that an analysis of the effect of repetition on 4 trials has enough power to be conducted. But future studies should look into this question further. We have now added in the discussion a line on this nice suggestion:

Page 20: "Lastly, to date, no study has investigated the impact of time, and the stability of emotional biases over time. Future studies could be designed to address such questions, which our current design, involving only 2 to 4 repetitions per condition, did not permit."

R2.10. Given that no power analysis was conducted, I am not sure that the sample sizes are appropriate. For example, in Experiment 2, there are only 34 mothers interacting with their own and 34 mothers interacting with another child. These sample sizes seem quite low for dyadic research questions. Sensitivity analyses could help to see what effect sizes could be detected with these sample sizes.

Response: The reviewer rightly pointed out that we did not conduct any power analyses prior to conducting the study.

However, we agree that given the computation of our new biases and our new set-up, it is adequate to conduct sensitivity analyses. Using G*power, we ran a post-hoc power analysis for the computed effect size of our main findings (Experiment 2 Mothers' group: EB in the 'own' group Vs EB in the 'other' group)). Using an effect size of Cohen's $d = 0.66$ (calculated using independent t-tests, ($t = 2.73$, $p < .05$, $95\%CI[0.17;1.17]$; $M_{EB_own} = 16.36$, $SE = 3.54$; $M_{EB_Other} = 4.77$, $SE = 3.53$) and $\alpha = 0.05$ yielded a power of 0.76 and a critical t of 1.99. Note although the obtained power is slightly lower than 0.80 which is considered as good sensitivity, our calculated t value is higher: $t = 2.73$. Moreover, sensitivity analysis with 0.80 power and α of 0.05 yielded to an effect size of 0.68, while ours is 0.66. Given these

results, we believe that a sample size of 34 mothers in each experimental group is leading to enough power.

We now added this information to the SM (and refer to it in the Methods section) as follow page 1:

“Using G*power, we ran a post-hoc power analysis for the computed effect size of our main findings (Experiment 2 Mothers’ group: EB in the ‘own’ group Vs EB in the ‘other’ group). Using an effect size of cohen’s $d = 0.66$ (calculated based on independent t-tests, ($t = 2.73$, $p < .05$, $95\%CI[0.17;1.17]$; $M_{EB_own} = 16.36$, $SE = 3.54$; $M_{EB_Other} = 4.77$, $SE = 3.53$) and $\alpha = 0.05$ yielded a power of 0.76 and a critical t of 1.99. Note although the obtained power is slightly lower than 0.80 which is considered as good sensitivity, our t value is higher: $t = 2.73$; ensuring that our sample size was coherent with the expected results.”

R2.11. The description of the analyses is not detailed enough to replicate the analyses. For example, it would be important to know how the variables were coded, whether REML or FIML was used, how predictors were centered in the multilevel model, and so forth.

Response: Along the reviewer comment, we now clarified and added these details throughout the methods section. Moreover, we have added the full models to SM for full clarity (Tables 9-12S; along R2.18 comment).

Page 25:

“To examine the hypothesised disparities between biases (AB vs EB), a multilevel modelling (MLM; REML) approach was employed using R Studio (lm4). The bias score (as computed from AB and EB scores) was employed as the continuous dependent variable. Bias type (Altercentric vs. Egocentric) was included as a binary fixed factor, while participant age as a continuous variable, was incorporated as a covariate. Participants were treated as a random effect in the models to account for individual variability. Notably, age was integrated into the models due to findings from previous research indicating age-related variations in emotional egocentric and altercentric biases. For Experiment 2, the statistical analysis closely paralleled that of Experiment 1, with the exception of the use of type of pairs (Own vs. Other), that was introduced as an additional fixed binary factor”.

R2.12. Given that the EB/AB rely on a difference score (which can be positive or negative) and the bodily EB/AB rely on absolute differences (which can only be positive), testing whether these scores are non-zero does not appear to be equally meaningful to me. It seems somewhat unlikely to me that the average absolute differences scores for bodily AB/EB (which are, e.g., based on ratings from 0-100 in Experiment 1b) approach zero (on average). Given that participants will have some variation in their answers and given that there will be measurement error, it seems likely that some differences will emerge, and as all differences will be positive, the average score is likely to be different from zero. To me, this does not necessarily imply that these differences are inherently meaningful.

Response: Following this point and the reviewer's previous comment on using absolute scores (R2.4), we changed the way we computed the new bias scores, so this point does not apply anymore.

R2.13. For the EB/AB scores: An average score of zero for AB as in Experiment 1b (looking at Figure 2b), seems to imply here that there is a similar amount of people with negative scores as there are with positive – what does that mean? How do the authors interpret this finding? To me, it seems to show that there is no AB.

Response: As rightly pointed out by the reviewer, an average score of zero for AB means that on average in the sample of Exp1b, we observe no significant AB using inferential statistics, and confirmed by Bayes Factor ($BF_{10}=0.205$).

As a side note, and in response to R3.4, we have also conducted analyses to compare samples of Exp1a and Exp1b, to explore further the potential differences that could explain the different results. As explained in response to R3.4, we have added the results of these analyses in SM as exploratory analyses, and one sentence about these comparisons in the result section.

R2.14. The authors used the term “validation” when showing that the biases in their new task are non-zero and conclude that the task was validated. However, I think a validation would have to be more conclusive than that (also given comments #12/13) and should also be more comprehensive. I would expect more information about validity and reliability.

Response: We thank the reviewer for this point. We agree that the term ‘validity’ typically implies a research design with different and more steps, so we no longer refer to our task development results as ‘validation’ but rather as “results that are in line with previous findings that found positive EB in young adults”, and we removed any instance of “validation” from the manuscript.

For example in the abstract: “To investigate these biases and control for sensory priors, we first conducted two experiments in dyads of adult strangers (total N = 75), using ~~to validate~~ a novel, bodily Emotional Egocentricity Task that enables simultaneous affective tactile stimulation within a dyad.”

R2.15. The authors write that they “test our hypothesis regarding the size of the biases (AB and EB) [...]” (p.7), but I was not sure which hypothesis this pertains to/ I did not find it in the text.

Response: We thank the reviewer for flagging this ambiguity in the text. We have now clarified this and made sure we explain which hypothesis we refer to throughout the manuscript.

In the introduction page 8-9, we now reformulated the hypotheses as follow:

“We tested these biases (congruency and sensory-controlled EB and AB) in dyads of strangers (Experiment 1) and of mother-child pairs (Experiment 2) using the bodily, Emotional Egocentricity Task. In Experiment 1, we expected that, within our bodily Emotional Egocentricity Task, adults would exhibit both a larger *classical* but also a larger *sensory-controlled* EB, compared to AB. This would confirm and extend previous research conducted on dyads of adult strangers. In Experiment 2, we expected mothers to exhibit greater *classical* and *sensory-controlled* EBs compared to ABs. Additionally, we hypothesised that mothers would display greater EBs towards their own child in comparison to an unfamiliar child, given the importance of their own emotional perspective in understanding and regulating their children. It is noteworthy that our study is designed to address the aforementioned hypotheses regarding mothers’ emotional EB and AB towards their children, an area that has received limited attention in the existing literature. In contrast, several studies have been dedicated to studying egocentricity in children. Therefore, we conducted exploratory analyses within the latter category, expecting the presence of greater biases in EB¹⁹ and AB³⁹ in children. This is also expected to vary depending on the dyad composition (own mother vs. other mother) and the age of the participants¹⁹.”

And, for example in the results section page 9, we removed the wording “size” that was confusing and changed it to “Moreover, to test our hypothesis regarding the difference between the biases (*classical* AB and EB) multilevel models (MLM) were used” - to reflect the hypothesis that we expected greater EB than AB.

R2.16. It is reported that “neurotypical adults” were assessed (p. 7). If I understand it correctly, the autism questionnaire (AQ) was administered. However, a score on the AQ should not be interpreted as determining or excluding neurodiversity (since it is a self-report measure and also does not account for all types of neurodiversity, e.g., ADHD). It is also not clear whether any individuals scored above a cut-off and were excluded.

Response: We thank the reviewer for this comment, as it allows us to clarify this point in the methods section. We recruited “neurotypical adults” based on “self exclusion”, as we made clear to participants before recruiting them that exclusion criteria were any present or past neurological or psychiatric history. However, we acknowledge that we have not tested this via clinical questionnaires. We now make this point clearer in the manuscript.

See page 20:

“For all experiments, exclusion criteria included any self-reported present or past neurological and psychiatric disorders.”

Moreover, we have not used the AQ10 or the IRI as diagnostic tools, but rather as to measure interindividual differences, and to explore any potential correlations with emotional biases. Note that no cut-offs were considered and no one was excluded on that basis.”

R.2.17. Overall, I was wondering whether outliers in the data were identified. Some of the data points in the figures seem extreme (e.g., Figure 3A, EB).

Response: In experiment 1b, and Experiment 2 we used the comparison of the ratings of the unisensory Pleasant vs Unpleasant experience (Feeling tactile only), with the assumption that those who rated the unpleasant condition being more pleasant than the pleasant condition could be identified as possible ‘outliers’ (see below, and in response to R1.5). However, after removing those ‘outliers’ from the sample did not change the observed effects, and as such we decided to include the whole sample. We now report the analyses without outliers in the SM.

Regarding the ‘extreme’ point in Figure 3A it is indeed 2.5SD above the group EB average, however as before, we repeated our analysis with and without this point and results remained the same.

We have now added a paragraph on ‘Outliers detection’ in the method section (page 27) and added the results without these outliers in SM (Tables 13-17S):

Page 25-26:

“Outliers detection. For all experiments, we employed two methods to identify outliers. First, we examined participants' average response to the ‘Tactile only’ baseline unisensory pleasant condition (“Tactile only pleasant”) compared to the ‘Tactile only’ baseline unisensory unpleasant condition (“Tactile only unpleasant”). Individuals who rated the unpleasant stimuli as more pleasant than the pleasant stimuli (e.g., rating the touch of a scourer higher on average than that of a cotton ball), were highlighted as potential outliers. This examination identified 6 potential outliers only within the mother groups in Experiment 2. No outliers were identified in Experiment 1b nor in the children group in Experiment 2. In Experiment 1a, where we lacked a unisensory baseline condition, we compared the ‘self pleasant congruent’ condition to the ‘self unpleasant congruent’ condition. This comparison yielded one potential outlier. Subsequently, we conducted our analyses both with and without these potential outliers. However, the results remained consistent, leading us to retain them in the sample to account for possible individual differences (see SM for results without outliers; Tables 13S, 15S).

Second, we also examined potential outliers, based on criteria involving score above or below 2.5SD from the group’s average ratings scores across the different conditions. These analyses revealed one potential outlier in Experiment 1b, three in the mother group, and two in the children group in Experiment 2. Once again, our analyses were performed with and without these outliers, and as the observed effects remained unchanged, we decided to retain them in the sample (see SM for results without outliers; Tables 14S, 16S, 17S).”

R.2.18. It was difficult for me to follow the results. I would prefer to see the full model, probably in a table (this may be a personal preference).

Response: We now reformulated the results section and we have added the full models to SM for full clarity (Tables 9S-12S).

R.2.19. I was not sure why the authors seem to consider only the correlations with caution given the limited sample size (p. 13), but not the other results.

Response: We thank the reviewer for flagging this up and we removed this sentence from our report, as this was confusing the reader. We were mentioning this “caution” regarding the correlations in the children sample, in which we had many missing cases from the questionnaires (given data loss for questionnaires, N=28); while the sample size for the other samples (in experiment 1 and in mothers) were sufficient to conduct correlations analyses.

R.2.20. Also, with regard to the correlation between IRI-EC and classical EB, two p values are reported, and it was not clear to me which p-value the authors consider appropriate (uncorrected or corrected), and it was also unclear to me how the p-value was corrected. Finally, it was unclear to me why they controlled for age (given that there are many other potential confounds that cannot be controlled for).

Response: We thank the reviewer for highlighting these points, and we now clarify in the text. First, we controlled for false discovery rate correction using the ‘BH’ method of Benjamini & Hochberg (1995), which we now also report in the methods section. To avoid confusion of using two different p-values as we did in the original submission, we now report the p-values before correction and only report if it “survived” FDR corrections. See page 25: “Correlations were corrected for false discovery rate (FDR) correction using the ‘BH’ method of Benjamini & Hochberg (1995)”

Second, we acknowledge that there are other variables that could interact with our effects (and that could not be controlled for, as discussed now in the limitation section), but one of the main variables that has been found in previous studies to influence emotional egocentric and altercentric biases is age (Riva et al. 2016), this is the reason why we decided to include age in our models.

R.2.21. The authors claim: “Thus, this novel finding suggests the self-other distinction has an adaptive and changing role in parenting and opens new avenues for research in the relationship between the self-other distinction, affective regulation and parental interoception.” (p. 15). It seems to me that the only finding suggesting adaptive value of EB is the correlation with the IRI-EC, which, however, turned non-significant when applying correction methods (see comment #20). I would not agree that the cited claim is warranted.

Response: We thank the reviewer for commenting on this and allowing us to explain what we mean by the use of ‘adaptive role’. We have now extended this interpretation and other potential ones in the discussion, as explained in response to R2.1, R13 and R3.8 (see page 17-18 of the discussion).

It is to note that the new calculations of the sensory-controlled biases resulted in a significant correlation between the sensory-controlled EB and the empathic concern subscale of the IRI

and survived FDR corrections (i.e. $r_{(63)} = .40$, $p = 0.002$). Moreover, along comment of reviewer 1 (R1.8), we conducted two separated correlations, one per mother group (those who paired with their own child and those who were paired with a stranger) and found that the positive correlation is only significant in the 'own group' (sEB: $r = .58$, $p < .001$).

R.2.22. Finally, coming from a different research tradition, it was difficult for me to think about bias without accuracy or "truth" (see, e.g., West & Kenny, 2011). I would think that, given that there are interindividual differences in how pleasant a touch can be perceived in general, people also differ in how pleasant they consider a "shared" or "aligned" experience (such as being touched with the same material). In a shared experience, two people's experience may be similar, but maybe still not the same. Furthermore, projecting one's own experience to another person could be considered egocentric even when this projection (or also termed assumed similarity) is mostly accurate. Maybe the authors could take this into consideration. See, e.g.: West, T. V., & Kenny, D. A. (2011). The truth and bias model of judgment. *Psychological Review*, 118(2), 357–378. <https://doi.org/10.1037/a0022936>

Response: We thank the reviewer for referring us to this model by West & Kenny; 2011. Indeed, in our studies we follow the term bias has been used by Silani and other researchers in the social mirroring field and less so in the tradition of perceptual metacognition and other fields that use concepts such as 'ground truth'. Given that we focus on subjective sensory and emotional judgements, the role of ground truth is indeed rather complicated in this context and bias does not typically mean 'inaccurate', or 'subjective', but rather it is used to characterise a difference in judgement due to specific sensory and social conditions. Using our design, we were able to control for at least two sources of this difference, i.e. for differences in unisensory versus socially incongruent experience and for differences in concurrent congruent vs incongruent experience; please see also above for detailed discussion). However, as indeed the reviewer highlights it, this is a different commitment to the notion of ground truth and accuracy considerations. It is beyond the scope of this paper to address these interesting phenomena and complex relations between truth and bias, however we now refer our readers to this paper and we discuss the above differences in the introduction.

Page 7-8:

"It is important to note that the term "bias" bears different meanings in different fields of study. The current work places itself within the social mirroring field, wherein our focus is directed towards subjective sensory and emotional judgements (but see³⁸ for the links between emotional EB and AB and relative "accuracy" in the perceptual metacognition field; as well as the bias model of judgement, which extends beyond the scope of our present design)."

Reviewer #3:

The authors have built and validated a new emotional egocentricity task and, with it, examined how the egocentric and altercentric biases (EB and AB) of mothers differ between judging how their own child and an unknown child feels.

They have first replicated previously reported EB and AB effects and showed the existence of such biases for bodily signals. Then they have shown that mothers have a higher EB for their own child than for the unknown child, both for the emotional and bodily biases.

The study is well-designed and their results may have important implications for the field or even clinical parental psychology in general. But such implications require more cautious interpretations.

I don't have any major concerns but a series of minor issues:

R3.1. The interpretation according to which the mothers' egocentricity bias for their children is actually adaptive is not convincing enough yet. I suggest to document further empirical and theoretical evidence in favour of this interpretation. If the authors have already made a thorough literature search then they should acknowledge it is a novel and speculative interpretation that remains to be confirmed. I personally think that relying on egocentric bodily states is insufficient to provide adequate care. Children are more fragile and have more needs than adults and thus representing their bigger needs is not helped at all by relying in egocentric needs. But of course, I agree that egocentric needs can remind us of our children's needs if we haven't represented their needs yet. In any case, I find it not intuitive that the authors would expect that the mothers would show higher EB for their own child.

Response: We thank the reviewer for this comment, which goes along with comments from Reviewer 1 (R1.3) and reviewer 2 (R2.1 and R2.21). We agree with the reviewer that this is merely one interpretation among many (such as social referencing and heuristic mode) and that future studies could explore further these alternative interpretations. We have now added such alternatives in our manuscript and discussed them as hypotheses for future studies, indicated by our findings, rather than the definite interpretation of our findings. See changes in the Discussion Page 17-18.

R3.2. Biases based on internal bodily signals must be better explained. I have a hard time to grasp it despite I am familiar with the EB and AB.

Response: We thank the reviewer, and have now detailed how we computed these new bodily social emotional biases. As reported in response to the two other reviewers (R1.2 and R2.3), we revised our terminology and have chosen to now call the "bodily-emotional" EB/AB as "sensory-controlled" EB/AB.

We now explain this more clearly in the introduction as follow:

Page 6-7:

“This adapted paradigm facilitated the computation of both classical emotional EB and AB (subsequently referred to as “classical bias”, EB and AB), as well as new sensory-controlled EB and AB (subsequently referred to as “sensory-controlled bias” sEB and sAB— see Figure 1B, and Methods for details). Classically, based on the study by Silani et al. (2013), emotional egocentric and altercentric biases are computed as the difference in pleasantness between incongruent and congruent trials. A supplementary and beneficial approach to understanding these biases is to gauge the disparity between the pleasantness provided during incongruent trials (which involve multisensory and socially concurrent stimulation) and the pleasantness ratings given during unisensory trials (i.e. trials involving either seen or felt stimulation, devoid of concurrent social stimuli). To compute an emotional EB using unisensory baselines, we can subtract the pleasantness rating of the visual unisensory condition (vision prior). This subtraction illuminates the distinct and unique contribution of the self feeling to the EB bias. Similarly, to compute an AB grounded in unisensory baselines, we can subtract the pleasantness rating of the tactile unisensory condition (feel prior). This computation unveils the unique influence of observing another person’s touch on the AB. Having these added, complementary control measures becomes especially valuable when testing pairs of participants with diverse roles (e.g. parent, child), age disparities and familiarity variation (e.g. own-child or. other unfamiliar-child). Such factors can potentially yield differences in unisensory, multisensory, and multiagent attention and perception.”

R3.3. The Figure 1 must be improved: the stimuli pictures are pixelated and too small. It would be best to see a photograph of the top view (or even better: participant centred view) of the setup.

Response: We apologise for the poor quality of the figure. We have now included a high-resolution figure. Moreover, as mentioned in response to R1.4, we have added a picture of the set-up in Figure 1C, so the reader can better see that participants could not see each other's face (curtain), could only see the forearm of the other participant but could not see their own forearm.

A. Design Main Task

		What Participant SEE = OTHER (O)			
		Pleasant		Unpleasant	
What Participant FEEL = SELF (S)	Pleasant	S O	S O	S O	S O
	Unpleasant	S O	S O	S O	S O

C. Set-up Experiment 1

B. Procedure

1. Prior Estimates - Unisensory

2. Self and Other Pleasantness Estimates

D. Set-up Experiment 2

Figure 1. (...) C. **Experimental Set-up** for Experiment 1a and 1b. D. with a pictorial representation of the set up with two participants separated by a curtain. (..)

R3.4. In study 1 second sample, they did not find a significant AB, which I agree can be due to interindividual differences but I recommend the authors to test for demographic and questionnaires data differences between the two samples. E.g., higher % of women in sample 1.

Response: We thank the reviewer for their suggestion and for allowing us to test this. To explore further the variables that could explain the different results found in Exp1a and Exp1b (i.e. significant and positive AB in Exp 1a, and no significant AB in Exp1b), we conducted several comparisons on demographics (age and gender) and questionnaires data (IRI and AQ10). We have added the results of these analyses in SM as exploratory analyses, and one sentence about these comparisons in the result section.

Page 10 in the results section:

“The difference observed between the two samples might be due to the impact of interindividual differences (i.e., difference on average age of each sample and scores on the empathy questionnaire IRI, with participants in Exp1a being on average older and scoring higher on the IRI than participants in Exp1b, see SM).”

Page 1 of SM:

“Comparisons between Experiment 1a and Experiment 1b samples:

To explore further the variables that could explain the different results found in Exp1a and Exp1b (i.e. significant and positive AB in Exp 1a, and no significant AB in Exp1b), we conducted several independent samples tests on demographics (age and gender) and questionnaires data (IRI and AQ10). Note that for age and AQ10, Mann-Whitney U tests were conducted as the assumption of normality was violated. A threshold of $p < 0.01$ was considered as significant, to correct for multiple comparisons. A significant difference of age between the two samples was found ($U=1122$, $p < 0.001$, $r_B=0.662$), with the sample in Exp 1a being older than sample in Exp 1b (Exp 1a: $M_{age}=34.22$, $SD=10.59$; Exp 1b: $M_{age}=24.93$, $SD=7.79$). Given results by previous studies who found an effect of age on emotional egocentric and altercentric biases (Riva et al., 2016), this could explain some of the difference observed between Experiment 1a and 1b. Moreover, significant differences were found on the IRI subscales with the sample in Exp1a scoring higher on the fantasy scale (Exp1a: $M=17.69$, $SD=5.18$; Exp1b: $M=13.79$, $SD=3.59$; $t(69)=3.507$, $p < 0.001$, $d=.847$), higher on the empathic concern (Exp1a: $M=19.88$, $SD=4.66$; Exp1b: $M=14.37$, $SD=1.76$; $t(69)=6.049$, $p < 0.001$, $d=1.46$), higher on the perspective-taking subscale (Exp1a: $M=18.79$, $SD=4.34$; Exp1b: $M=15.31$, $SD=3.42$; $t(69)=3.603$, $p < 0.001$, $d=.870$) and a trend to score lower on the personal distress subscale (Exp1a: $M=11.55$, $SD=4.16$; Exp1b: $M=13.65$, $SD=3.16$; $t(69)=-2.303$, $p=0.024$, $d=-.56$). However, no significant was found on the AQ10 questionnaire ($U=497$, $p=.186$, $r_B=-.184$). Note that no difference in gender was found (64% female in Exp1a vs 60% in Exp1b)."

R3.5. In study 2, for mothers emotional biases, please report as well whether the AB and EB were significantly different from zero.

Response: We have added this to the results section, page 11:

"First, similar to Experiment 1, we run one sample t-tests and showed that both EB and AB were different from zero (EB: $t_{(67)}=4.89$, $p < 0.001$, Cohen's $d=0.58$, $BF_{10}=2.17e+3$; AB: $t_{(67)}=2.56$, $p=0.01$, Cohen's $d=0.31$, $BF_{10}=2.74$). " and page 14: "First, similar to Experiment 1, we ran one sample t-tests and showed that only sensory-controlled EB but not sensory-controlled AB ($sEB: t_{(66)}=2.96$, $p=0.004$, Cohen's $d=0.36$, $BF_{10}=7.23$; $sAB: t_{(66)}=-1.56$, $p=0.12$, Cohen's $d=-0.19$, $BF_{10}=0.42$) was different from zero (Figure 3.B)."

R3.6. In study 2, for mother bodily biases, I don't follow why they didn't test the same interaction pair x type of bias. Instead they tested effect of pair separately for EB and AB.

Response: Following this comment and comments from Reviewer 2 (R2.4), we have now used the same analysis as we did for the classical emotional egocentric and altercentric biases. New results are reported page 13-14 as follow:

"First, similar to Experiment 1, we ran one sample t-tests and showed that only sensory-controlled EB but not sensory-controlled AB ($sEB: t_{(66)}=2.96$, $p=0.004$, Cohen's $d=0.36$, $BF_{10}=7.23$; $sAB: t_{(66)}=-1.56$, $p=0.12$, Cohen's $d=-0.19$, $BF_{10}=0.42$) was different from zero (Figure 3.B). Second, MLM was used in order to test our hypothesis regarding the impact of the dyadic pair (Own vs. Other) on the sensory-controlled biases. Bias scores were

served as a continuous dependent variable in a MLM, with bias type (sAB vs. sEB), mother's age, child's age and pairs (Own vs. Other) as fixed factors and participants as a random effect. Similar to Experiment 1, we found a main effect of sensory-controlled bias type ($b=17.84$, $SE=3.51$, $p<0.001$; $95\%CI [10.96;24.72]$). Crucially, a two-way interaction between the sensory-controlled bias type (sEB vs sAB) and being paired with own vs. other child emerged ($b=-13.88$, $SE=4.96$, $p=.005$, $95\%CI[-23.61;-4.16]$; Figure 3.A). Probing this interaction using planned comparisons, revealed that this effect was driven from a larger sEB in the group that was paired with their own child as compared to the group that was paired with a stranger (i.e., other child) ($t_{(65)}=2.30$, $p=.02$, $95\%CI[1.51;20.30]$; $M_{sEB_own}=13.42$, $SE=4.00$; $M_{sEB_Other}=2.51$, $SE=3.48$; $BF_{10}=1.49$). Similar to the classical AB, there was no significant difference in sensory-controlled AB between the two groups ($t_{(65)}=-0.69$, $p=.49$, $95\%CI[-12.88;6.21]$; $M_{sAB_own} = -4.42$, $SE=3.32$; $M_{sAB_Other}=-1.09$, $SE=3.42$; $BF_{10}=0.32$). In addition, the difference between sensory-controlled EB and sensory-controlled AB was significant only in the group that was paired with their own child (own child: $t_{(33)}=-5.12$, $p<.001$, $95\%CI [-24.79;-10.88]$, $BF_{10}=930.36$; other child $t_{(32)}=1.11$, $p=.26$, $95\%CI [-11.01;3.10]$ $BF_{10}=0.34$..”

R3.7. Line 265: “empathic concern” not “empathy concern”.

Response: We thank you for spotting this mistake, we have now corrected it.

R3.8. Another interpretation regarding the authors' main finding is that mothers were worried to perform better with an unknown child than a known child. There is plenty of literature in social psychology showing that less uncertainty, more familiarity, more similarity, being in “comfort zone” lead us to adopt a more heuristic mode of functioning which is to rely on egocentric information. In contrast, an unknown child is more likely to activate a more controlled and effortful consideration of what the child thinks.

Response: Thank you for referring us to this possible interpretation of our findings. Indeed the reliance on egocentric information or ‘mental shortcuts’ could increase when individuals are familiar with a situation (in this case being with your own child). We added this possibility in the discussion along with other possible interpretations (see comment R1.3 and R2.21, and R3.1).

Page 18: “Alternatively, our results could pertain to the distinction between familiar situations (i.e., being paired with their own child) and unfamiliar or novel situations (i.e., being paired with an unfamiliar child). In familiar instances, the parent is accustomed to the scenario of gauging or guessing their own feelings, potentially promoting the utilisation of egocentric information, which in turn results in a larger egocentricity bias. In contrast, in unfamiliar conditions, participants might engage in more deliberate, controlled thinking about the child's perspective, leading to reduced reliance on ‘perspective shortcuts’ and a diminished egocentric bias (though see 44 for the application of a heuristic mode in uncertain situations).

R3.9. Altogether, given the message the authors want to convey about mothers and their child I would recommend a more cautious tone. The authors should remind the readers that the effects are statistically significant at the group level but maybe not so significant for everyday life given the effect sizes and the high interindividual heterogeneity.

Response: We thank the reviewer for this comment and we adjusted our interpretations and toned down our claims throughout the manuscript. We also highlighted some individual differences and added this point to the discussion as follow:

Page 18:

“It is crucial to acknowledge the presence of interindividual variability observed in both the EB and AB scores. While, on average, mothers exhibited a significant and large EB, suggesting a heightened influence of their own feelings when evaluating their own child, this does not hold true for every mother. As illustrated in Figure 3, it becomes evident that some mothers even displayed a negative EB, suggesting that they were not influenced by their own feelings when their child was experiencing an incongruent sensation. These differences could potentially arise from individual factors such as heightened concerns for others, as suggested by the present study, or attachment styles that warrant further investigation in future research.”

20th Sep 23

Dear Dr. Kirsch,

Thank you for submitting your revised manuscript titled "Mother knows best: Mothers are more egocentric towards their own child's bodily feelings" to Communications Psychology. We have given the paper our careful consideration and are interested in sending it back to review. However, due to certain shortcomings we are concerned that sending the current manuscript out to review could lead to unnecessary delays and quite possibly an undesirable outcome of the review process.

In particular, we would like to see full statistical reporting in line with our guidelines: <https://www.nature.com/commspsychol/submit/submission-guidelines#statistical-guidelines>.

For each model, please specify how the variables entered into the model are coded including information on any centering or standardization. For example consider the model reported in Table 11S, please add information on how age and child age are coded—is it age in years? Was age centered? For bias and pair, what type of binary coding is used and which code is assigned to each predictor (e.g., 0 = own, 1 = other or another coding scheme). For every model, please either add a line to the table indicating the estimates for the intercept or specify that no intercept was modeled.

In cases where an a priori power analysis was not conducted, we recommend conducting a sensitivity analysis (as opposed to a post-hoc power analysis) to find the smallest effect that you could have detected with high probability given the sample size used.

We would highly encourage you to deposit and make accessible your code.

We would therefore like to invite you to revise your manuscript to address these concerns before we make a final determination on whether to send your manuscript for external review.

We shall hope to receive your revised version as soon as you are able to complete the suggested revisions. If something similar is published in the interim we will have to consider the impact it has on the novelty of a revised manuscript.

If you anticipate a delay of more than four weeks, please let us know. Should your manuscript be substantially delayed without notifying us in advance and your article is eventually published, the received date may be that of the revised, not the original, version.

If you are not interested in submitting a suitably revised manuscript in the future please let me know immediately so we can close your file. If you have any questions, please contact me.

Please use the link below when you are prepared to resubmit.
[link redacted]

Thank you for your interest in Communications Psychology.

Best regards,
Jennifer Bellingtier

Jennifer Bellingtier, PhD
Senior Editor
Communications Psychology

Rebuttal Letter - COMMSPSYCHOL-22-0043A

We would like to thank the three anonymous reviewers for providing helpful critical feedback on our manuscript. We believe that we have answered all the Reviewers' concerns and that it greatly improved the manuscript with respect to its original version (please see below a point by point response, and main changes are written in blue in the revised manuscript). Specifically, we:

1. Explained in more details our task and its relevance to investigate our hypotheses, and why we believe it is a bodily task. Following the reviewers' comments, we have revised our terminology and have chosen to now call the "bodily-emotional" EB/AB as "*sensory-controlled*" EB/AB.
2. As suggested, we have added relevant developmental theoretical interpretations to our findings, such as social referencing and epistemic trust, and related limitations to our discussion.
3. We clarified the description of our experimental design and added further details about the set-up and procedure in the introduction and in the methods (including justification of our sample size).
4. Following the reviewers comments we also explored the possibility of outliers and added an 'outliers section' in our results, as well as supplementary results.
5. Last, we revised our discussion to remain closer to the data, including a discussion of the limitations of the study.

Reviewer #1

The hypothesis is interesting and makes sense; the studies do a fine job of testing it.

R1.1. The relationship between this task and the 'classical' one as used in Silani et al (2013) needs to be explained more clearly. Their task also included tactile sensations, so in what sense is this paradigm more bodily? In what sense is it an advance?

Response: We thank the reviewer for giving us the opportunity to explain better the novelty of our task, and why we believe it is a more bodily task. While we agree with the reviewer that both tasks involve felt tactile sensations, there are two important differences: (1) the nature of the seen, vicarious touch is different between the tasks, with one being embodied and the other not; (2) the tactile stimulation parameters differ between the paradigms. Both of these differences have important and well-established implications in sensory neuroscience, as we now explain in the manuscript and below.

(1) First as regards, the 'seen touch'. There are two key reasons why our task can be considered a bodily task. We now explain each of these reasons in the text as follows:
p. 5-6:

"Dyads of participants were tested in proximal space, using a set-up that allows tactile stimulation of participant's own unseen forearm at the same time as they are watching

another participant's forearm being touched. This approach moves beyond the utilisation of a disembodied stimulus, as seen in previous tasks, to include a vicarious touch stimulus (see also below and Figure 1). Consequently, this setup facilitates a concurrent experience of self, through unisensory touch (felt touch) and other, through unisensory observed touch (vicarious seen touch). As depicted in Figure 1, participants were asked to judge how pleasant was the touch they felt (self) or saw (other), without knowing in advance which question they would need to answer. This required their attention to be focused on both self and other experiences. This simultaneous sensory experience can either be congruent or incongruent in terms of pleasantness (pleasant or unpleasant; see the design and set-up in Figure 1).

A large tradition of sensory and multisensory cognitive and neuroscientific research has studied vicarious (or mirror) sensory phenomena. Notably, studies within this domain have highlighted the unique effects of seeing body parts being stimulated, in contrast to other non-body stimuli, as well as many other embodiment effects on vicarious touch (e.g.³¹⁻³³). For example, neuroimaging studies have identified areas within the occipito-temporal cortex, such as the extrastriate body area³⁴, and fusiform body area³⁵ responding selectively to seen bodies and body parts, and a corresponding topographic map of viewed body parts throughout occipitotemporal cortex³⁶. This specificity of the human body (both our own and that of others) as a visual stimulus and a representation, particularly during vicarious responding, is one of the two reasons we named this task the 'bodily Emotional Egocentricity Task'. This task extends beyond 'imagining' the other person being touched with a stimulus while one feels touch on their own body, it also encompasses the representation of the other person's body and the seen touch itself during the vicarious touch experience. The second, related reason pertains to the capability of our task to provide subject-specific, important control (non-concurrent) measures for both the felt and the seen unisensory perception of touch. Various studies on vicarious, or mirror perception and visual enhancement of touch have shown that there are both intersubjective and intermodal effects on concurrent perception of felt and seen touch. While traditional emotional egocentricity tasks aim to mitigate some of these effects by employing emotionally 'congruent' conditions as a baseline, these tasks still remain within the context of concurrent, multimodal conditions. By contrast, our task gives the opportunity to have this control, as well as unisensory measures for each modality, without the concurrent stimulation. Indeed, in addition to the congruent and incongruent conditions tested during the main task, we also tested unisensory (non-concurrent) perception for both the felt touch on the self and the seen touch on the other. These control conditions serve the purpose of disentangling sensory from the multisensory effects inherent to the task. Importantly, none of these conditions are susceptible to the top-down effects that object or animal stimuli could potentially exert on participants, as often encountered in the original egocentricity tasks that use pictorial stimuli for the seen touch and to characterise the felt touch."

(2) Second, as regard to the 'tactile stimulation parameters', we now explain the key difference and advance of our task as follow:

Page 7:

"Furthermore, our task not only allows for participants to see the touch the other person is receiving on their body with a given material, but it also allows us to match the specific stimulation parameters of the tactile and the vicarious stimuli, i.e. the self and other

stimuli. Participants are in the same room, and they can see and feel specific tactile stimuli that could be either congruent or incongruent in pleasantness, applied in a specific synchronous velocity of 3 cm/s (as done in Filippetti et al., 2019 ; see also Methods). This sensory matching is not possible in the original task¹⁶, as participants are only presented with pictorial stimuli of ‘what’ touches the other person (not ‘how’ the other person is actually touched), while also seeing similar pictorial stimuli representing what is supposedly touching their body, as they are being touched with different materials out of sight (rather than actual spiders, swans etc). In light of the aforementioned distinctions, our task constitutes a more embodied rendition of an egocentricity task and allows us to test specifically the existence of emotional EB and AB in a bodily context.”

R1.2. Please explain more clearly in the main text what the advantages are of calculating the ‘bodily-emotional’ EB/AB as opposed to the ‘classical’ one. And consider using a different label for the former, as it does not appear to be any more or less bodily than the classical one.

Response: We thank the reviewer for this point that allows us to clarify the value of our added control conditions and the more bodily nature of our tasks and biases, as better explained now in the introduction (see response to point R1.1 above). It also allows us to revise our terminology and we chose to now call the “bodily-emotional” EB/AB as “sensory-controlled” EB/AB.

Classically, based on Silani et al. 2013 study, emotional egocentric and altercentric biases are computed as the difference in pleasantness between incongruent and congruent trials. These biases can account for the differential effects of valence between such conditions but they cannot distinguish between the effects of sensory stimulation (i.e. being exposed to sensory stimuli in different modalities) and socially congruent stimulation (being exposed to both self and other stimuli), as both of these conditions share these characteristics. For example, some people may find visual stimuli far more salient than tactile stimuli. So, the difference between congruent and incongruent trials may be explained by the fact that they focused more on the seen aspects of the experience rather than the felt ones. Quantitatively, the effect may be the same with other people that focused more on the other rather than the self, but the reasons for this bias would be different. Accordingly, a further, useful, complementary way to look at emotional egocentric and altercentric biases is to compute the discrepancy between the pleasantness rating we give in incongruent (multisensory and socially concurrent stimulation) trials and pleasantness ratings given during unisensory trials (i.e. seen or felt stimulation, without the socially concurrent stimuli). In order to compute an emotional egocentric bias based on unisensory baselines, we can subtract the pleasantness for the visual unisensory condition (vision prior), to get the unique contribution of the self-feeling on the EB bias. Similarly, to compute an altercentric bias based on unisensory baselines, we can subtract the pleasantness for the tactile unisensory condition (feel prior), to get the unique contribution of the vision of the other on the AB bias. For these reasons we call the new biases the “sensory-controlled biases”. Furthermore, having this added, complementary control

measures this has even greater added value when testing pairs of participants that may have different roles (e.g. parent, child), age differences and familiarity differences (e.g. own-other child), with all of which possibly causing differences in unisensory and multisensory, and multiagent attention and perception.

We now explain this more clearly in the introduction as follow:

Page 6-7:

“This adapted paradigm facilitated the computation of both classical emotional EB and AB (subsequently referred to as “classical bias”, EB and AB), as well as new sensory-controlled EB and AB (subsequently referred to as “sensory-controlled bias”sEB and sAB– see Figure 1B, and Methods for details). Classically, based on the study by Silani et al. (2013), emotional egocentric and altercentric biases are computed as the difference in pleasantness between incongruent and congruent trials. A supplementary and beneficial approach to understanding these biases is to gauge the disparity between the pleasantness provided during incongruent trials (which involve multisensory and socially concurrent stimulation) and the pleasantness ratings given during unisensory trials (i.e. trials involving either seen or felt stimulation, devoid of concurrent social stimuli). To compute an emotional EB using unisensory baselines, we can subtract the pleasantness rating of the visual unisensory condition (vision prior). This subtraction illuminates the distinct and unique contribution of the self feeling to the EB bias. Similarly, to compute an AB grounded in unisensory baselines, we can subtract the pleasantness rating of the tactile unisensory condition (feel prior). This computation unveils the unique influence of observing another person’s touch on the AB. Having these added, complementary control measures becomes especially valuable when testing pairs of participants with diverse roles (e.g. parent, child), age disparities and familiarity variation (e.g. own-child or. other unfamiliar-child). Such factors can potentially yield differences in unisensory, multisensory, and multiagent attention and perception.”

R1.3. A stronger EB towards one’s children could also be functional in relation to social referencing, where children do not have a definite emotion or interpretation of the valence of a situation or a stimulus, and use the adult as a reference point. These are cases where the adult in effect shows the child what to feel; it would be counterproductive for the adult to try to identify the child’s emotion here. It could enrich the paper to link the findings with this research as well.

Response: We thank the reviewer for suggesting this intriguing and relevant interpretation to our finding. We have added this and another related social interpretation on the possible use of a more heuristic mode of functioning as a possible explanation of the observed larger egocentric bias in the mother ‘own’ group (along a suggestion by reviewer 3, see R3.8).

We now explain it in more detail in the discussion (integrated with the suggestions from R2.1. R3.1 and R3.8):

Discussion Page 17-18:

“..... Furthermore, our results are also in line with the “social referencing theory” (Klennert et al., 1986) according to which the greater egocentricity bias of mothers towards their

own child could also be related to the parents' social role as a reference point for guiding the child's response. Children often seek their caregiver's responses (e.g., facial cues) to guide their own responses, with caregivers acting as significant figures from whom children gather informational cues. In the context of our findings, this larger egocentricity bias could signify that parents focus more on their own affective responses to communicate and guide their child, rather than solely attending to their child's reaction. This is also consistent with studies suggesting that the parent's role as source of information is intertwined with their adaptive or pedagogic epistemic role within a dynamic environment (Csibra & Gergely, 2009). Consequently, the greater egocentricity bias within the 'own group' may highlight the parent's function as a reliable source of information transmitted to their child, thus facilitating social learning (...)"

R1.4. The description of the procedure and setup is not entirely clear to me. It looks in Fig 1b like participants should be able to see their own hand just as well as the other participant's hand, but I think I understand the text to indicate that they have only tactile information about their own hand and only visual information about the other's hand. How does that work?

Response: We apologise if our original description of this set-up was not clear. We have now added further details about the set-up and procedure in the introduction and in the methods. Moreover, we have added a picture of the set-up in Figure 1C, so the reader can better see that participants could not see each other's face (curtain), could only see the forearm of the other participant but could not see their own forearm based on specially constructed boxes with different opaque and transparent sides.

We have now added more details in the methods about the set-up as follow:

Page 21:

"Two custom-made boxes were designed specifically to allow a direct view of the other participant, while hiding the self. This configuration allowed the experimenter to simultaneously see and be able to touch both participants synchronously (dimensions: 40*17*20cm). The boxes were positioned in a mirrored manner, so that they were respectively open on the right and left sides, granting each participant the vision of the other participant's forearm exclusively. The upper part of the box was made opaque to block the view of their own forearm. Participants were seated next to each other, separated by a curtain to prevent any visual contact with each other. They were touched on their respective right and left forearms (see Figure 1C). The distance between the respective right and left index fingers of the two participants were kept at a constant distance of 30 cm (participants rested their index fingers on a tape affixed inside the box)."

A. Design Main Task

		What Participant SEE = OTHER (O)			
		Pleasant		Unpleasant	
What Participant FEEL = SELF (S)	Pleasant	S O	S O	S O	S O
	Unpleasant	S O	S O	S O	S O

C. Set-up Experiment 1

B. Procedure

1. Prior Estimates - Unisensory

2. Self and Other Pleasantness Estimates

D. Set-up Experiment 2

Figure 1. (...) C. *Experimental Set-up* for Experiment 1a and 1b. D. with a pictorial representation of the set up with two participants separated by a curtain. (..)

R1.5. I do not understand how the stimulus validation worked. Some people might find velcro pleasant for example; was it just assumed that some stimuli are unpleasant or was this determined for each participant for each stimulus?

Response: We thank the reviewer for raising this issue and allowing us to clarify this validation phase in more details. The choice of the fabrics was based on pilot data from a different sample (as used also in Filippetti et al. 2019), whereby the pleasantness of 12 materials of different degrees of pleasantness was rated in visual only, tactile only, and visuo-tactile conditions. Based on these data, we chose the two most pleasant and two least pleasant fabrics with similar visual appearance (i.e. size and colour), as rated by 12 participants. Therefore, our choices of pleasant and unpleasant fabrics were derived from pre-designated stimuli terms based on central hypotheses and piloting performed prior to our main experiments.

Moreover, our sensory baselines (felt and seen) allowed us to check for the individual preferences, and ensure that participants do not prefer for example the scourer to the cotton ball. We have now elaborated on these criteria in the outliers sections (see below). Note that none of the participants in Experiment 1b and Experiment 2 (children group) were excluded based on that criteria. In the mothers groups of Exp 2, 6 out of 67 could be excluded on this criteria (i.e., rated the pleasant stimuli as less pleasant than the unpleasant one). However,

given that analysis with and without these 'outliers' remained the same we kept these participants in (see supplementary for our main findings without these 6 possible outliers). Moreover, please note that previous studies did not exclude participants based on this criterion as it does not allow to capture the individual differences between participants. This information is now included in the manuscript as follow:

Page 21: *Fabrics piloting*:

"The selection of fabrics was informed by the same pilot data used in the study by Filippetti et al. (2019). In this pilot phase, the pleasantness of 12 materials of different degrees of pleasantness was rated under visual-only, tactile-only, and visuo-tactile conditions by an additional group of 12 participants. Based on these data, we opted for the two most pleasant and two least pleasant fabrics, while ensuring similar visual attributes (i.e. size and colour). As such, our choices of fabrics perceived as pleasant or unpleasant were derived from pre-established stimuli, built on core hypotheses and piloting conducted prior to our primary experiments."

Page 25-26:

"Outliers detection. For all experiments, we employed two methods to identify outliers. First, we examined participants' average response to the 'Tactile only' baseline unisensory pleasant condition ("Tactile only pleasant") compared to the 'Tactile only' baseline unisensory unpleasant condition ("Tactile only unpleasant"). Individuals who rated the unpleasant stimuli as more pleasant than the pleasant stimuli (e.g., rating the touch of a scourer higher on average than that of a cotton ball), were highlighted as potential outliers. This examination identified 6 potential outliers only within the mother groups in Experiment 2. No outliers were identified in Experiment 1b nor in the children group in Experiment 2. In Experiment 1a, where we lacked a unisensory baseline condition, we compared the 'self pleasant congruent' condition to the 'self unpleasant congruent' condition. This comparison yielded one potential outlier. Subsequently, we conducted our analyses both with and without these potential outliers. However, the results remained consistent, leading us to retain them in the sample to account for possible individual differences (see SM for results without outliers; Tables 13S, 15S).

Second, we also examined potential outliers, based on criteria involving score above or below 2.5 SD from the group's average ratings scores across the different conditions. These analyses revealed one potential outlier in Experiment 1b, three in the mother group, and two in the children group in Experiment 2. Once again, our analyses were performed with and without these outliers, and as the observed effects remained unchanged, we decided to retain them in the sample (see SM for results without outliers; Tables 14S, 16S, 17S)."

R1.6. Image quality: The images of the stimuli in Fig 1 (a and b) are not of sufficient quality for me to see them properly.

Response: We have now uploaded a higher resolution figure.

R1.7. It is indicated in the methods that the research was carried out in accordance with the Declaration of Helsinki, but this is not accurate given that the experiments were not pre-registered (See paragraph 35 here:

<https://www.wma.net/policies-post/wma-declaration-of-helsinki-ethical-principles-for-medical-research-involving-human-subjects/>)

Response: We thank the reviewer for pointing this out, as we admit we had not realised this clause in the declaration. We have now added this info in the methods section, as follows:

Page 20:

“All three experiments were conducted in accordance with the Declaration of Helsinki (except for pre-registration) and were approved by the Ethics Committee of the Research Department of Clinical, Educational and Health Psychology, University College London”.

R1.8. p.13: “Partial correlation analysis yielded positive correlations between IRI empathic concern subscale and the classical emotional EB ($r(63) = .33$, $p = .008$, $pFDRcorrected = .053$), suggesting that those who show higher EB also report more feelings of sympathy and concern for unfortunate others.” Theoretically, this should be surprising and demands explanation. It conflicts for example with work by Peter Hobson (Hobson, R. P. (1999). Beyond cognition: A theory of autism. Perspectives on the Nature of Autism, 253-281.)

The authors do give a reason for expecting mothers to exhibit a stronger EB towards their own children than towards unfamiliar children, but not for this more general statement.

Response: We thank the reviewer for this comment, and giving us the opportunity to discuss this result further. Indeed, this result could at first be surprising, suggesting that people with higher empathic concern exhibit higher egocentric biases. But several considerations need to be taken into account.

First and most importantly, these results are from the mother-child experiment, which could bias the results due to the context. The IRI-empathic concern subscale is classically a measure for sympathy and concern for unfortunate others. So indeed, at first it could seem in contradiction with an egocentric bias; as typical, greater EB can be seen as less empathy or less ‘other-oriented’. Noticeably, in adult-adult dyads in Experiment 1, no correlation reached significance between IRI subscales and egocentric biases. The only significant correlation was found between the AB and the IRI fantasy scale, suggesting that those who show higher altercentric bias also report higher tendency to transpose themselves imaginatively into the feelings and actions of fictitious characters (see SM).

Second, in the present study we hypothesised that a parent's bodily egocentricity towards their child could constitute an adaptive bias and, in that case, egocentric biases may not necessarily signify failures of empathic understanding, but serve anticipatory regulation or a reference point.

Moreover, in the current version, based on this comment by the reviewer, we explored further our finding on the positive correlation between IRI empathic concern subscale and EB

and conducted two separated correlations (those who paired with their own child and those who were paired with a stranger), and found that the positive correlation is only significant in the 'own group' with a trend in the "other" group. As such, our results suggest that the more sympathy and concerns for others a parent has, the larger their egocentricity bias for their own child is (and not in general). Relatedly, according to Lamm et al., 2019, "empathy has to be distinguished from sympathy and compassion (...) sympathy and compassion do not only involve affect sharing or "feeling as" the other, but rather a "feeling for" and thus of concern for the other", which could be measured by the IRI-empathic concern and could be consistent with higher EB in the 'own' mother group.

We thus removed the original, general statement from the results section and added this post hoc analysis to the results section.

Page 15:

"To further explore these correlations we explored the two groups separately (those who paired with their own child and those who were paired with a stranger), and found that the positive correlation with IRI empathic concern is only significant in the 'own group' (classical EB: $r_{(32)}=.43$, $p=.02$, $95\%CI[0.06;0.70]$ and sensory-controlled EB: $r_{(31)}=.58$, $p<.001$, $95\%CI[0.24;0.81]$), with a trend in the "other" group (classical EB: $r_{(31)}=.32$, $p=.09$, $95\%CI[-0.09;0.70]$) and sensory-controlled EB: $r_{(30)}=.33$, $p=.08$, $95\%CI[-0.06;0.65]$). Similarly, the negative correlation between sensory-controlled EB and personal distress was only significant in the group that was paired with their own child ($r_{(31)}=-.43$, $p=.02$, $95\%CI[-0.06;0.71]$), and in the other group showing only a trend ($r_{(30)}=-.35$, $p=.06$, $95\%CI[-0.73;0.12]$)."

R1.9. p.14: "we found evidence supporting our main hypothesis that mothers show greater emotional EB towards their own child than an unfamiliar child. Importantly, this was the case for both classical emotional EB and the sensory-controlled, bodily EB measure, suggesting also a bodily emotional egocentric effect." It is not clear to me what inference the authors are suggesting we make here.

Response: We agree with the reviewer that the end of this sentence was unclear, and we have now removed the part "suggesting also a bodily emotional egocentric effect". We now extend the conclusions in the discussion as follow:

Page 16:

"By employing the bodily emotional egocentricity task, we uncovered evidence that supports our central hypothesis - specifically, that mothers exhibit a more pronounced emotional EB towards their own child in comparison to an unfamiliar child. Importantly, this observation held true for both classical EB and sensory-controlled EB measurements. This suggests that in situations where mothers witness their own child undergoing tactile sensations different from their own, they tend to project their own sensations onto their child. This bias cannot be explained by any relative differences in valence between self and the child during simultaneous stimulation (original bias calculations). Nor can it be explained by any biases in how the mothers perceive such stimuli on their own bodies while experiencing them in isolation (novel bias calculations employing unisensory controls)."

R1.10. Limitations: It would have been ideal to use familiar other children rather than unfamiliar other children as a control. As it is, we can't be sure whether the key difference is that the mothers' own children were their own or that they were more familiar to them.

Response: We agree with the reviewer that our effects may be related to familiarity, but as we explain below this is both central to our hypothesis and to a lesser degree a possible confound that could be controlled by testing other familiar children (friends). We acknowledge this limitation and its complexity in our manuscript, as follows:

Page 20:

“Thirdly, in our design, we compared being paired with one’s “own” child vs an unfamiliar child, which did not directly control for the influence of familiarity. However, we acknowledge that the role of familiarity in this study is not solely a ‘confounding’ factor; it is also a hypothesised main effect. Specifically, we do believe that parents might exhibit distinct biases towards their own child due to familiarity with them, and with their habitual needs and their need for parental regulation. While there are other dimensions of familiarity that can act as confounds, such as perceptual or attentional familiarity, ease during testing, and lower-level perceptual factors, these could be partially controlled by testing other familiar children in future studies. Nevertheless, we note that this approach could introduce new confounds, given that family and friendship roles and levels of familiarity are not identical beyond superficial levels. Importantly, our unisensory baselines do control for some of these lower-level perceptual familiarity aspects, as they enable the calculation of biases based on unisensory measures of both seen and felt touch, unique to each tested child.”

R1.11. Language: It would be good to get a native speaker to proofread the language. In particular, the use of the apostrophe in possessives is incorrect in most instances (e.g. in the abstract: ‘adult stranger’s dyads’)

Response: The revised version of the manuscript has now been proofread by a native English speaker, and we reformulated many sentences to improve clarity.

Reviewer #2 (Remarks to the Author):

I was interested in reviewing this work on egocentric biases in mothers towards their children, which seems like a timely and relevant topic that may advance understanding in the field. However, I could not always follow the authors' reasoning. I had quite a few methodological questions that emerged during reading that made it difficult for me to evaluate this work in its entirety.

R2.1. The first and major claim of the paper seems to be that mothers display a stronger egocentric bias (EB) towards their own children's than towards other children's bodily

feelings. The results seem to support this idea, although I have to say that it was not 100% clear to me how “bodily feelings” were defined and whether the terms feelings and emotions were used synonymous here. I also have various questions with regard to the methods (see below). Second, it is implied that the mothers’ EG bias is adaptive. I do not believe that convincing evidence was presented for this claim (see also comment #21). Additionally, the authors claim to have validated a new, “bodily” emotional egocentricity task. I tend to disagree (see also the following comments #2-#5; #12-#14).

Response: We thank the reviewer for their feedback and giving us the opportunity to better explain (1) what we mean with bodily feelings and how they are defined in relationship to emotions, (2) the possibility that the biases we found are adaptive. We also respond fully below to the methodological and terminological issues the reviewer raises more specifically in his further comments below, regarding our methods and the validation of a ‘bodily’ task.

Regarding the first comment, indeed we used the word feelings and emotional feelings interchangeably to designate the pleasantness arising from the tactile stimulation, measured via the behavioural ratings of “how pleasant was the touch”. In this, we admit that we just followed this existing research tradition that refers to the exploration of “emotional egocentricity biases” and more generally uses the word emotion to designate the judgement of the self-other affective states, via pleasantness ratings (e.g. Silani et al. 2013). We are also aware that different research traditions and studies use the terms feeling, emotion and affect in different and at times inconsistent ways and hence, in hindsight we acknowledge this terminological issue in the manuscript and importantly we now define exactly what we mean by ‘bodily feelings’ in text as follows:

Page 18:

“Noteworthy, in the current paper, we adhere to the established research tradition that explores “emotional EBs” and generally employs the term “emotion” to refer to the evaluation of self-other affective states through pleasantness ratings (e.g. Silani et al. 2013). Accordingly, we employ the terms “bodily” and “emotional feelings” as designating the pleasantness arising from tactile stimulation, as measured by the behavioural ratings of “how pleasant was the touch”. Given this specific definition, the inferences that can be drawn from the present study are confined to judgments of tactile pleasantness, and do not necessarily encompass more intricate emotional states such as anger and happiness.”

We also note that we have not used the word emotion on its own, except in the title, but only in association to “emotional egocentric/altercentric biases”. In order to make it more consistent, we now changed the title to **“Mother knows best: Mothers are more egocentric towards their own child’s bodily feelings”**.

Second, regarding our claim that mother’s EB is adaptive, we toned down our interpretations and stayed closer to our findings (i.e., a positive correlation between EB and empathic concern in the ‘own’ group). In addition, and in agreement with the reviewer that our manuscript has not provided direct evidence for an ‘adaptive’ role, we have now suggested this interpretation as one possibility (i.e., anticipatory bodily and affective

regulation) among other alternative interpretations such as ‘social-referencing’ (see comment R1.3) or ‘uncertainty condition’ (see comment R3.8).

See page 17-18 for the extended discussion.

R2.2. The authors argue that they present a new “bodily” emotional egocentricity task. When reading this, I was under the impression that the original emotional egocentricity task as presented by Silani et al. (2013) did not incorporate tactile stimulation. However, this is not the case, as the original task features visuo-tactile stimulation.

Response: We thank the reviewer for this point, which is closely in line with Reviewer 1’s comment (R1.1), and, which gives us the opportunity to better explain our task, and why we believe it is a more bodily task. While we agree with the reviewer that both tasks involve visuotactile sensations, there are two important differences: (1) the nature of the seen, vicarious touch is different between the tasks, with one being embodied and the other not; (2) the tactile stimulation parameters differ between the paradigms. Both of these differences have important and well-established implications in sensory neuroscience, as we now explain in the manuscript and below.

(1) First as regards, the ‘seen touch’. There are two key reasons why our task can be considered a bodily task. We now explain each of these reasons in the text as follows:

p. 5-6:

“Dyads of participants were tested in proximal space, using a set-up that allows tactile stimulation of participant’s own unseen forearm at the same time as they are watching another participant’s forearm being touched. This approach moves beyond the utilisation of a disembodied stimulus, as seen in previous tasks, to include a vicarious touch stimulus (see also below and Figure 1). Consequently, this setup facilitates a concurrent experience of self, through unisensory touch (felt touch) and other, through unisensory observed touch (vicarious seen touch). As depicted in Figure 1, participants were asked to judge how pleasant was the touch they felt (self) or saw (other), without knowing in advance which question they would need to answer. This required their attention to be focused on both self and other experiences. This simultaneous sensory experience can either be congruent or incongruent in terms of pleasantness (pleasant or unpleasant; see the design and set-up in Figure 1).

A large tradition of sensory and multisensory cognitive and neuroscientific research has studied vicarious (or mirror) sensory phenomena. Notably, studies within this domain have highlighted the unique effects of seeing body parts being stimulated, in contrast to other non-body stimuli, as well as many other embodiment effects on vicarious touch (e.g.³¹⁻³³). For example, neuroimaging studies have identified areas within the occipito-temporal cortex, such as the extrastriate body area³⁴, and fusiform body area³⁵ responding selectively to seen bodies and body parts, and a corresponding topographic map of viewed body parts throughout occipitotemporal cortex³⁶. This specificity of the human body (both our own and that of others) as a visual stimulus and a representation, particularly during vicarious responding, is one of the two reasons we named this task the ‘bodily Emotional Egocentricity Task’. This task extends beyond ‘imagining’ the other person being touched

with a stimulus while one feels touch on their own body, it also encompasses the representation of the other person's body and the seen touch itself during the vicarious touch experience. The second, related reason pertains to the capability of our task to provide subject-specific, important control (non-concurrent) measures for both the felt and the seen unisensory perception of touch. Various studies on vicarious, or mirror perception and visual enhancement of touch have shown that there are both intersubjective and intermodal effects on concurrent perception of felt and seen touch. While traditional emotional egocentricity tasks aim to mitigate some of these effects by employing emotionally 'congruent' conditions as a baseline, these tasks still remain within the context of concurrent, multimodal conditions. By contrast, our task gives the opportunity to have this control, as well as unisensory measures for each modality, without the concurrent stimulation. Indeed, in addition to the congruent and incongruent conditions tested during the main task, we also tested unisensory (non-concurrent) perception for both the felt touch on the self and the seen touch on the other. These control conditions serve the purpose of disentangling sensory from the multisensory effects inherent to the task. Importantly, none of these conditions are susceptible to the top-down effects that object or animal stimuli could potentially exert on participants, as often encountered in the original egocentricity tasks that use pictorial stimuli for the seen touch and to characterise the felt touch"

(2) Second, as regard to the 'tactile stimulation parameters', we now explain the key difference and advance of our task as follow:

Page 6

"Furthermore, our task not only allows for participants to see the touch the other person is receiving on their body with a given material, but it also allows us to match the specific stimulation parameters of the tactile and the vicarious stimuli, i.e. the self and other stimuli. Participants are in the same room, and they can see and feel specific tactile stimuli that could be either congruent or incongruent in pleasantness, applied in a specific synchronous velocity of 3 cm/s (as done in³⁷ ; see also Methods). This sensory matching is not possible in the original task¹⁶, as participants are only presented with pictorial stimuli of 'what' touches the other person (not 'how' the other person is actually touched), while also seeing similar pictorial stimuli representing what is supposedly touching their body, as they are being touched with different materials out of sight (rather than actual spiders, swans etc). In light of the aforementioned distinctions, our task constitutes a more embodied rendition of an egocentricity task and allows us to test specifically the existence of emotional EB and AB in a bodily context."

R2.3. I did not understand why the newly added "bodily" scores are considered "bodily" as compared to the "classical" scores in this task. When looking at the methods and the calculation of the scores, it seemed clear to me all terms pertain to ratings of experienced (self) or inferred (other) pleasantness of touch, e.g. for EB:

EB = ((Incongruent Other Unpleasant-Congruent Other Unpleasant)+(-1*(Incongruent Other Pleasant-Congruent Other Pleasant)))/2

Bodily EB =

$$\frac{(|(\text{Other Incongruent Unpleasant} - \text{Only Vision Unpleasant})| + |(\text{Other Incongruent Pleasant} - \text{Only Vision Pleasant})|)}{2}$$

If I understand it correctly, for the classical EB, assumed pleasantness of touch in the congruent trials is subtracted from assumed pleasantness in the incongruent trials, whereas, for the bodily EB, the assumed pleasantness of touch before the trials (also called bodily priors) is subtracted from the incongruent trials (i.e., only the subtrahend changes). Naming one type of score “bodily” and the other “classical” seems a bit misleading to me as all scores pertain to ratings of pleasantness of bodily experiences.

Response: We thank the reviewer for this point that allows us to clarify the value of our added control conditions and the more bodily nature of our tasks and biases, as better explained now in the introduction (see response above to R2.2, but also in response to reviewer 1 that had a similar comment, i.e. R1.2). It also allows us to revise our terminology and we chose to now call the “bodily-emotional” EB/AB as “sensory-controlled” EB/AB.

Classically, based on Silani et al. 2013 study, emotional egocentric and altercentric biases are computed as the difference in pleasantness between incongruent and congruent trials. These biases can account for the differential effects of valence between such conditions but they cannot distinguish between the effects of sensory stimulation (i.e. being exposed to sensory stimuli in different modalities) and socially congruent stimulation (being exposed to both self and other stimuli), as both of these conditions share these characteristics. For example, some people may find visual stimuli far more salient than tactile stimuli. So, the difference between congruent and incongruent trials may be explained by the fact that they focused more on the seen aspects of the experience rather than the felt ones. Quantitatively, the effect may be the same with other people that focused more on the other rather than the self, but the reasons for this bias would be different. Accordingly, a further, useful, complementary way to look at emotional egocentric and altercentric biases is to compute the discrepancy between the pleasantness rating we give in incongruent (multisensory and socially concurrent stimulation) trials and pleasantness ratings given during unisensory trials (i.e. seen or felt stimulation, without the socially concurrent stimuli). In order to compute an emotional egocentric bias based on unisensory baselines, we can subtract the pleasantness for the visual unisensory condition (vision prior), to get the unique contribution of the self feeling on the EB bias. Similarly, to compute an altercentric bias based on unisensory baselines, we can subtract the pleasantness for the tactile unisensory condition (feel prior), to get the unique contribution of the vision of the other on the AB bias. For these reasons we call the new biases the “sensory-controlled biases”. Furthermore, having this added, complementary control measures this has even greater added value when testing pairs of participants that may have different roles (e.g. parent, child), age differences and familiarity differences (e.g. own-other child), with all of which possibly causing differences in unisensory and multisensory, and multiagent attention and perception.

We now explain this more clearly in the introduction as follow:

Page 6-7:

“This adapted paradigm facilitated the computation of both classical emotional EB and AB (subsequently referred to as “classical bias”, EB and AB), as well as new sensory-controlled EB and AB (subsequently referred to as “sensory-controlled bias” sEB and sAB— see Figure 1B, and Methods for details). Classically, based on the study by Silani et al. (2013), emotional egocentric and altercentric biases are computed as the difference in pleasantness between incongruent and congruent trials. A supplementary and beneficial approach to understanding these biases is to gauge the disparity between the pleasantness provided during incongruent trials (which involve multisensory and socially concurrent stimulation) and the pleasantness ratings given during unisensory trials (i.e. trials involving either seen or felt stimulation, devoid of concurrent social stimuli). To compute an emotional EB using unisensory baselines, we can subtract the pleasantness rating of the visual unisensory condition (vision prior). This subtraction illuminates the distinct and unique contribution of the self feeling to the EB bias. Similarly, to compute an AB grounded in unisensory baselines, we can subtract the pleasantness rating of the tactile unisensory condition (feel prior). This computation unveils the unique influence of observing another person’s touch on the AB. Having these added, complementary control measures becomes especially valuable when testing pairs of participants with diverse roles (e.g. parent, child), age disparities and familiarity variation (e.g. own-child or. other unfamiliar-child). Such factors can potentially yield differences in unisensory, multisensory, and multiagent attention and perception”.

R2.4. I did not understand why for the classical scores difference scores are used, but for the bodily scores absolute difference scores. Bodily EB/AB scores are thereby constrained to be positive, whereas classical EB/AB can also be negative. This also influences the interpretation of the scores, as the direction of a bias can only be disentangled when using differences scores, not absolute differences. However, just as for classical EB/AB, it may be possible for the bodily EB/AB scores that the direction of differences varies between individuals, which I think should be captured in the scores. It seems to me that the bias can only be considered egocentric when a person imposes their own experience onto the other person’s experience. For the classical EB score, this would mean that a person A overestimates the assumed pleasantness of touch for person B when person A receives a pleasant stimulation while B receives an unpleasant stimulation (“other incongruent unpleasant” condition) and underestimates the pleasantness in the “other incongruent pleasant” condition, resulting in an overall positive difference score. However, if a person answers in the opposite way (resulting in a negative score), this could probably not be considered egocentric, as the person would not use their own bodily sensations to judge the other. In essence, I think the bodily EB/AB should also be calculated using difference scores.

Response: We thank the reviewer for highlighting this difference, and in agreement with this comment we have now changed how we compute the new bodily sensory-controlled

egocentric and altercentric biases to follow the same logic as the classical one, and without using absolute scores and thus ensuring greater coherence between the two methodologies:

new calculated sensory-controlled EB = ((Incong UnP - Feeling UnP)-(Incong P - Feeling P))/2

new calculated sensory-controlled AB = ((Incong UnP - SeeingUnP)-(Incong P - Seeing P))/2

Our main findings remained the same as reported before (see Page 10 for results of Experiment 1, and Page 13-15 for results in Experiment 2).

R2.5. Overall, I was not really sure what the bodily EB/AB scores measure and why subtracting the “priors” from the response in the incongruent trials is considered an appropriate measure for a bodily bias. To me, it seems that both the “congruent” and the “only vision” condition are different ways to create a reference to which the incongruent condition can be compared. I currently cannot follow as to why the “only vision” condition is better suited to “get internal bodily biases created by affective incongruency between self and other” (p. 5) than the congruence condition. In the congruence condition, it is assumed that a person A can rely on their own experience to make a judgment about the other person B (aligned experience). In the “only vision” condition (or “prior”), it is unclear what person A’s inference is based on really. It is possible that person A still uses information about how they themselves would feel if touched by the different materials – it seems to me that we just do not know?

Response: Thank you for pointing this out, and allowing us to better explain the rationale behind the two types of biases. As explained in the response to point R2.3, the sensory-controlled EB/AB biases are taking into account sensory baselines, as explained now in the introduction page 7.

Please also note that as we now explain in the manuscript (please see above points R2.2 and R2.3) the unisensory control comparisons are complementary to the ‘congruent’ control comparison exactly because they can separate between multisensory and multiagent and unisensory, non-concurrent stimulation effects. To use the example that the reviewer suggests, while it is possible that during the “only vision” condition person A uses information about how they themselves would feel if touched by the different materials to vicariously perceive the pleasantness of touch on the other (please note that our task allows for specific, body-based vicarious touch perception as explained above, rather than just the imagination of tactile sensations based on pictures of object/animal stimuli) they are not actually touched in that same moment and hence they do not need to divide their attention, effort etc between agents or modalities. Hence, this control does not conflate the multisensory with the multiagent aspects of this experience. Please also see the following point where we explain the added value in interpretation of our novel bias measures.

R2.6. I think it would be really helpful if the authors could provide examples of how they interpret the scores. For example, if classical EB is lower than zero, I would think that a person behaves directly opposite to what is hypothesized (i.e, not egocentric). How to interpret the bodily scores (in relation to how they are calculated)?

Response: We thank the reviewer for this comment. In response to comment R2.4, we have now changed the computation of the “sensory-controlled biases” (formerly bodily scores); so, both types of biases can now be compared to 0. Indeed, a negative classical and sensory-controlled egocentric bias (EB and sEB) suggests that a person is not egocentric. The two measures complement each other, as EB control for effects of congruent conditions (with imagined stimuli for vicarious perception), and sEB controls for any differences in unisensory, non-concurrent perception (with bodily stimuli for vicarious perception and more precise matching of sensory conditions). As suggested, we now give an example for each type of egocentric bias:

Page 7:

“To facilitate the interpretation of the biases, we can provide the following examples. Firstly, for the classical EB (controlled by congruency): consider a scenario where participant A is touched by cotton (pleasant), while simultaneously participant B is touched by a sponge (incongruent, unpleasant). If participant A rates the pleasantness of the sponge as more pleasant than when both participants are touched by a sponge, this will suggest that participant A is biased by the pleasant sensation from the felt cotton when evaluating the pleasantness of the sponge touching the participant B. This inclination will manifest as a positive classical egocentricity bias. For the *sensory-controlled EB*, analogous conclusions can be drawn. However, instead of comparing incongruent to congruent conditions, incongruent stimulation is compared to vicarious perception, wherein the self is not concurrently touched (the “vision only” condition). Thus, if during the ‘incongruency’ condition, a participant judges the pleasantness for the other person as more pronounced than in the “vision only” condition, this additional pleasantness (the discrepancy) can be attributed to what the participant projects to another person during incongruent, but concurrent multiagent stimulation. This differential in pleasantness does not arise from any inherent individual differences between seeing and feeling touch per se.”

R2.7. It would be important to see the correlation between the EB/ bodily EB and the AB/bodily AB, given that both scores rely on overlapping information. More generally, I would be interested in a correlation matrix for all variables.

Response: As suggested, we have now added correlation matrices for each experiment in SM (Tables 2S, 4S, 6S, 8S).

Indeed, as expected given that both biases share some information as the Reviewer says (ratings in incongruent conditions) and we used these biases as complementary measures to inform us on different aspects of the egocentric bias (as the unshared variance depends on the control we use in each case), EB and sEB are positively correlated.

R2.8. I would like to see descriptive information (means and SDs) on how each condition was rated (before differences were computed).

Response: We have now added all the descriptive information to the SM with Mean and SDs for each condition (congruent, incongruent, pleasant, unpleasant, own vs. other child) for each experiment and each group of experiment 2 (Tables 1S, 3S, 5S, 7S).

R2.9. I would also be interested in within-person variability across trials. For example, in Experiment 1, there are 4 conditions, and there are 2 Blocks with 8 trials. This would mean that participants were subjected to each condition 4 times. In the statistical analyses, it is mentioned that scores were computed as the average of the difference. I was wondering how variable/ reliable the answers for each condition were?

Response: Even though we do agree with the reviewer that investigating variability across time is interesting, we do not believe our design was optimised to test this added issue and that an analysis of the effect of repetition on 4 trials has enough power to be conducted. But future studies should look into this question further. We have now added in the discussion a line on this nice suggestion:

Page 20: “Lastly, to date, no study has investigated the impact of time, and the stability of emotional biases over time. Future studies could be designed to address such questions, which our current design, involving only 2 to 4 repetitions per condition, did not permit.”

R2.10. Given that no power analysis was conducted, I am not sure that the sample sizes are appropriate. For example, in Experiment 2, there are only 34 mothers interacting with their own and 34 mothers interacting with another child. These sample sizes seem quite low for dyadic research questions. Sensitivity analyses could help to see what effect sizes could be detected with these sample sizes.

Response: The reviewer rightly pointed out that we did not conduct any power analyses prior to conducting the study.

However, we agree that given the computation of our new biases and our new set-up, it is adequate to conduct sensitivity analyses. Using G*power, we ran a post-hoc power analysis for the computed effect size of our main findings (Experiment 2 Mothers' group: EB in the 'own' group Vs EB in the 'other' group)). Using an effect size of cohen's $d = 0.66$ (calculated using independent t-tests, ($t=2.73$, $p<.05$, $95\%CI[0.17;1.17]$; $M_{EB_own}=16.36$, $SE=3.54$; $M_{EB_Other}=4.77$, $SE=3.53$) and $\alpha=0.05$ yielded a power of 0.76 and a critical t of 1.99. Note although the obtained power is slightly lower than 0.80 which is considered as good

sensitivity, our calculated t value is higher: $t=2.73$. Moreover, sensitivity analysis with 0.80 power and alpha of 0.05 yielded to an effect size of 0.68, while ours is 0.66. Moreover, we also ran a post-hoc sensitivity analysis using $\alpha=0.05$ and $\text{power}=0.80$ and $n=68$ yielded a critical t of 1.66 and effect size of 0.29 which are both lower than our findings. Given these results, we believe that a sample size of 34 mothers in each experimental group is leading to enough power.

We now added this information to the SM (and refer to it in the Methods section) as follow page 1:

“Using G*power, we ran a post-hoc power analysis for the computed effect size of our main findings (Experiment 2 Mothers’ group: EB in the ‘own’ group Vs EB in the ‘other’ group). Using an effect size of Cohen’s $d=0.66$ (calculated based on independent t-tests, ($t=2.73$, $p<.05$, $95\%CI[0.17;1.17]$; $M_{EB_own}=16.36$, $SE=3.54$; $M_{EB_Other}=4.77$, $SE=3.53$) and $\alpha=0.05$ yielded a power of 0.76 and a critical t of 1.99. Note although the obtained power is slightly lower than 0.80 which is considered as good sensitivity, our t value is higher: $t=2.73$; ensuring that our sample size was coherent with the expected results. Moreover, we also ran a post-hoc sensitivity analysis using $\alpha=0.05$ and $\text{power}=0.80$ and $n=68$ yielded a critical t of 1.66 and effect size of 0.29 which are both lower than our findings.”

R2.11. The description of the analyses is not detailed enough to replicate the analyses. For example, it would be important to know how the variables were coded, whether REML or FIML was used, how predictors were centered in the multilevel model, and so forth.

Response: Along the reviewer comment, we now clarified and added these details throughout the methods section. Moreover, we have added the full models to SM for full clarity (Tables 9-12S; along R2.18 comment).

Page 25:

“To examine the hypothesised disparities between biases (AB vs EB), a multilevel modelling (MLM; REML) approach was employed using R Studio (lm4). The bias score (as computed from AB and EB scores) was employed as the continuous dependent variable. Bias type (Altercentric vs. Egocentric) was included as a binary categorical fixed factor (coded with $AB=0$ and $EB=1$), while participant age in years was entered as a continuous variable. Participants were treated as a random effect in the models to account for individual variability. Notably, age was integrated into the models due to findings from previous research indicating age-related variations in emotional egocentric and altercentric biases. For Experiment 2, the statistical analysis closely paralleled that of Experiment 1, with the exception of the use of type of pairs (Own vs. Other), that was introduced as an additional fixed binary factor (coded with $Own=0$ and $Other=1$). Additional details and full model results tables can be found in SM.”

R2.12. Given that the EB/AB rely on a difference score (which can be positive or negative) and the bodily EB/AB rely on absolute differences (which can only be positive), testing whether these scores are non-zero does not appear to be equally meaningful to me. It seems somewhat unlikely to me that the average absolute differences scores for bodily AB/EB (which are, e.g., based on ratings from 0-100 in Experiment 1b) approach zero (on average). Given that participants will have some variation in their answers and given that there will be measurement error, it seems likely that some differences will emerge, and as all differences will be positive, the average score is likely to be different from zero. To me, this does not necessarily imply that these differences are inherently meaningful.

Response: Following this point and the reviewer's previous comment on using absolute scores (R2.4), we changed the way we computed the new bias scores, so this point does not apply anymore.

R2.13. For the EB/AB scores: An average score of zero for AB as in Experiment 1b (looking at Figure 2b), seems to imply here that there is a similar amount of people with negative scores as there are with positive – what does that mean? How do the authors interpret this finding? To me, it seems to show that there is no AB.

Response: As rightly pointed out by the reviewer, an average score of zero for AB means that on average in the sample of Exp1b, we observe no significant AB using inferential statistics, and confirmed by Bayes Factor ($BF_{10}=0.205$).

As a side note, and in response to R3.4, we have also conducted analyses to compare samples of Exp1a and Exp1b, to explore further the potential differences that could explain the different results. As explained in response to R3.4, we have added the results of these analyses in SM as exploratory analyses, and one sentence about these comparisons in the result section.

R2.14. The authors used the term “validation” when showing that the biases in their new task are non-zero and conclude that the task was validated. However, I think a validation would have to be more conclusive than that (also given comments #12/13) and should also be more comprehensive. I would expect more information about validity and reliability.

Response: We thank the reviewer for this point. We agree that the term ‘validity’ typically implies a research design with different and more steps, so we no longer refer to our task development results as ‘validation’ but rather as “results that are in line with previous findings that found positive EB in young adults”, and we removed any instance of “validation” from the manuscript.

For example in the abstract: “To investigate these biases and control for sensory priors, we first conducted two experiments in dyads of adult strangers (total N = 75), using to validate a novel, bodily Emotional Egocentricity Task that enables simultaneous affective tactile stimulation within a dyad.”

R2.15. The authors write that they “test our hypothesis regarding the size of the biases (AB and EB) [...]” (p.7), but I was not sure which hypothesis this pertains to/ I did not find it in the text.

Response: We thank the reviewer for flagging this ambiguity in the text. We have now clarified this and made sure we explain which hypothesis we refer to throughout the manuscript.

In the introduction page 8-9, we now reformulated the hypotheses as follow:

“We tested these biases (congruency and sensory-controlled EB and AB) in dyads of strangers (Experiment 1) and of mother-child pairs (Experiment 2) using the bodily, Emotional Egocentricity Task. In Experiment 1, we expected that, within our bodily Emotional Egocentricity Task, adults would exhibit both a larger *classical* but also a larger *sensory-controlled* EB, compared to AB. This would confirm and extend previous research conducted on dyads of adult strangers. In Experiment 2, we expected mothers to exhibit greater *classical* and *sensory-controlled* EBs compared to ABs. Additionally, we hypothesised that mothers would display greater EBs towards their own child in comparison to an unfamiliar child, given the importance of their own emotional perspective in understanding and regulating their children. It is noteworthy that our study is designed to address the aforementioned hypotheses regarding mothers’ emotional EB and AB towards their children, an area that has received limited attention in the existing literature. In contrast, several studies have been dedicated to studying egocentricity in children. Therefore, we conducted exploratory analyses within the latter category, expecting the presence of greater biases in EB¹⁹ and AB³⁹ in children. This is also expected to vary depending on the dyad composition (own mother vs. other mother) and the age of the participants¹⁹.”

And, for example in the results section page 9, we removed the wording “size” that was confusing and changed it to “Moreover, to test our hypothesis regarding the difference between the biases (*classical* AB and EB) multilevel models (MLM) were used” - to reflect the hypothesis that we expected greater EB than AB.

R2.16. It is reported that “neurotypical adults” were assessed (p. 7). If I understand it correctly, the autism questionnaire (AQ) was administered. However, a score on the AQ should not be interpreted as determining or excluding neurodiversity (since it is a self-report measure and also does not account for all types of neurodiversity, e.g., ADHD). It is also not clear whether any individuals scored above a cut-off and were excluded.

Response: We thank the reviewer for this comment, as it allows us to clarify this point in the methods section. We recruited “neurotypical adults” based on “self exclusion”, as we made clear to participants before recruiting them that exclusion criteria were any present or past neurological or psychiatric history. However, we acknowledge that we have not tested this via clinical questionnaires. We now make this point clearer in the manuscript.

See page 20:

“For all experiments, exclusion criteria included any self-reported present or past neurological and psychiatric disorders.”

Moreover, we have not used the AQ10 or the IRI as diagnostic tools, but rather as to measure interindividual differences, and to explore any potential correlations with emotional biases. Note that no cut-offs were considered and no one was excluded on that basis.”

R.2.17. Overall, I was wondering whether outliers in the data were identified. Some of the data points in the figures seem extreme (e.g., Figure 3A, EB).

Response: In experiment 1b, and Experiment 2 we used the comparison of the ratings of the unisensory Pleasant vs Unpleasant experience (Feeling tactile only), with the assumption that those who rated the unpleasant condition being more pleasant than the pleasant condition could be identified as possible ‘outliers’ (see below, and in response to R1.5). However, after removing those ‘outliers’ from the sample did not change the observed effects, and as such we decided to include the whole sample. We now report the analyses without outliers in the SM.

Regarding the ‘extreme’ point in Figure 3A it is indeed 2.5SD above the group EB average, however as before, we repeated our analysis with and without this point and results remained the same.

We have now added a paragraph on ‘Outliers detection’ in the method section (page 27) and added the results without these outliers in SM (Tables 13-17S):

Page 25-26:

“Outliers detection. For all experiments, we employed two methods to identify outliers. First, we examined participants' average response to the ‘Tactile only’ baseline unisensory pleasant condition (“Tactile only pleasant”) compared to the ‘Tactile only’ baseline unisensory unpleasant condition (“Tactile only unpleasant”). Individuals who rated the unpleasant stimuli as more pleasant than the pleasant stimuli (e.g., rating the touch of a scourer higher on average than that of a cotton ball), were highlighted as potential outliers. This examination identified 6 potential outliers only within the mother groups in Experiment 2. No outliers were identified in Experiment 1b nor in the children group in Experiment 2. In Experiment 1a, where we lacked a unisensory baseline condition, we compared the ‘self pleasant congruent’ condition to the ‘self unpleasant congruent’ condition. This comparison yielded one potential outlier. Subsequently, we conducted our analyses both with and without these potential outliers. However, the results remained

consistent, leading us to retain them in the sample to account for possible individual differences (see SM for results without outliers; Tables 13S, 15S).

Second, we also examined potential outliers, based on criteria involving score above or below 2.5SD from the group's average ratings scores across the different conditions. These analyses revealed one potential outlier in Experiment 1b, three in the mother group, and two in the children group in Experiment 2. Once again, our analyses were performed with and without these outliers, and as the observed effects remained unchanged, we decided to retain them in the sample (see SM for results without outliers; Tables 14S, 16S, 17S)."

R.2.18. It was difficult for me to follow the results. I would prefer to see the full model, probably in a table (this may be a personal preference).

Response: We now reformulated the results section and we have added the full models to SM for full clarity (Tables 9S-12S).

R.2.19. I was not sure why the authors seem to consider only the correlations with caution given the limited sample size (p. 13), but not the other results.

Response: We thank the reviewer for flagging this up and we removed this sentence from our report, as this was confusing the reader. We were mentioning this "caution" regarding the correlations in the children sample, in which we had many missing cases from the questionnaires (given data loss for questionnaires, N=28); while the sample size for the other samples (in experiment 1 and in mothers) were sufficient to conduct correlations analyses.

R.2.20. Also, with regard to the correlation between IRI-EC and classical EB, two p values are reported, and it was not clear to me which p-value the authors consider appropriate (uncorrected or corrected), and it was also unclear to me how the p-value was corrected. Finally, it was unclear to me why they controlled for age (given that there are many other potential confounds that cannot be controlled for).

Response: We thank the reviewer for highlighting these points, and we now clarify in the text. First, we controlled for false discovery rate correction using the 'BH' method of Benjamini & Hochberg (1995), which we now also report in the methods section. To avoid confusion of using two different p-values as we did in the original submission, we now report the p-values before correction and only report if it "survived" FDR corrections. See page 25: "Correlations were corrected for false discovery rate (FDR) correction using the 'BH' method of Benjamini & Hochberg (1995)"

Second, we acknowledge that there are other variables that could interact with our effects (and that could not be controlled for, as discussed now in the limitation section), but one of

the main variables that has been found in previous studies to influence emotional egocentric and altercentric biases is age (Riva et al. 2016), this is the reason why we decided to include age in our models.

R.2.21. The authors claim: “Thus, this novel finding suggests the self-other distinction has an adaptive and changing role in parenting and opens new avenues for research in the relationship between the self-other distinction, affective regulation and parental interoception.” (p. 15). It seems to me that the only finding suggesting adaptive value of EB is the correlation with the IRI-EC, which, however, turned non-significant when applying correction methods (see comment #20). I would not agree that the cited claim is warranted.

Response: We thank the reviewer for commenting on this and allowing us to explain what we mean by the use of ‘adaptive role’. We have now extended this interpretation and other potential ones in the discussion, as explained in response to R2.1, R13 and R3.8 (see page 17-18 of the discussion).

It is to note that the new calculations of the sensory-controlled biases resulted in a significant correlation between the sensory-controlled EB and the empathic concern subscale of the IRI and survived FDR corrections (i.e. $r_{(63)} = .41$, $p = 0.001$). Moreover, along comment of reviewer 1 (R1.8), we conducted two separated correlations, one per mother group (those who paired with their own child and those who were paired with a stranger) and found that the positive correlation is only significant in the ‘own group’ (EB: $r = .43$, $p = 0.02$; sEB: $r = .58$, $p < .001$).

R.2.22. Finally, coming from a different research tradition, it was difficult for me to think about bias without accuracy or “truth” (see, e.g., West & Kenny, 2011). I would think that, given that there are interindividual differences in how pleasant a touch can be perceived in general, people also differ in how pleasant they consider a “shared” or “aligned” experience (such as being touched with the same material). In a shared experience, two people’s experience may be similar, but maybe still not the same. Furthermore, projecting one’s own experience to another person could be considered egocentric even when this projection (or also termed assumed similarity) is mostly accurate. Maybe the authors could take this into consideration. See, e.g.: West, T. V., & Kenny, D. A. (2011). The truth and bias model of judgment. *Psychological Review*, 118(2), 357–378. <https://doi.org/10.1037/a0022936>

Response: We thank the reviewer for referring us to this model by West & Kenny; 2011. Indeed, in our studies we follow the term bias has been used by Silani and other researchers in the social mirroring field and less so in the tradition of perceptual metacognition and other fields that use concepts such as ‘ground truth’. Given that we focus on subjective sensory and emotional judgements, the role of ground truth is indeed rather complicated in this context and bias does not typically mean ‘inaccurate’, or ‘subjective’, but rather it is used to

characterise a difference in judgement due to specific sensory and social conditions. Using our design, we were able to control for at least two sources of this difference, i.e. for differences in unisensory versus socially incongruent experience and for differences in concurrent congruent vs incongruent experience; please see also above for detailed discussion). However, as indeed the reviewer highlights it, this is a different commitment to the notion of ground truth and accuracy considerations. It is beyond the scope of this paper to address these interesting phenomena and complex relations between truth and bias, however we now refer our readers to this paper and we discuss the above differences in the introduction.

Page 7-8:

“It is important to note that the term “bias” bears different meanings in different fields of study. The current work places itself within the social mirroring field, wherein our focus is directed towards subjective sensory and emotional judgements (but see³⁸for the links between emotional EB and AB and relative “accuracy” in the perceptual metacognition field; as well as the bias model of judgement, which extends beyond the scope of our present design).”

Reviewer #3:

The authors have built and validated a new emotional egocentricity task and, with it, examined how the egocentric and altercentric biases (EB and AB) of mothers differ between judging how their own child and an unknown child feels.

They have first replicated previously reported EB and AB effects and showed the existence of such biases for bodily signals. Then they have shown that mothers have a higher EB for their own child than for the unknown child, both for the emotional and bodily biases.

The study is well-designed and their results may have important implications for the field or even clinical parental psychology in general. But such implications require more cautious interpretations.

I don't any major concerns but a series of minor issues:

R3.1. The interpretation according to which the mothers' egocentricity bias for their children is actually adaptive is not convincing enough yet. I suggest to document further empirical and theoretical evidence in favour of this interpretation. If the authors have already made a thorough literature search then they should acknowledge is a novel and speculative interpretation that remains to be confirmed. I personally think that relying on egocentric bodily states is insufficient to provide adequate care. Children are more fragile and have more needs than adults and thus representing their bigger needs is not helped at all by relying in egocentric needs. But of course, I agree that egocentric needs can remind us of our children's needs if we haven't represented their needs yet. In any case, I find it not intuitive that the authors would expect that the mothers would show higher EB for their own child.

Response: We thank the reviewer for this comment, which goes along with comments from Reviewer 1 (R1.3) and reviewer 2 (R2.1 and R2.21). We agree with the reviewer that this is merely one interpretation among many (such as social referencing and heuristic mode) and that future studies could explore further these alternative interpretations. We have now added such alternatives in our manuscript and discussed them as hypotheses for future studies, indicated by our findings, rather than the definite interpretation of our findings. See changes in the Discussion Page 17-18.

R3.2. Biases based on internal bodily signals must be better explained. I have a hard time to grasp it despite I am familiar with the EB and AB.

Response: We thank the reviewer, and have now detailed how we computed these new bodily social emotional biases. As reported in response to the two other reviewers (R1.2 and R2.3), we revised our terminology and have chosen to now call the “bodily-emotional” EB/AB as “sensory-controlled” EB/AB.

We now explain this more clearly in the introduction as follow:

Page 6-7:

“This adapted paradigm facilitated the computation of both classical emotional EB and AB (subsequently referred to as “classical bias”, EB and AB), as well as new sensory-controlled EB and AB (subsequently referred to as “sensory-controlled bias” sEB and sAB— see Figure 1B, and Methods for details). Classically, based on the study by Silani et al. (2013), emotional egocentric and altercentric biases are computed as the difference in pleasantness between incongruent and congruent trials. A supplementary and beneficial approach to understanding these biases is to gauge the disparity between the pleasantness provided during incongruent trials (which involve multisensory and socially concurrent stimulation) and the pleasantness ratings given during unisensory trials (i.e. trials involving either seen or felt stimulation, devoid of concurrent social stimuli). To compute an emotional EB using unisensory baselines, we can subtract the pleasantness rating of the visual unisensory condition (vision prior). This subtraction illuminates the distinct and unique contribution of the self feeling to the EB bias. Similarly, to compute an AB grounded in unisensory baselines, we can subtract the pleasantness rating of the tactile unisensory condition (feel prior). This computation unveils the unique influence of observing another person’s touch on the AB. Having these added, complementary control measures becomes especially valuable when testing pairs of participants with diverse roles (e.g. parent, child), age disparities and familiarity variation (e.g. own-child or. other unfamiliar-child). Such factors can potentially yield differences in unisensory, multisensory, and multiagent attention and perception.”

R3.3. The Figure 1 must be improved: the stimuli pictures are pixelated and too small. It would be best to see a photograph of the top view (or even better: participant centred view) of the setup.

Response: We apologise for the poor quality of the figure. We have now included a high-resolution figure. Moreover, as mentioned in response to R1.4, we have added a picture of the set-up in Figure 1C, so the reader can better see that participants could not see each other's face (curtain), could only see the forearm of the other participant but could not see their own forearm.

Figure 1. (...) C. *Experimental Set-up* for Experiment 1a and 1b. D. with a pictorial representation of the set up with two participants separated by a curtain. (..)

R3.4. In study 1 second sample, they did not find a significant AB, which I agree can be due to interindividual differences but I recommend the authors to test for demographic and questionnaires data differences between the two samples. E.g., higher % of women in sample 1.

Response: We thank the reviewer for their suggestion and for allowing us to test this. To explore further the variables that could explain the different results found in Exp1a and Exp1b (i.e. significant and positive AB in Exp 1a, and no significant AB in Exp1b), we conducted several comparisons on demographics (age and gender) and questionnaires data (IRI and AQ10). We have added the results of these analyses in SM as exploratory analyses, and one sentence about these comparisons in the result section.

Page 10 in the results section:

“The difference observed between the two samples might be due to the impact of interindividual differences (i.e., difference on average age of each sample and scores on

the empathy questionnaire IRI, with participants in Exp1a being on average older and scoring higher on the IRI than participants in Exp1b, see SM).”

Page 1 of SM:

“Comparisons between Experiment 1a and Experiment 1b samples:

To explore further the variables that could explain the different results found in Exp1a and Exp1b (i.e. significant and positive AB in Exp 1a, and no significant AB in Exp1b), we conducted several independent samples tests on demographics (age and gender) and questionnaires data (IRI and AQ10). Note that for age and AQ10, Mann-Whitney U tests were conducted as the assumption of normality was violated. A threshold of $p < 0.01$ was considered as significant, to correct for multiple comparisons. A significant difference of age between the two samples was found ($U=1122$, $p < 0.001$, $r_B=0.662$), with the sample in Exp 1a being older than sample in Exp 1b (Exp 1a: $M_{age}=34.22$, $SD=10.59$; Exp 1b: $M_{age}=24.93$, $SD=7.79$). Given results by previous studies who found an effect of age on emotional egocentric and altercentric biases (Riva et al., 2016), this could explain some of the difference observed between Experiment 1a and 1b. Moreover, significant differences were found on the IRI subscales with the sample in Exp1a scoring higher on the fantasy scale (Exp1a: $M=17.69$, $SD=5.18$; Exp1b: $M=13.79$, $SD=3.59$; $t(69)=3.507$, $p < 0.001$, $d=.847$), higher on the empathic concern (Exp1a: $M=19.88$, $SD=4.66$; Exp1b: $M=14.37$, $SD=1.76$; $t(69)=6.049$, $p < 0.001$, $d=1.46$), higher on the perspective-taking subscale (Exp1a: $M=18.79$, $SD=4.34$; Exp1b: $M=15.31$, $SD=3.42$; $t(69)=3.603$, $p < 0.001$, $d=.870$) and a trend to score lower on the personal distress subscale (Exp1a: $M=11.55$, $SD=4.16$; Exp1b: $M=13.65$, $SD=3.16$; $t(69)=-2.303$, $p=0.024$, $d=-.56$). However, no significant was found on the AQ10 questionnaire ($U=497$, $p=.186$, $r_B=-.184$). Note that no difference in gender was found (64% female in Exp1a vs 60% in Exp1b).”

R3.5. In study 2, for mothers emotional biases, please report as well whether the AB and EB were significantly different from zero.

Response: We have added this to the results section, page 11:

“First, similar to Experiment 1, we ran one sample t-tests and showed that both EB and AB were different from zero (EB: $M_{EB}= 10.73$, $SD_{EB}=18.30$, $t_{(67)}=4.84$, $p < 0.001$, Cohen’s $d=0.59$, 95% CI [6.31, 15.16], $BF_{10}=2.17e+3$; AB: $M_{AB}=3.14$, $SD_{AB}=10.11$, $t_{(67)}=2.56$, $p=0.01$, Cohen’s $d=0.31$, 95% CI [0.70, 5.59], $BF_{10}=2.74$).”

and page 14:

“First, similar to Experiment 1, we ran one sample t-tests and showed that only sensory-controlled EB but not sensory-controlled AB ($sEB: M_{sEB}=8.09$, $SD_{sEB}=22.30$, $t_{(66)}=2.97$, $p=0.004$, Cohen’s $d=0.36$, 95% CI [2.65, 13.53], $BF_{10}=7.23$; $sAB: M_{sAB}=-2.91$, $SD_{sAB}=15.29$, $t_{(66)}=-1.56$, $p=0.12$, Cohen’s $d=-0.19$, 95% CI [-6.639, 0.819], $BF_{10}=0.42$) was different from zero (Figure 3.B).”

R3.6. In study 2, for mother bodily biases, I don’t follow why they didn’t test the same interaction pair x type of bias. Instead they tested effect of pair separately for EB and AB.

Response: Following this comment and comments from Reviewer 2 (R2.4), we have now used the same analysis as we did for the classical emotional egocentric and altercentric biases. New results are reported page 13-14 as follow:

“Second, MLM was used in order to test our hypothesis regarding the impact of the dyadic pair (Own vs. Other) on the sensory-controlled biases. Bias scores were served as a continuous dependent variable in a MLM, with bias type (sAB vs. sEB), mother’s age, child’s age and pairs (Own vs. Other) as fixed factors and participants as a random effect. Similar to Experiment 1, we found a main effect of sensory-controlled bias type ($b=17.84$, $SE=3.51$, $p<0.001$; $95\%CI$ [10.96;24.72]). Crucially, a two-way interaction between the sensory-controlled bias type (sEB vs sAB) and being paired with own vs. other child emerged ($b=-13.88$, $SE=4.96$, $p=.006$, $95\%CI$ [-23.71;-4.06]; see Figure 3.B and full results in Supplementary Table 11Sb). Probing this interaction using planned comparisons, revealed that this effect was driven from a larger sEB in the group that was paired with their own child as compared to the group that was paired with a stranger (i.e., other child) ($t_{(65)}=2.30$, $p=.02$, $95\%CI$ [1.51;20.30]; $M_{sEB_own}=13.42$, $SE=4.00$; $M_{sEB_Other}=2.51$, $SE=3.48$; $BF_{10}=1.49$). Similar to the classical AB, there was no significant difference in sensory-controlled AB between the two groups ($t_{(65)}=-0.69$, $p=.49$, $95\%CI$ [-12.88;6.21]; $M_{sAB_own}=-4.42$, $SE=3.32$; $M_{sAB_Other}=-1.09$, $SE=3.42$; $BF_{10}=0.32$). In addition, the difference between sensory-controlled EB and sensory-controlled AB was significant only in the group that was paired with their own child (own child: $t_{(33)}=-5.12$, $p<.001$, $95\%CI$ [-24.79;-10.88], $BF_{10}=930.36$; other child $t_{(32)}=1.11$, $p=.26$, $95\%CI$ [-11.01;3.10] $BF_{10}=0.34$).

R3.7. Line 265: “empathic concern” not “empathy concern”.

Response: We thank you for spotting this mistake, we have now corrected it.

R3.8. Another interpretation regarding the authors’ main finding is that mothers were worried to perform better with an unknown child than a known child. There is plenty of literature in social psychology showing that less uncertainty, more familiarity, more similarity, being in “comfort zone” lead us to adopt a more heuristic mode of functioning which is to rely on egocentric information. In contrast, an unknown child is more likely to activate a more controlled and effortful consideration of what the child thinks.

Response: Thank you for referring us to this possible interpretation of our findings. Indeed the reliance on egocentric information or ‘mental shortcuts’ could increase when individuals are familiar with a situation (in this case being with your own child). We added this possibility in the discussion along with other possible interpretations (see comment R1.3 and R2.21, and R3.1).

Page 18: “Alternatively, our results could pertain to the distinction between familiar situations (i.e., being paired with their own child) and unfamiliar or novel situations (i.e., being paired with an unfamiliar child). In familiar instances, the parent is accustomed to the scenario of

gauging or guessing their own feelings, potentially promoting the utilisation of egocentric information, which in turn results in a larger egocentricity bias. In contrast, in unfamiliar conditions, participants might engage in more deliberate, controlled thinking about the child's perspective, leading to reduced reliance on 'perspective shortcuts' and a diminished egocentric bias (though see 44 for the application of a heuristic mode in uncertain situations).

R3.9. Altogether, given the message the authors want to convey about mothers and their child I would recommend a more cautious tone. The authors should remind the readers that the effects are statistically significant at the group level but maybe not so significant for everyday life given the effect sizes and the high interindividual heterogeneity.

Response: We thank the reviewer for this comment and we adjusted our interpretations and toned down our claims throughout the manuscript. We also highlighted some individual differences and added this point to the discussion as follow:

Page 18:

"It is crucial to acknowledge the presence of interindividual variability observed in both the EB and AB scores. While, on average, mothers exhibited a significant and large EB, suggesting a heightened influence of their own feelings when evaluating their own child, this does not hold true for every mother. As illustrated in Figure 3, it becomes evident that some mothers even displayed a negative EB, suggesting that they were not influenced by their own feelings when their child was experiencing an incongruent sensation. These differences could potentially arise from individual factors such as heightened concerns for others, as suggested by the present study, or attachment styles that warrant further investigation in future research."

30th Oct 23

Dear Dr. Kirsch,

Your manuscript titled "Mother knows best: Mothers are more egocentric towards their own child's bodily feelings" has now been seen by our reviewers, whose comments appear below. In light of their advice I am delighted to say that we are happy, in principle, to publish a suitably revised version in Communications Psychology under the open access CC BY license (Creative Commons Attribution v4.0 International License).

We therefore invite you to revise your paper one last time to address the remaining concerns of our reviewers and a list of editorial requests. At the same time we ask that you edit your manuscript to comply with our format requirements and to maximise the accessibility and therefore the impact of your work.

Please note that it may still be possible for your paper to be published before the end of 2023, but in order to do this we will need you to address these points as quickly as possible so that we can move forward with your paper.

EDITORIAL REQUESTS:

SUBMISSION INFORMATION:

OPEN ACCESS:

Communications Psychology is a fully open access journal. Articles are made freely accessible on publication under a [CC BY](http://creativecommons.org/licenses/by/4.0) license (Creative Commons Attribution 4.0 International License). This license allows maximum dissemination and re-use of open access materials and is preferred by many research funding bodies.

For further information about article processing charges, open access funding, and advice and support from Nature Research, please visit <https://www.nature.com/commspsychol/article-processing-charges>

At acceptance, you will be provided with instructions for completing this CC BY license on behalf of all authors. This grants us the necessary permissions to publish your paper. Additionally, you will be asked to declare that all required third party permissions have been obtained, and to provide billing information in order to pay the article-processing charge (APC).

* TRANSPARENT PEER REVIEW: Communications Psychology uses a transparent peer review system. On author request, confidential information and data can be removed from the published reviewer reports and rebuttal letters prior to publication. If you are concerned about the release of confidential data, please let us know specifically what information you would like to have removed. Please note that we cannot incorporate redactions for any other reasons.

* CODE AVAILABILITY: All Communications Psychology manuscripts must include a section titled "Code Availability" at the end of the methods section. We require that the custom analysis code supporting your conclusions is made available in a publicly accessible repository at this stage; please choose a repository that generates a digital object identifier (DOI) for the code; the link to the repository and the DOI must be included in the Code Availability statement. Publication as Supplementary Information will not suffice.

* DATA AVAILABILITY:

[link redacted]

Best regards,

Jennifer Bellingtier

Jennifer Bellingtier, PhD
Senior Editor
Communications Psychology

REVIEWERS' EXPERTISE:

Reviewer 1 emotional biases
Reviewer 2 empathic accuracy
Reviewer 3 emotional biases

REVIEWERS' COMMENTS:

Reviewer #1 (Remarks to the Author):

The authors have taken all the points seriously raised by all 3 reviewers, and they have diligently addressed them. The one point that I find unclear still is that in describing the paradigm and comparing it to two earlier studies, they say that the one is more embodied and the other disembodied. What do these terms actually mean and what do they add? To say something is disembodied suggests that people were somehow removed from their bodies prior to the experiment. Assuming that was not in fact the case, what does this term mean here?

Reviewer #2 (Remarks to the Author):

I very much appreciate the very thorough rebuttal that the authors prepared. It was great to read. I am now able to gain a much better understanding of the studies.

I only have a few minor comments that came up when re-reading the paper.

1) I used [statcheck.io](https://michelenuijten.shinyapps.io/statcheck-web/) (<https://michelenuijten.shinyapps.io/statcheck-web/>) and some of the statistics came out as inconsistent, e.g., on p.16: "in the other group showing only a trend ($r(30) = -.35$, $p = .06$, $95\%CI[-0.73;0.12]$)" -> for a correlation with $N = 32$, $df = 30$ and $r = -.35$ the p value should be .04956. I think maybe the authors reported the sample size instead of the df in brackets? I think these inconsistencies should be resolved.

2) I was confused about the positive correlations between AB and EB, meaning that people who show more of an egocentric bias also show more of an altercentric bias (Tables S4 and S6). In the introduction, these two seemed more like opposites to me: (p. 3): "... Conversely, the disproportionate influence of other people's mental states on our own is termed the altercentricity bias (AB)." Maybe the authors could add an explanation of this finding.

3) In the abstract, the second sentence seems to imply that biases in parent-child dyads have not really been studied at all. Maybe the authors could be more specific here.

Reviewer #3 (Remarks to the Author):

In my opinion, the authors have addressed all raised concerns.

I have no more issues or concerns regarding the manuscript.

I thank the authors for the efforts made and for their patience.

Rebuttal Letter - COMMSPSYCHOL-22-0043C

We would like to thank the three anonymous reviewers for providing helpful critical feedback on our manuscript. We believe that we have answered all the final Reviewers' concerns and that it greatly improved the manuscript with respect to its original version.

Reviewer #1

The authors have taken all the points seriously raised by all 3 reviewers, and they have diligently addressed them. The one point that I find unclear still is that in describing the paradigm and comparing it to two earlier studies, they say that the one is more embodied and the other disembodied. What do these terms actually mean and what do they add? To say something is disembodied suggests that people were somehow removed from their bodies prior to the experiment. Assuming that was not in fact the case, what does this term mean here?

Response:

We thank the reviewer for their positive feedback.

Indeed, we agree with Reviewer 1, that the wording used page 6 could be misleading, as we did not mean that previous studies used a 'disembodied' stimulus' as an 'out of body' stimulus, rather that the stimuli that were previously used were pictorial and not embodied. Along these lines, we removed the word disembodied and reworded as follow page 5 « This approach moves beyond the utilisation of a **pictorial stimulus, as seen in previous tasks, to include a **bodily vicarious touch stimulus**. »**

Reviewer #2

I very much appreciate the very thorough rebuttal that the authors prepared. It was great to read. I am now able to gain a much better understanding of the studies.

I only have a few minor comments that came up when re-reading the paper.

Response:

We thank the reviewer for their positive feedback.

1) I used statcheck.io (<https://michelenuijten.shinyapps.io/statcheck-web/>) and some of the statistics came out as inconsistent, e.g., on p.16: "in the other group showing only a trend ($r(30)=-.35$, $p=.06$, $95\%CI[-0.73;0.12]$)" -> for a correlation with $N = 32$, $df = 30$ and $r = -.35$ the p value should be .04956. I think maybe the authors reported the sample size instead of the df in brackets? I think these inconsistencies should be resolved.

Response: We have now also run the statcheck application on our manuscript to check our statistics and have made sure to correct any inconsistencies. These were due to differences in correction, and for the correlations results, it was related to the fact that they are partial-correlations, not full correlations, and Spearman's, not Pearson's correlations (for which statcheck application is not accounting for).

2) I was confused about the positive correlations between AB and EB, meaning that people who show more of an egocentric bias also show more of an altercentric bias (Tables S4 and

S6). In the introduction, these two seemed more like opposites to me: (p. 3): "... Conversely, the disproportionate influence of other people's mental states on our own is termed the altercentricity bias (AB)." Maybe the authors could add an explanation of this finding.

Response: We understand that it might be seen at first kind of counterintuitive, and the wording might have been confusing. To avoid this confusion, we rewrote some sentences in the introduction, in particular page 3:

~~"Failure to make this distinction during incongruent experiences (for instance, when one individual feels sad while another feels happy) can result in difficulties in inhibiting one's own mental states while judging the ones of others, leading to what is known as the egocentricity bias (EB^{11,12}). This failure can also result in difficulties in inhibiting the influence of other people's mental states while judging our own, leading to what is known as the altercentricity bias (AB¹³) Conversely, the disproportionate influence of other people's mental states on our own is termed the altercentricity bias (AB¹³)."~~

Actually, altercentric and egocentric biases are not incompatible nor mutually exclusive, they are tapping into different concepts and mechanisms. Indeed, to measure altercentric biases we are asking participants to rate the pleasantness for themselves, while to measure egocentric biases we are asking them to rate pleasantness for the other. Thus, one can be influenced by their own feelings when rating others, and also be influenced by others experiences when rating their own states. Other studies have found similar correlation between EB and AB (e.g. Hoffmann et al. 2016). However, as this was not the primary nor secondary aim of the current study, we have decided to not discuss this finding.

Reference: Hoffmann, F., Banzhaf, C., Kanske, P., Gärtner, M., Bermpohl, F., & Singer, T. (2016). Empathy in depression: Egocentric and altercentric biases and the role of alexithymia. *Journal of Affective Disorders*, 199, 23-29.

3) In the abstract, the second sentence seems to imply that biases in parent-child dyads have not really been studied at all. Maybe the authors could be more specific here.

Response: We have now reformulated the second sentence of the abstract as follows: « Our emotional state can influence how we understand other people's emotions, leading to biases in social understanding. Yet emotional egocentric biases in specific relationships such as parent-child dyads, where not only understanding but also emotional and bodily regulation is key, remain relatively unexplored. »

Reviewer #3 (Remarks to the Author):

In my opinion, the authors have addressed all raised concerns.
I have no more issues or concerns regarding the manuscript.
I thank the authors for the efforts made and for their patience.

Response: We thank reviewer 3 for their positive feedback.